# Reactivation of Myc transcription in the mouse heart unlocks its proliferative capacity

Megan J. Bywater[1,2], Deborah L. Burkhart [1], Jasmin Straube[2], Arianna Sabò [3], Vera Pendino [4],
James E. Hudson[2], Gregory A. Quaife-Ryan [2], Enzo R. Porrello[5,6], James Rae[7], Robert G. Parton [7,8],
Theresia R. Kress[4], Bruno Amati [3], Trevor D. Littlewood [1], Gerard I. Evan [1,10✉] &
Catherine H. Wilson [1,9,10✉]

It is unclear why some tissues are refractory to the mitogenic effects of the oncogene Myc. Here we show that Myc activation induces rapid transcriptional responses followed by proliferation in some, but not all, organs. Despite such disparities in proliferative response, Myc is bound to DNA at open elements in responsive (liver) and non-responsive (heart) tissues, but fails to induce a robust transcriptional and proliferative response in the heart. Using heart as an exemplar of a non-responsive tissue, we show that Myc-driven transcription is re-engaged in mature cardiomyocytes by elevating levels of the positive transcription elongation factor (P-TEFb), instating a large proliferative response. Hence, P-TEFb activity is a key limiting determinant of whether the heart is permissive for Myc transcriptional activation. These data provide a greater understanding of how Myc transcriptional activity is determined and indicate modification of P-TEFb levels could be utilised to drive regeneration of adult cardiomyocytes for the treatment of heart myopathies.

[1] Department of Biochemistry, University of Cambridge, 80 Tennis Court Road, Cambridge CB2 1GA, UK. [2] QIMR Berghofer Medical Research Institute, Herston, QLD, Australia. [3] Department of Experimental Oncology, European Institute of Oncology (IEO) - IRCCS, Via Adamello 16, 20139 Milan, Italy. [4] Center for Genomic Science of IIT@SEMM, Fondazione Istituto Italiano di Tecnologia (IIT), Via Adamello 16, 20139 Milan, Italy. [5] Murdoch Children's Research Institute, The Royal Children's Hospital, Parkville, VIC 3052, Australia. [6] Department of Physiology, School of Biomedical Sciences, The University of Melbourne, Parkville, VIC 3010, Australia. [7] Institute for Molecular Bioscience, The University of Queensland, St Lucia 4072 QLD, Australia. [8] Centre for Microscopy and Microanalysis, The University of Queensland, St Lucia 4072 QLD, Australia. [9] Department of Pharmacology, University of Cambridge, 80 Tennis Court Road, Cambridge CB2 1PD, UK. [10] These authors contributed equally: Gerard I. Evan, Catherine H. Wilson. ✉email: gie20@cam.ac.uk; chw39@cam.ac.uk

Myc is a basic helix-loop-helix–leucine zipper (bHLH–LZ) transcription factor that binds preferentially to specific sequences in the genome, termed E-boxes[1], via association with its bHLH–LZ heterodimerisation partner Max[2–4]. Myc functions principally as a transcriptional activator by potentiating transcription initiation via association with various cofactors, such as TRRAP and associated histone acetyl-transferases[5–7] and facilitating productive transcriptional elongation by promoting RNA polymerase II (PolII) loading and via its association with the positive transcription elongation factor (P-TEFb)[8–11]. P-TEFb is comprised of CDK9 and Cyclin T1 that are stringently regulated by various transcriptional and post-transcriptional mechanisms[12–15], and dynamically controlled by an association with an inactivation complex comprised of 7SK snRNA, Larp7, MEPCE and HEXIM[16–18]. P-TEFb phosphorylates DSIF, NELF and serine 2 of the C-terminal domain (CTD) of paused RNA PolII leading to productive elongation[12–15].

Myc is a highly pleiotropic proto oncogene that coordinates multiple transcriptional programmes underpinning normal cell replication, differentiation, metabolism and apoptosis[2,19–22]. Together with the dedicated role in controlling the growth and division of normal cells, Myc also governs diverse cell extrinsic processes required for tissue regeneration in a manner that is tightly tailored to the tissue, in which Myc is activated[23–25]. Since deregulated and elevated Myc expression is a pervasive and causal attribute of most, perhaps all, tumours, understanding how tissue-specific responses to Myc are determined at a molecular level is imperative.

Comprehensive analysis of Myc transcriptional output in individual cell types indicates that Myc regulates the expression of thousands of genes, perhaps a third of the transcriptome[26–32]. Such studies show great diversity across experimental platforms[29,32] and hint that components of the transcriptional repertoire of Myc are highly context specific. In particular, genome-wide analysis of Myc occupancy indicates the presence of Myc on virtually all promoters with open chromatin[26,27,30,33,34], suggesting that tissue-specific variations in Myc activity arise by engagement of specific, pre-configured resident cellular transcriptional programmes. However, this large and diverse repertoire of potential Myc target genes, and the lack of comparative analysis of transcriptional responses to Myc in different tissues[28–30], have together confounded reliable identification of common and tissue-specific Myc-dependent transcriptional programmes. Moreover, recruitment of Myc to a given gene does not always correlate with its level of transcription[29,32], and binding efficiency and transcriptional outputs are influenced significantly by different levels of Myc expression[31,35,36]. Consequently, both the precise mechanism as to how Myc targets genes and what factors govern the transcriptional consequence of Myc binding in differing tissue types remains unclear.

We show that Myc transcription is dependent on the level of P-TEFb within a cell. Overexpression of both Myc and Cyclin T1 in the normally non-responsive heart facilitates Myc-driven transcription and proliferation.

## Results

**Myc-driven proliferation is tissue restricted.** To compare the responses of different tissues to the acute activation of a similar level of Myc, we generated a mouse strain in which supraphysiological levels of the switchable Myc protein c-MycER$^{T2}$ are ubiquitously expressed from a common promoter at comparable levels across tissues. In this knock-in mouse strain, the $R26^{LSL-c-MycER}$ locus ($R26^{MER}$)[36] is modified to include a $CAG$ (chicken beta actin/CMV) enhancer that augments MycER$^{T2}$ expression (Supplementary Fig. 1a). $R26^{CAG-LSL-c-MycERT2}$ mice were crossed into the $Tg(Zp3-cre)93Knw$ strain, in which Cre is active in the oocyte, efficiently

excising the $LSL$ stop cassette in all adult tissues of the resultant $R26^{CMER}$ mice[37]. Compared with the physiological level of Myc expressed in $R26^{MER/+}$ cells (heterozygous for the $R26^{MER}$ allele)[36], $R26^{CMER/+}$ cells express around eightfold higher levels of $MycER^{T2}$ RNA (Supplementary Fig. 1b) in each tissue tested. $R26^{CAG-LSL-MER}$ and $R26^{LSL-MER}$ mice were interbred to generate an allelic series of ascending levels of MycER$^{T2}$ expression ($R26^{+/+}$, $R26^{LSL-MER/+}$, $R26^{LSL-MER/LSL-MER}$, $R26^{CAG-LSL-MER/+}$, $R26^{CAG-LSL-MER/LSL-MER}$ and $R26^{CAG-LSL-MER/CAG-LSL-MER}$). To measure directly relative levels of MycER$^{T2}$ in this allelic series, mouse embryonic fibroblasts (MEFs) were isolated from each strain and the $LSL$ stop cassette excised by infection in vitro with a Cre-expressing adenovirus to engage constitutive MycER$^{T2}$ expression. Western blot analysis of cell lysates confirmed an ascending allelic series of MycER$^{T2}$ expression, with homozygous $R26^{MER}$ and $R26^{CMER}$ cells expressing twice the level of their respective heterozygous counterparts (Fig. 1a). This allelic expression series was precisely mirrored in tissues from heterozygous and homozygous $R26^{CMER}$ mice (Supplementary Fig. 1c).

To rule out the possibility that high levels of Myc might modulate the $Rosa26$ promoter—and hence elicit artefactual feedback effects—we crossed $R26^{CMER}$ mice to mice carrying a $R26^{mTmG}$ ($Gt(ROSA)26Sor^{tm4(ACTB-tdTomato,-EGFP)Luo}$) reporter allele[38]. Systemic activation of MycER$^{T2}$ in $R26^{CMER/mTmG}$ mice had no impact on expression of either $Tomato$ or $MycER^{T2}$ transcripts in any tested tissues (Supplementary Fig. 1d). Hence, elevated MycER$^{T2}$ activity does not modulate activity of the $Rosa26$ promoter.

We next determined whether acute activation of MycER$^{T2}$ elicits a proliferative response in tissues of $R26^{CMER/+}$ mice. MycER$^{T2}$ was activated for 24 h by systemic administration of tamoxifen[39] and proliferation assessed by immunohistochemical staining of Ki67, bromo-2′-deoxyuridine (BrdU) and the mitotic marker phospho-histone H3 (p-H3). We observed a consistent pattern of proliferative responses to supraphysiological Myc in tissues (Fig. 1b, c) that fell into three general classes: (1) adult tissues, such as liver, lung and pancreas with normally low levels of endogenous Myc (Fig. 1d), but capable of significant regeneration after injury. Such tissues showed a marked induction of proliferation upon Myc activation. (2) Adult tissues, such as kidney, heart and brain, with normally low levels of endogenous Myc and a limited capacity to regenerate. In these, Myc elicited only a negligible rise in proliferation. (3) Tissues with constitutively high proliferative rates and substantial constitutive levels of endogenous Myc (Fig. 1d), such as thymus and spleen, in which activation of ectopic Myc elicited no significant additional proliferation above the already high basal level. Neither sustaining MycER$^{T2}$ activation for 3 days nor increasing the expression level of MycER$^{T2}$ from one to two copies of $R26^{CMER}$ changed the proliferative responses to Myc in any tissue (Supplementary Fig. 1e–h). Hence, different tissues harbour different inherent sensitivities to Myc-driven proliferation.

**Myc binding does not correlate with efficient transcription.** There are several plausible explanations for the failure of acute Myc activation to induce a proliferative response in tissues, such as heart and brain. First, non-responsive tissues might express high levels of one or more of the Mxd proteins, which could directly antagonise Myc's transcriptional function by sequestering Max and/or competing for its binding to shared target genes. To investigate this possibility, we determined the levels of $Mxd$ expression in various tissues. Although expression of $Mxd$ family members varied across tissues (Supplementary Fig. 2a), we saw no correlation between the extent of Myc-induced proliferation and the levels of $Mxd$ transcripts. Next, we determined whether Myc is still able to bind its target genes using ChIP-sequencing analysis

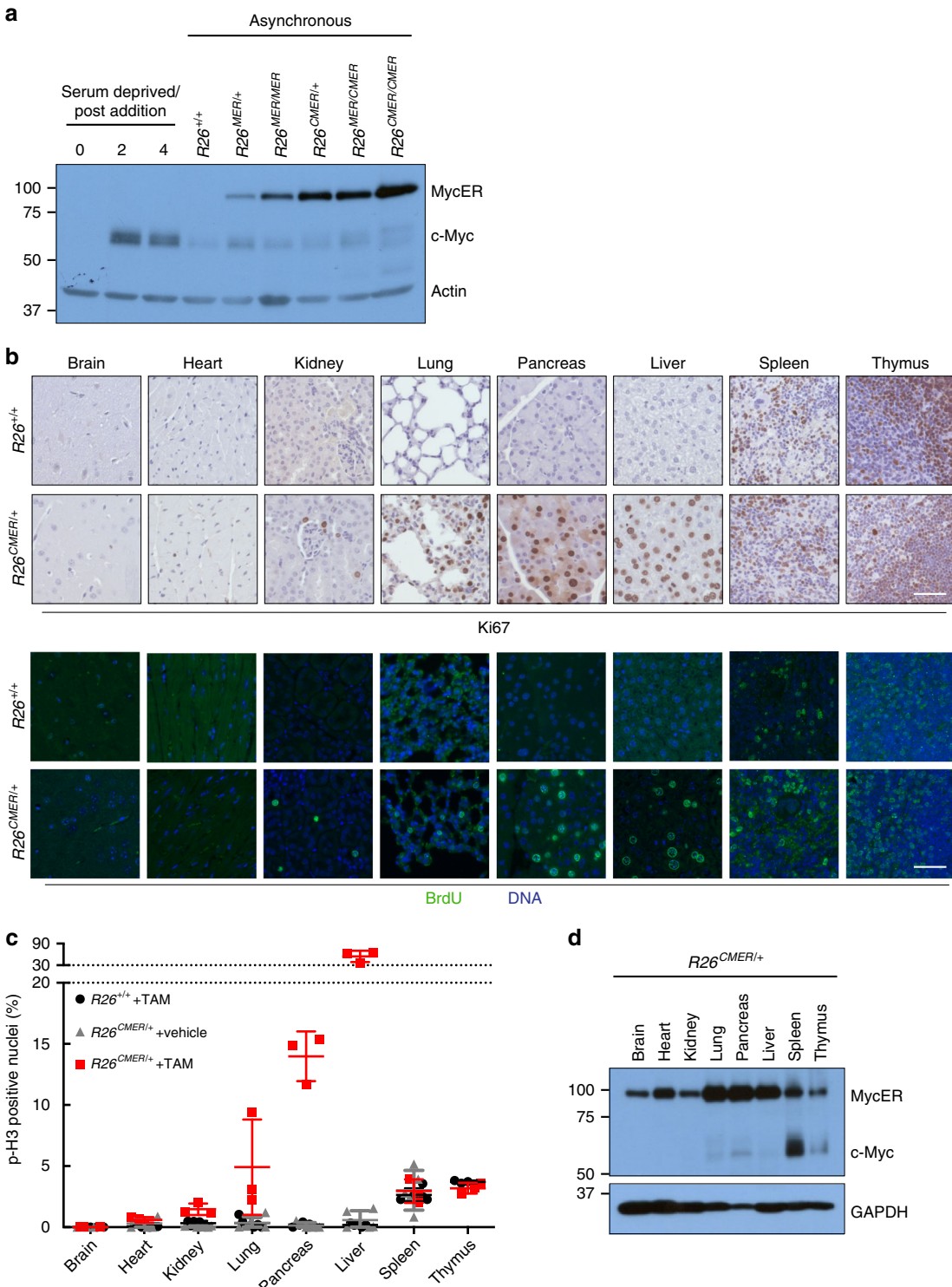

of Myc binding sites 4 h after activation in exemplar responsive (liver) and non-responsive (heart) tissue. We observed a dramatic increase in gene occupancy upon MycER$^{T2}$ activation in both tissues, from a few hundred sites in wild-type control, reflecting the low levels of endogenous Myc expression in these tissues, to ~30,000 after Myc activation (Fig. 2a). Hence, the failure of the heart to proliferate in response to Myc activation cannot be attributed to the inability of ectopic Myc to access its target genes. Around a quarter of Myc target binding sites were common to both tissues, with a significant overlap principally at promoter elements (Fig. 2b, c). Motif analysis demonstrated significant

enrichment for the E-box consensus sequences at these regions (Fig. 2d) and gene ontology indicated that these overlapping promoter binding sites located to 'classic' core Myc-dependent genes implicated in control of gene expression, mRNA processing and mitotic cell cycle (Fig. 2d). In addition to these shared targets, however, Myc also bound discrete sets of promoter elements that differed between the two tissues (Fig. 2b). These tissue-specific sites were also enriched for E-box consensus sequences but demonstrated a generally lower affinity for Myc in comparison to the shared core elements (Fig. 2e, f). Interestingly, these tissue-specific promoter elements were associated with genes whose

**Fig. 1 The proliferative response to supraphysiological Myc is very variable across different tissues. a** Immunoblot analysis of MycER[T2] and endogenous c-Myc protein levels in wild-type ($R26^{+/+}$) mouse embryonic fibroblasts (MEFs) maintained in serum-deprived media, and at the indicated type points (in hours) post addition of serum, all compared with asynchronous $R26^{+/+}$, $R26^{MER/+}$, $R26^{MER/MER}$, $R26^{CMER/+}$, $R26^{MER/CMER}$ and $R26^{CMER/CMER}$ MEFs. Expression of actin is included as a loading control. Image represents the results from six individual mice. **b** Immunohistochemical and immunofluorescence staining of Ki67 and BrdU in the brain, heart, kidney, lung, pancreas, liver, spleen and thymus isolated from wild-type ($R26^{+/+}$) and $R26^{CMER/+}$ mice 24 h post administration of tamoxifen. Representative images based on analysis of five independent mice. Scale bar represents 50 μm. **c** Quantification of p-H3-positive nuclei percentage from brain, heart, kidney, lung, pancreas, liver (hepatocytes), spleen (red pulp) and thymus isolated from oil-treated $R26^{CMER/+}$ ($n = 5$ for heart, liver, lung, kidney, spleen and pancreas, and $n = 3$ for brain) mice or wild-type ($R26^{+/+}$, $n = 5$ for heart, liver, lung, kidney and spleen, $n = 4$ for pancreas, and $n = 3$ for brain and thymus) and $R26^{CMER/+}$ ($n = 3$) mice 24 h post administration of tamoxifen. Mean of five images per mouse; mean and s.e.m shown. **d** Immunoblot analysis of MycER[T2] and endogenous c-Myc expression in the brain, heart, kidney, lung, pancreas, liver, spleen and thymus isolated from $R26^{CMER/+}$ mice. Sample loading was normalised for equal protein content, as determined by a bicinchoninic acid assay. Expression of GAPDH is included as a confirmation of efficient protein isolation. Representative results based on analysis of four independent mice.

expression defines each tissue (i.e., hepatocytes vs cardiomyocytes respectively—Fig. 2e, f), consistent with the idea that Myc binds already active/open promoters[26,27,30,32–34]. Myc ChIP-seq profiles, RNA PolII ChIP-seq in normal liver and heart, ATAC-seq in purified cardiomyocytes[40], publicly available data for chromatin marks of active transcription (H3K27 acetylation H3K4 tri-methylation and H3K4 mono-methylation) and DNAse-seq in heart and liver[41] all concurred that Myc binding closely overlaps with transcriptionally active promoter and distal elements in each tissue. This holds true across both common and tissue-specific Myc-bound sites. For example, promoters bound in the heart but not liver are characterised by absence of active chromatin marks and RNA PolII binding in the liver. Conversely, promoters bound by Myc only in liver lack active chromatin marks or associated RNA PolII in heart (Fig. 3a, b, Supplementary Fig. 3a, b). To confirm Myc binding to genes implicated in the transcriptional control of cell cycle in the heart, liver and purified cardiomyocytes, we performed sub analysis of Myc binding and open chromatin marks in a mitotic cell cycle gene set (GO: 0000278). Over 60% of these sites were marked as open and Myc bound (Supplementary Fig. 3c, d). These data confirm that Myc binds "open" chromatin at both promoters and distal sites (enhancers), and that the disposition of chromatin accessibility to Myc is pre-configured in a tissue-specific way. Together, these data indicate that adult terminally differentiated heart retains an open chromatin architecture at sites required for Myc to drive transcriptional control of proliferation and that the regenerative capacity of the heart is lost due to the transcriptional control rather than an epigenetic block.

Since Myc efficiently binds target gene DNA in both heart and liver, we investigated other potential mechanism(s) that might limit proliferative output in the non-responsive heart. We first determined the overall transcriptional output in various tissues (heart, kidney, liver and lung) of $R26^{CMER/+}$ and $R26^{+/+}$ control mice, following acute Myc activation for 4 h. Differentially expressed genes (DEGs) were called in each tissue based on the fold change (FC) in mRNA levels between $R26^{CMER/+}$ and $R26^{+/+}$ control mice (Log2FC ≥ 0.5, $q$ ≤ 0.05). In agreement with previous observations[28,30–32], the total number of DEGs in each tissue (Fig. 4a, Supplementary Data 1) was much lower than the number of Myc-bound promoters, albeit with significant inclusive overlap (Fig. 4b). Comparison of DEGs in the liver, lung, kidney and heart indicated considerable commonality among up-regulated genes (Fig. 4c) implicated in prototypical core Myc-regulated processes, such as ncRNA, rRNA and tRNA, metabolic processes (Fig. 4d). By contrast, Myc-repressed genes showed very little overlap across tissues, although gene ontology analysis did suggest that genes overlapping in two or more organs favoured involvement in negative regulation of cell movement, locomotion and migration (Fig. 4e, f). Tissue-specific genes down-regulated by Myc showed significant overlap with genes normally expressed within that

tissue, (Fig. 4g, Supplementary Data 2) and their functions were generally associated with differentiated tissue-specific processes (Fig. 4h, Supplementary Data 2).

Remarkably, DEG numbers also correlated well with each tissue's proliferative response to MycER[T2] (Figs. 4a and 1b, c): tissues with the lowest overall transcriptional changes displayed the least proliferative responses. However, while activation of ectopic Myc induced significant transcriptional changes in a greater number of target genes in liver than in heart, quantitative analysis of the RNA showed that most of the mRNAs that were induced in liver also exhibited modest, if non-significant, responses in the heart (Fig. 4i). Indeed, gene-set enrichment analysis in the heart confirmed a significant trend in the increased expression of common Myc targets, even when defined as targets commonly up-regulated in response to Myc in all tissues, excluding the heart (Fig. 4j). Hence, the transcriptional response to activation of ectopic Myc in the heart is not absent but highly attenuated and below that required to initiate cell proliferation. This refractoriness of the heart transcriptome to Myc remained evident even after prolonged (24 h) Myc activation or in response to very high Myc levels (homozygous $R26^{CMER}$ mice; Supplementary Fig. 4a, b).

**Myc transcriptional output is limited by P-TEFb.** The most plausible explanation for our data is that the attenuated transcriptional and biological response to Myc in heart is due to either active transcriptional inhibition and/or insufficiency of requisite transcriptional cofactors or machinery. Consistent with this, both the total and phosphorylated levels of the CTD of RNA PolII (Rpb1) were much lower in non-responsive heart and kidney than in Myc-responsive tissues, such as the liver and lung (Fig. 5a). Since recruitment of P-TEFb is necessary for transcriptional elongation[8–10], we next assessed the expression levels of the components of P-TEFb—CDK9 and Cyclin T1. Surprisingly, both CDK9 and Cyclin T1 proteins were present at considerably lower levels in the heart and kidney compared with lung or liver, along with components of the pause factors NELF and DSIF (Fig. 5a, Supplementary Fig. 5a). Conversely, both Larp7, a negative regulator of P-TEFb, and the 7SK short non-coding RNA that serves as the scaffold for the P-TEFb inactivation complex were relatively more abundant in the heart and kidney (Fig. 5a, Supplementary Fig. 5b). This intimated that the inability of Myc to elicit transcriptional changes in the heart arises from a tissue-specific deficiency in core transcriptional cofactors, notably P-TEFb and RNA PolII itself, that are required for efficient transcriptional initiation and elongation. Since paused RNA PolII is dependent upon P-TEFb for progression[42], we directly tested whether transcriptional elongation is generally rate limiting for Myc-driven transcription using the specific CDK9 inhibitor AZ5576 (Astra-Zeneca) to inhibit P-TEFb in liver. Wild-type and $R26^{CMER/+}$ mice were concurrently treated with 4-hydroxytamoxifen (4-OHT;

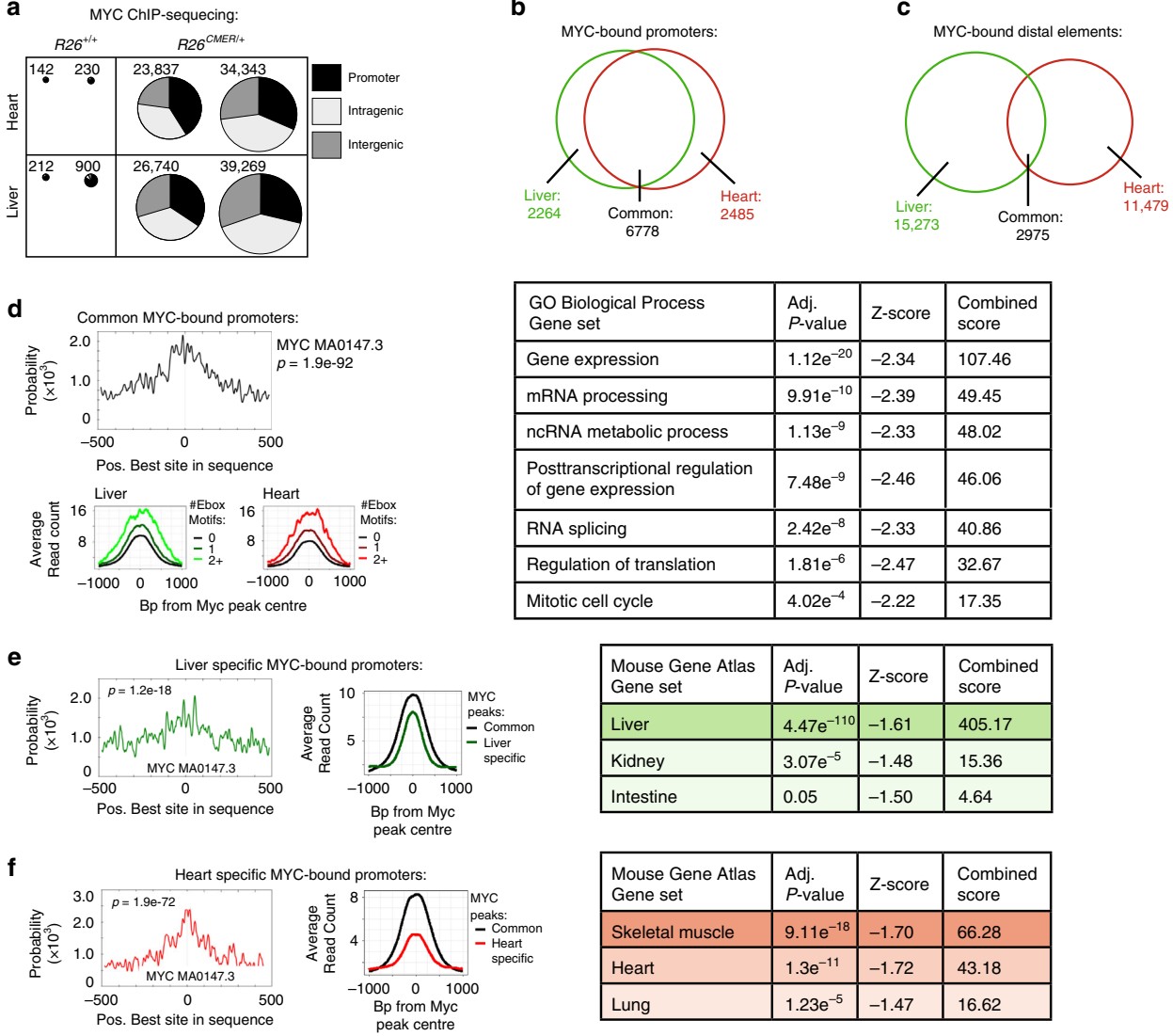

**Fig. 2 Myc binding differs between tissues. a** The number of Myc peaks and their location (promoter, intragenic and intergenic). Replicates are derived from independent mice. **b** Venn diagram of the overlap of peaks called within promoter regions (−2 kb to +1 kb from the nearest TSS). **c** Venn diagram of the overlap of peaks called within distal elements. **d** Motif probability curves for common Myc-bound promoters show the probability of an E-box consensus sequence occurring at a given position relative to the Myc ChIP peak at each common promoter site, as determined by CentriMo (top left). Average read count of Myc peaks at common promoters containing 0, 1 or >2 E-box motifs within 1000 bp from the peak centre in liver and heart chromatin (bottom left). Selected significant GO Biological Process Gene sets that overlap with common Myc-bound promoter elements (right). **e** Motif probability curves for liver-specific promoters show the probability of an E-box consensus sequence occurring at a given position relative to the Myc ChIP peak at each liver-specific promoter site, as determined by CentriMo (left). Average read count of Myc peaks shown at common (black) and liver-specific (green) promoter sites (centre). Selected significant Mouse Gene Atlas Gene sets are shown that overlap with liver-specific Myc-bound promoter elements (right). **f** As for **e** but for heart-specific Myc-bound promoters bound by Myc only in the heart. Significant Mouse Gene Atlas Gene sets are shown that overlap with heart-specific Myc-bound promoter elements (right). All analysis determined by Myc ChIP sequencing performed from hearts and livers harvested from wild-type ($R26^{+/+}$ $n = 2$) and $R26^{CMER/+}$ ($n = 2$) mice at 4 h post administration of 4-OHT. MemeSuite Centrimo computed $P$-values are derived by a binomial test. Enrichr computed $P$-value (Fisher's exact test), $Z$-score (modified Fisher's exact test to determine deviation from an expected rank) and combined score ($C=\ln(P) \times Z$) are shown. Source data are provided as a Source Data file.

to activate MycER[T2]) and AZ5576, and livers collected 4 h later. AZ5576 effectively inhibited CDK9 activity, as attested by a decrease in phosphorylated Rpb1 levels (Fig. 5b) and, notwithstanding the presence of active MycER[T2], such inhibition of CDK9 significantly attenuated transcription of *Smpdl3b*, *Cad*, *Gnl3* and *Polr3g*, genes regulated by Myc in multiple tissues (Fig. 5c). Hence, inhibition of P-TEFb, and consequential decreased phosphorylation of RNA PolII, renders liver nonpermissive for Myc response. The same was observed in MEFs

(Supplementary Fig. 5c, e). Conversely, ectopic overexpression of Cyclin T1 in adult $R26^{CMER/+}$ cardiomyocytes in vitro efficiently abrogated their normal refractoriness to Myc and permitted efficient expression of Myc target genes previously shown to be Myc unresponsive in adult heart (*Bzw2*, *Pinx1*, *Polr3d*, *St6* and *Cdc25a*; Fig. 5d, e, Supplementary Fig. 5f, g). Together, these data support the notion that transcriptional activity of Myc is limited by P-TEFb availability. Further experiments would be required to determine the mechanistic basis of this dependency.

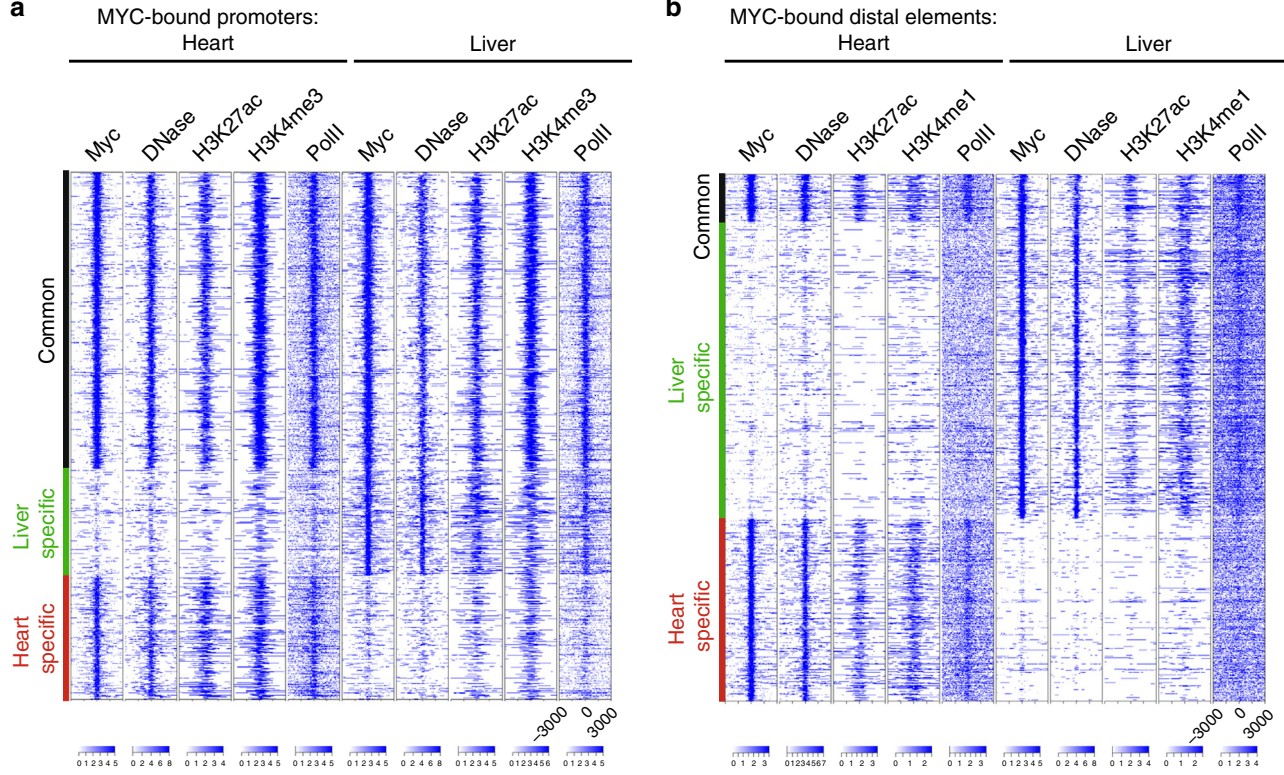

**Fig. 3 Myc binding is dictated by chromatin access. a** Heatmap of peaks called by DNAse treated, acetylated H3K27 (H3K27ac), tri-methylated H3K4 (H3K4me3) and RNA polymerase II (PolII) ChIP sequencing at Myc-bound promoter elements that are common between both the liver and heart (common-black), or specific for an individual tissue (liver specific—green, heart specific—red). Myc ChIP sequencing was performed on the heart and livers isolated from $R26^{CMER/+}$ mice 4 h post administration of 4-OHT, overlap of $n = 2$. PolII ChIP sequencing performed on the heart and livers isolated from wild-type ($R26^{+/+}$) mice, overlap of $n = 2$. ChIP sequencing data for DNAse treated, H3K27ac and H3K4me3 were taken from the ENCODE Project (accession numbers; GSM1014166, GSM1000093, GSM769017, GSM1014195, GSM1000140 and GSM769014). **b** As in **a** but for distal elements.

**Enabling Myc transcription in heart restores proliferation.**
Expression of CDK9, Cyclin T1 and RNA PolII all progressively decrease during post-natal cardiac maturation[43]. We therefore first determined the extent of Myc-dependent transcriptional responsiveness in juvenile heart tissue. Unlike the adult heart, 15-day-old juvenile hearts express levels of P-TEFb, RNA PolII comparable to those in adult liver (Fig. 6a, Supplementary Fig. 6a, b). Acute (4 h) MycER^T2 activation in $R26^{CMER/+}$ juvenile heart induced expression of a panel of common Myc transcriptional target genes (*Bzw2*, *Pinx1*, *Polr3d*, *St6* and *Cdc25a*) normally up-regulated by Myc in the adult liver but not in adult heart (Fig. 6b). RNA sequencing indicated a proficient global transcriptional response in juvenile heart (1601 DEGs up-regulated and 1133 down-regulated; Fig. 6c, Supplementary Data 3), with significant enrichment for common Myc targets (Supplementary Fig. 6c), defined from the overlap of genes up-regulated in response to the activation of Myc in all four tissues (Fig. 4c). Following 24 h of tamoxifen treatment, juvenile $R26^{CMER/+}$ mice heart exhibited a marked proliferative response as indicated by substantial induction of both G1 (*Cdk4*) and G2 (*Cdk1/Ccnb1*) cell cycle genes (Supplementary Fig. 6d), enrichment of pro-cell cycle progression gene sets (Supplementary Fig. 6e) and significant levels of mitosis (Fig. 6d). The extent of Myc-induced cardiomyocyte proliferation was comparable to that observed in tamoxifen-activated $R26^{CMER/+}$ adult liver (Fig. 6d). To confirm that Myc was acting to induce proliferation directly in cardiomyocytes themselves, we restricted expression of MycER^T2 to cardiomyocytes using the *Myh6Cre* allele (*Tg(Myh6-cre)2182Mds/J*[44]), in which Cre recombinase is driven from the *Myh6* promoter that directs expression of the Myosin heavy chain α isoform solely in cardiomyocytes. *Myh6Cre; R26^{LSL-CMER/+}* mice were generated and

at 15 days of age given a single bolus of tamoxifen to activate MycER^T2. Within 48 h, *Myh6Cre; R26^{LSL-CMER/+}* hearts displayed both p-H3 and Ki67 positivity specifically in cardiac troponin T and PCM1-positive cardiomyocytes, displayed aurora B kinase positivity at centrally located mid-bodies (Fig. 7a, Supplementary Fig. 7a) and had doubled in size compared with $R26^{LSL-CMER/+}$ controls (Fig. 7b, c). Overt cardiomyocyte division was evident from a 1.8-fold increase in cardiomyocyte number without any discernible change in cardiomyocyte size (Fig. 7c, d, Supplementary Fig. 7b–d). To further confirm cytokinesis, transmission electron micrographs (TEM) from *Myh6Cre; R26^{LSL-CMER/+}* mice 48 h post tamoxifen treatment were generated and 3D rendering used to identify cardiomyocytes in mitosis and cytokinesis (Fig. 7e, Supplementary Movies 1 and 2). Over the longer term, transient elevated MycER^T2 activity in 15-day-old *Myh6Cre; R26^{LSL-CMER/+}* mice caused a large increase in heart size and increased p-H3 immunoreactivity. Furthermore, a large proportion of cardiomyocytes displayed disassembled sarcomeres and mice did not survive beyond 4 days post MycER^T2 activation (Supplemental Fig. 7e).

To directly test if increasing P-TEFb levels within cardiomyocytes of the adult heart in vivo is permissive for efficient Myc transcriptional activation and proliferation, we infected wild type, $R26^{CMER/+}$, *Myh6Cre; R26^{LSL-CMER/+}* and $R26^{LSL-CMER/+}$ mice with an adeno-associated virus directing ectopic expression of Cyclin T1 (encoded by *Ccnt1*) or RFP under the troponin T promoter specifically in cardiomyocytes. After 4 weeks, we confirmed viral Cyclin T1 overexpression in 34% of cardiomyocytes by immunohistochemistry (Fig. 8a). As described previously[43,45] and confirmed here, elevated expression of Cyclin T1 in cardiomyocytes led to an increase in CDK9 and phosphorylated RNA PolII (S2; Fig. 8b).

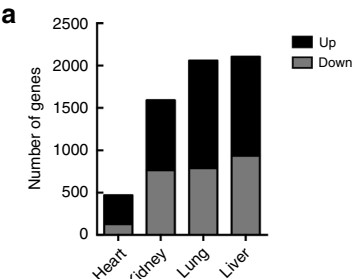

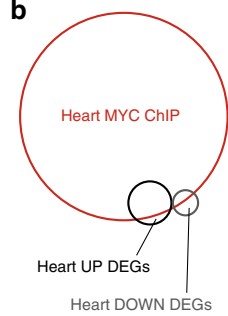

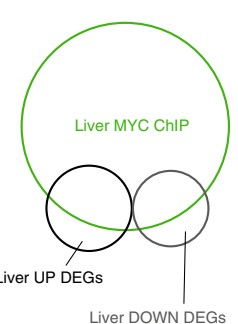

**b** Heart MYC ChIP — Heart UP DEGs — Heart DOWN DEGs — Liver MYC ChIP — Liver UP DEGs — Liver DOWN DEGs

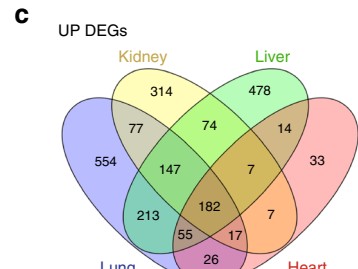

**c** UP DEGs

**d** Common UP DEGs: Biological process

| GO Biological process Gene set | Adj. P-value | Z-score | Combined score |
|---|---|---|---|
| ncRNA metabolic process | $2.61e^{-57}$ | −2.34 | 304.39 |
| ncRNA processing | $6.66e^{-48}$ | −2.30 | 250.08 |
| rRNA metabolic process | $1.51e^{-31}$ | −2.21 | 156.86 |
| rRNA processing | $1.51e^{-31}$ | −2.20 | 156.07 |
| tRNA metabolic process | $4.81e^{-23}$ | −2.24 | 115.29 |
| tRNA processing | $8.59e^{-17}$ | −2.15 | 79.50 |

**e** DOWN DEGs

**f** Common DOWN DEGs: Biological process

| GO Biological process Gene set | Adj. P-value | Z-score | Combined score |
|---|---|---|---|
| Negative regulation of cellular component movement | $6.24e^{-4}$ | −2.31 | 17.01 |
| Negative regulation of locomotion | $6.24e^{-4}$ | −2.30 | 16.98 |
| Negative regulation of cell motility | $3.29e^{-3}$ | −2.29 | 13.12 |
| Negative regulation of cell migration | $3.29e^{-3}$ | −2.29 | 13.07 |
| Negative regulation of lipid metabolic porocess | $1.09e^{-2}$ | −2.42 | 10.94 |
| Regulation of vasculature development | $1.18e^{-2}$ | −2.31 | 10.26 |

**g** Liver specitic DOWN DEGs: Mouse gene atlas

| Mouse gene atlas Gene set | Adj. P-value | Z-score | Combined score |
|---|---|---|---|
| Liver | $4.70e^{-69}$ | −1.60 | 253.18 |
| Kidney | $2.55e^{-03}$ | −1.47 | 8.83 |
| Adipose_brown | 1.00 | −1.59 | $1.44^{-10}$ |

**h** Liver specitic DOWN DEGs: Biological process

| GO Biological process Gene set | Adj. P-value | Z-score | Combined score |
|---|---|---|---|
| Monocarboxylic acid metabolic process | $2.19e^{-7}$ | −2.35 | 36.11 |
| Organic hydroxy compound metabolic process | $1.31e^{-6}$ | −2.36 | 31.91 |
| Small molecule catabolic process | $2.05e^{-5}$ | −2.35 | 25.38 |
| Sulfur compound metabolic process | $2.05e^{-5}$ | −2.32 | 25.05 |
| Alcohol metabolic process | $2.17e^{-5}$ | −2.33 | 25.03 |
| Steroid metabolic process | $2.05e^{-5}$ | −2.30 | 24.81 |

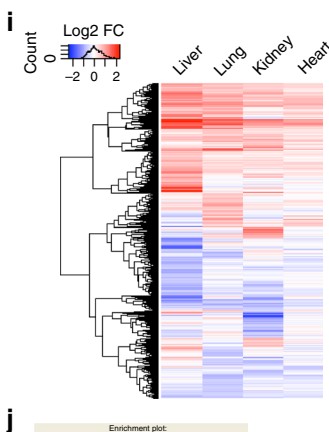

**i**

**j** Enrichment plot: BYWATER_MYC_CORE_UP_EXLCUDING_HEART — NES:2.29 — FDR:0.0 — $R26^{CMER/+}$ — $R26^{+/+}$

Acute MycER activation in the presence of overexpressed Cyclin T1 induced the transcription of a panel of common Myc target genes normally up-regulated by Myc in the liver but not in the adult heart (Fig. 8c, Supplementary Fig. 8a). RNA sequencing indicated a proficient global transcriptional response to the activation of Myc in the presence of ectopic Cyclin T1 in the adult heart or purified cardiomyocytes (Supplementary Data 4 and 5), with positive enrichment for common Myc targets and pro-cell cycle progression gene sets (Supplementary Fig. 8b). Positive enrichment of common Myc targets was also observed in cardiomyocytes purified from Ccnt1-infected Myh6Cre; $R26^{LSL-CMER/+}$ hearts vs uninfected Myh6Cre; $R26^{LSL-CMER/+}$ hearts 4 h post activation of Myc (Fig. 8d).

**Fig. 4 Myc-directed transcriptional targets differ between tissues. a** The number of differentially expressed genes (DEGs) between organs. **b** Venn diagram depicting overlap of peaks within promoter regions identified by c-Myc ChIP sequencing from heart (red) or liver (green) isolated from $R26^{CMER/+}$ mice together with genes differentially expressed in response to Myc (UP DEGs—black, DOWN DEGs—grey). Areas are proportional to the numbers of promoters or DEGs within each cohort. **c** Venn diagram depicting numbers of genes showing increased expression in response to Myc (UP DEGs) in the kidney (yellow), liver (green), lung (blue) and heart (red); (FDR < 0.05 and abs(log2FC) > 0.5). **d** Six most significant GO Biological Process Gene sets that overlap with listed genes, showing a differential increase in expression (UP DEGs) in two or more tissues (kidney, liver, lung and heart). **e** Venn diagram depicting numbers of genes showing decreased expression in response to Myc (DOWN DEGs) in the kidney (yellow), liver (green), lung (blue) and heart (red). **f** Six most significant GO Biological Process Gene sets that overlap with listed genes that show a differential decrease in expression (DOWN DEGs) in two or more tissues (kidney, liver, lung and heart). **g** Three most significant Mouse Gene Atlas Gene sets that overlap with listed genes that show a differential decrease in expression (DOWN DEGs) only in the liver (liver specific). **h** Six most significant GO Biological Process Gene sets that overlap with listed genes that show a differential decrease in expression (DOWN DEGs) only in the liver (liver specific, absent in kidney, heart and lung). **i** Heatmap showing the union of DEGs called in each tissue; liver, lung, kidney and heart. Shown are mRNA expression fold changes (Log2) upon MycER$^{T2}$ activation relative to wild type. **j** Hearts isolated from $R26^{CMER/+}$ mice show enrichment of common Myc targets (exclusive to the liver, lung and kidney) in comparison to wild type ($R26^{+/+}$). All analysis determined by RNA sequencing performed in the heart ($n = 5$), kidney, lung and liver ($n = 3$) of $R26^{+/+}$ compared to wild-type ($R26^{+/+}$) mice at 4 h post administration of 4-OHT. Enrichr computed $P$-value (Fisher's exact test), $Z$-score (modified Fisher's exact) and combined score ($C = \ln(P) \times Z$) are shown.

A short pulse of MycER$^{T2}$ activation in combination with Cyclin T1 overexpression resulted in a marked proliferative response in hearts collected at 48 h post a single dose of tamoxifen, cardiomyocytes displaying markers of cell cycle, mitosis, cytokinesis (Fig. 9a, b, Supplementary Fig. 9a, b), expression of mitosis and cytokinesis gene sets (Supplementary Fig. 9c, Supplementary Data 6), increased heart size (Fig. 9c, d, Supplementary Fig. 9d). At 72 h post MycER$^{T2}$ activation in combination with Cyclin T1 overexpression heart size was further increased along with increased cardiomyocyte number, but without any measurable change in cardiomyocyte size (Fig. 9c–e) or nucleation (Supplementary Fig. 9e). A single transient burst of MycER$^{T2}$ activity in Cyclin T1 expressing $Myh6Cre$; $R26^{LSL-CMER/+}$ adult cardiomyocytes was compatible with long-term survival, with no histological abnormalities detected at 28 days after MycER$^{T2}$ activation (Supplementary Fig. 9f). Collectively, these data illustrate that Myc transcriptional activity can be facilitated in adult cardiomyocytes by co-activation of the transcriptional co-factor P-TEFb, thus enabling adult heart tissue to re-enter the cell cycle.

## Discussion

Myc is widely acknowledged to play a pivotal coordinating role in the transcriptional control of somatic cell proliferation and tissue regeneration, and its activity is deregulated and typically elevated in the majority of aggressive cancers[2,21,22]. Past studies have largely focused on identifying Myc transcriptional programmes that are common across cell types. However, Myc exhibits great cell type variability, both with respect to how efficiently it can drive cell proliferation and the phenotypes its activities elicit in different tissues[23,25,46,47]. We developed a unique mouse model whereby a switchable variant of Myc is driven at a comparable and defined oncogenic level from an augmented version of the endogenous $Rosa26$ promoter in target cells. The variant, augmented $Rosa26$ promoter incorporates an additional $CAG$ ($CMV$ and $\beta$-actin promoter elements) that increases the expression level of Myc to supraphysiological levels frequent in many cancers. Unlike most tissue-specific promoters used to drive transgenic oncogenes, neither $Rosa26$ nor $RosaCAG$ promoters are repressed by Myc activation, obviating a major confounding artefact in most conventional transgenic models. Replacing Myc with the well validated, rapid and reversible switchable 4-OHT-dependent MycER$^{T2}$ variant allows the analysis of the immediate and direct impact of Myc on its target genome[48,49]. Inclusion of tissue-specific regulation via a Cre recombinase-excisable transcriptional STOP element provides the option for tissue restricted MycER$^{T2}$ expression.

Using proliferation as the phenotypic signature of Myc output, we identified three general classes of response to activation of oncogenic levels of Myc: tissues that proliferated in response to MycER$^{T2}$ activation; tissues that did not; and tissues that exhibited innately high endogenous Myc and proliferative indices. The correlation between proliferative response to ectopic Myc in these tissues (Fig. 1b, c) and the innate regenerative potential of each tissue is striking and suggests an underlying mechanistic connection[50].

ChIP-seq analysis 4 h post MycER$^{T2}$ activation in exemplar Myc-responsive vs non-responsive tissues (respectively, liver and heart) showed a dramatic increase in gene occupancy by Myc in both tissues. Hence, tissue non-responsiveness to Myc is not due to failure of ectopic Myc to access its chromatin targets. On the contrary, we found striking similarities between the numbers of sites to which Myc is bound across the different tissues analysed, underscoring that the phenotypic outcome effected by Myc is not solely dictated at the level of recruitment to chromatin. The significant overlap in Myc-bound promoter regions across both the liver and heart affirms the widely held notion that Myc binds a discrete set of target genes common to multiple cell types and implicated in governing core processes required for cell growth and cell cycle progression. Nonetheless, Myc also binds a set of promoter elements specific to each tissue, indicating that the overall Myc-dependent cistrome is cell type specific. We also note that this tissue-specific Myc binding repertoire is characterised by the presence of chromatin marks indicative of open chromatin, in keeping with the findings of others[26,27,30,32–34].

Interestingly, as early as 4-h post Myc activation, we clearly see a significant number of genes to be transcriptionally repressed. These repressed genes principally articulate differentiated cell type functions and show very little overlap between tissue types. Moreover, around a third (34%) of the repressed genes appear to be indirect targets, since Myc is not detected at their promoters (Fig. 4b). Suppression of differentiation-specific genes by oncogenic Myc has been widely observed irrespective of cell or tissue type and is an example of the robust inverse generic relationship between cell proliferation and differentiation[51].

It has been proposed that stringent normalisation strategies employed in the analysis of bulk RNA sequencing data can mask the more global impact of Myc on transcription that might be predicted from its promiscuous binding to open chromatin[26,27]. Such general amplification is in no way excluded by our analyses. Nonetheless, it is clear that the magnitude of transcriptional change that the activation of ectopic Myc can instruct is significantly limited in tissues that lack regenerative potential, like the heart.

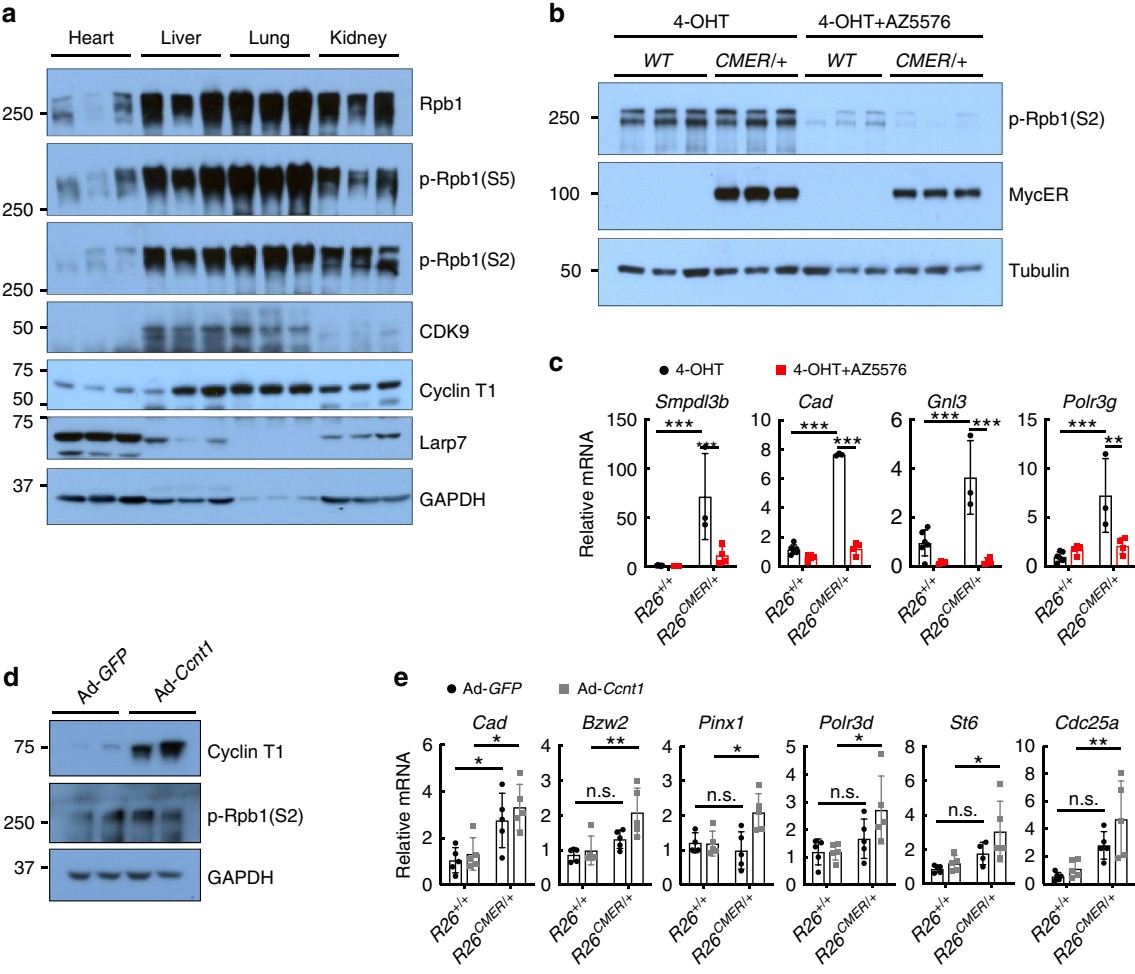

**Fig. 5 Inhibition or overexpression of P-TEFb modulates efficiency of Myc-driven transcription. a** Immunoblot analysis of the C-terminal domain (CTD) of RNA Polymerase II—total (Rpb1) and phosphorylated (p-Rpb1(S5) and p-Rpb1(S2)), CDK9, Cyclin T1 and Larp7 expression in the heart, liver, lung and kidney isolated from wild-type ($R26^{+/+}$) mice. The composite figure is generated from the same samples loaded across multiple blots and a representative image for GAPDH is shown. **b** Immunoblot analysis of the CTD of RNA Polymerase II, phosphorylated (p-Rpb1(S2)), and MycER$^{T2}$ protein expression in wild type ($R26^{+/+}$) or $R26^{CMER/+}$ livers, isolated 4 h post administration of 4-OHT either alone (4-OHT) or in combination with 60 mg/kg AZ5576 (4-OHT + AZ5576). **c** Quantitative RT-PCR analysis of *Smpdl3b, Cad, Gnl3* and *Polr3g* in wild type ($R26^{+/+}$) and $R26^{CMER/+}$ livers, isolated 4 h post administration of 4-OHT ($n = 6$ $R26^{+/+}$, $n = 3$ $R26^{CMER/+}$) either alone or in combination with 60 mg/kg AZ5576 (4-OHT + AZ5576, $n = 4$). Expression is relative to the respective wild type ($R26^{+/+}$). Mean and s.d shown. Two-way ANOVA with Tukey's multiple comparisons test; $R26^{+/+}$ vs 4-OHT: ***$P =$ 0.001 (*Smpdl3b, Cad, Gnl3* and *Polr3g*), $R26^{CMER/+}$ 4-OHT vs 4-OHT + AZ5576: ***$P = 0.001$ (*Smpdl3b, Cad* and *Gnl3*), **$P \leq 0.01$ (*Polr3g*). **d** Immunoblot analysis of Cyclin T1 and phosphorylated CTD of RNA polymerase II (p-Rpb1(S2)) in $R26^{CMER/+}$ primary cardiomyocytes infected with an adenovirus encoding either *GFP* (*Ad-GFP*) or *Ccnt1* (*Ad-Ccnt1*). Replicate samples are derived from independent primary cardiomyocyte isolations. **e** Quantitative RT-PCR analysis of *Cad, Bzw2, Pinx1, Polr3d, St6* and *Cdc25a* in wild type ($R26^{+/+}$, $n = 5$ except *Pinx1* where $n = 4$ for *Ad-GFP* control) and $R26^{CMER/+}$ ($n = 5$ except *St6* where $n = 4$ for *Ad-GFP* control) primary cardiomyocytes infected with an adenovirus encoding either *GFP* (*Ad-GFP*) or *Ccnt1* (*Ad-Ccnt1*), 4 h post addition of 100 nM 4-OHT. Expression is relative to an individual wild type ($R26^{+/+}$) *Ad-GFP* control. Mean and s.d shown. One-way ANOVA with Tukey's multiple comparisons test; Ad-GFP $R26^{+/+}$ vs $R26^{CMER/+}$: *$P = 0.05$ (*Cad*), Ad-Ccnt1 $R26^{+/+}$ vs $R26^{CMER/+}$: *$P = 0.05$ (*Cad, Pinx1, Polr3d* and *St6*) **$P = 0.01$ (*Bzw2* and *Cdc25a*). Replicate samples are derived from independent primary cardiomyocyte isolations and independent mice. Source data are provided as a Source Data file.

A simple hypothetical explanation for the pervasive suppression of differentiation by Myc is that certain essential components of the transcriptional machinery, such as P-TEFb or even total levels of Rbp1, available for loading onto promoters are limiting and that repression of differentiation-specific genes is a mere consequence of these limiting factors' being redeployed to Myc target genes, as recently proposed in 3T3 fibroblasts[11]. Analogous scarcities in transcriptional machinery could also explain why some tissues fail to respond transcriptionally to Myc. We observed that both total and phosphorylated levels of Rpb1 are significantly lower in the adult heart in comparison to the liver (Supplemental Fig. 6b). However, we presume that the levels of

Rpb1 are sufficient to allow the transcription of genes required to maintain heart function. Indeed, when we performed RNA PolII ChIP sequencing in the adult heart, we detected heart-specific genes loaded with significant amounts of RNA PolII. There are several potential mechanisms that may concentrate the available RNA PolII at specific locations. First, it is possible that the generation of "transcriptional hotspots" might concentrate essential factors at the relevant genes[52]. Second, transcription in the presence of low Rpb1 may be aided by tissue-specific super enhancers that recruits a large portion of the enhancer-associated RNA PolII and its associated cofactors and chromatin regulators[53], unfortunately, testing of these hypotheses lies outside the scope of

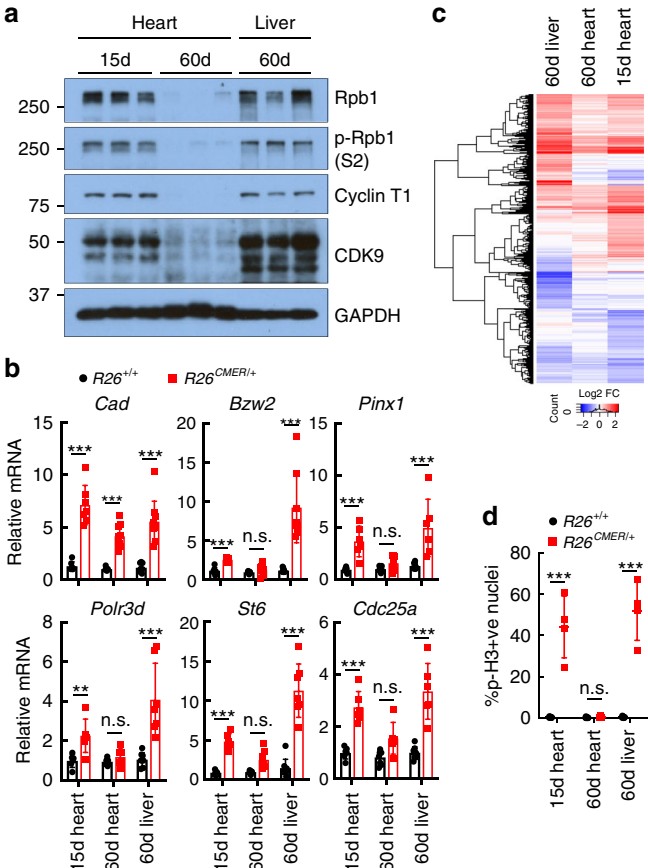

**Fig. 6 The juvenile heart is permissive to Myc-driven transcriptional activation. a** Immunoblot analysis of the C-terminal domain of RNA polymerase II, total (Rpb1) and phosphorylated (p-Rpb1(S2)), Cyclin T1 and CDK9 protein expression in wild-type heart and liver isolated from 15-day-old (15d) and 60-day-old (60d) mice. Replicate samples are derived from independent mice. **b** Quantitative RT-PCR analysis of *Cad*, *Bzw2*, *Pinx1*, *Polr3d*, *St6* and *Cdc25a* expression in wild-type ($R26^{+/+}$) and $R26^{CMER/+}$ heart isolated from 15-day-old (15d, *Bzw2* and *Cad* n = 6, *Cdc25a*, *Pinx1*, *Polr3d* and *St6* n = 7) mice vs heart (*Bzw2*, $R26^{+/+}$ *Pinx1* and $R26^{CMER/+}$ *Cad* n = 9, *Cdc25a*, *Polr2d*, *St6*, $R26^{+/+}$ *Cad* and $R26^{CMER/+}$ *Pinx1* n = 8, $R26^{CMER/+}$ *St6* n = 7, $R26^{CMER/+}$ *Cdc25a*, *Polr3d* n = 6) and liver ($R26^{+/+}$ *Bzw2*, *Pinx1*, *Cad*, *Cdc25a* and *St6* n = 9, $R26^{+/+}$ *Polr2d* n = 8, $R26^{CMER/+}$ *Cad* n = 10, $R26^{CMER/+}$ *Bzw2* n = 8, $R26^{CMER/+}$ *St6* n = 7, $R26^{CMER/+}$ *Cdc25a*, *Pinx1*, *Polr3d* n = 6) isolated from 60-day-old (60d) mice 4 h post administration of 4-OHT. Expression is relative to the respective wild type ($R26^{+/+}$). Mean and s.d shown. One-way ANOVA with Tukey's multiple comparisons test; 15 day $R26^{+/+}$ 4-OHT vs $R26^{CMER/+}$ 4-OHT: ***P = 0.001 (*Cad*, Bzw2, Pinx1, St6 and Cdc25a), 15 day $R26^{+/+}$ 4-OHT vs $R26^{CMER/+}$ 4-OHT: **P = 0.001 (*Polr3d*), adult heart $R26^{+/+}$ 4-OHT vs $R26^{CMER/+}$ 4-OHT: ***P = 0.001 (*Cad*), adult liver $R26^{+/+}$ 4-OHT vs $R26^{CMER/+}$ 4-OHT: ***P = 0.001 (*Cad*, Bzw2, Pinx1, St6 and Cdc25a). Replicate samples are derived from independent mice. **c** Heatmap showing the union of DEGs called in each tissue; adult liver (n = 3), adult heart (n = 5) and 15-day-old (15d) heart. mRNA expression fold changes (Log2) upon MycER activation are shown relative to wild type, as determined by RNA sequencing of $R26^{CMER/+}$ (n = 3) tissues relative to wild type ($R26^{+/+}$, n = 3) at 4 h post administration of 4-OHT. **d** Quantification of p-H3-positive nuclei percentage in heart (positive cardiomyocyte nuclei only) and liver (positive hepatocyte nuclei only) isolated from 15-day-old (15d) and 60-day-old (60d) wild-type ($R26^{+/+}$, n = 10 15d, n = 8 60d) and $R26^{CMER/+}$ (n = 4) mice 24 h post administration of tamoxifen. Means are taken from five images per mouse; mean and s.d. shown. One-way ANOVA with Tukey's multiple comparisons test; $R26^{+/+}$ 4-OHT vs $R26^{CMER/+}$ 4-OHT: ***P = 0.001 (15-day heart and adult liver). Source data are provided as a Source Data file.

this manuscript. Myc has been shown to modify both RNA PolII loading and its phosphorylation by P-TEFb (refs. [8–12,42,54]). We showed that responsiveness of individual tissues to Myc correlates with the levels of expression of the basal transcription factors, P-TEFb and RNA PolII, the pause factors, DSIF and NELF, and, inversely, with the expression of the P-TEFb repression complex.

Adult mammalian heart is a prototypical, terminally differentiated[55], Myc non-responsive tissue in which Cyclin T1 is known to be a limiting transcriptional regulator, and whose chronic ectopic overexpression elicits promiscuous RNA polymerase activity and cardiac hypertrophy[43,56]. While P1 neonatal heart retains proliferative potential, this rapidly declines after birth and is largely lost by P7 (refs. [57–59]). This decline in neonatal cardiac proliferative potential correlates with down-regulation of multiple genes involved in cell cycle[40,55,60–63], including endogenous Myc, which is high at P1 but significantly reduced by P5 (Supplementary Fig. 6a). Although changes in microRNAs (refs. [59,64]), modulation of the Hippo pathway[65,66], and wholesale metabolic switching from glycolysis to oxidative phosphorylation[67,68] have all been implicated, the mechanism underlying post-natal cardiac maturation and loss of regenerative potential remains unknown. Interestingly, both neonatal mouse heart and the zebrafish larval heart express high levels of P-TEFb. However, whereas the zebrafish adult heart retains 70% of its larval levels of P-TEFb and remains competent to regenerate throughout development, P-TEFb levels in the non-regenerative adult mouse heart drop by 85% (refs. [43,69]). Little is known regarding the regulation of P-TEFb during mammalian cardiac development. Levels of Cyclin T1 are regulated at both transcriptional and post-transcriptional levels[12–15], with mitogens and cytokine signalling known to increase Cyclin T1 protein stability[70,71]. We and others have shown that the level of Cyclin T1 is the key factor regulating the level and activity of P-TEFb within cardiomyocytes[43,45]. Oncogenic Ras signalling increases Cyclin T1 in the heart, leading to cardiac hypertrophy[56]. It is tempting to speculate that the noted oncogenic cooperation between Ras and Myc may, in part, arise through Ras-dependent induction of P-TEFb, so promoting the efficiency of Myc-driven transcription.

Re-establishing regenerative proliferation in the adult heart has proven very difficult. Inhibition of Hippo signalling, enforced expression of both G1 (CDK4 and Ccnd1) and G2/M (CDK1 and Ccnb1) factors, and hypoxia all elicit modest regeneration[66,72–74], but at levels well below those of P1 neonatal heart. While the adult heart is refractory to Myc-induced proliferation, enforced ectopic Myc expression during neonatal life did extend the window of neonatal cardiomyocyte proliferation up to around P15, a period that that coincides with high endogenous cardiac P-TEFb expression. The large increase in cardiomyocyte number and aurora B localisation to centrally located mid-bodies suggests that cardiomyocytes isolated from juvenile mice can progress through the cell cycle and many can complete cytokinesis. Remarkably, we show that an increase in the level of P-TEFb in the adult heart is sufficient to support ectopic Myc-dependent transcription, proliferation and cytokinesis of cardiomyocytes in vivo, resulting in increased heart size and cardiomyocyte number within a very short timeframe. Importantly, this transient wave of proliferation induced in a smaller fraction of total cardiomyocytes than in juvenile mice is compatible with long-term survival. It will be of great interest to determine if this genetic combination will prove beneficial in models of cardiac injury.

Despite the significant changes in regenerative capacity and responsiveness to Myc that attend cardiac development from embryonic stem cells through mesoderm to precursor cardiomyocyte to cardiomyocyte, the chromatin status at promoters for genes involved in basic cellular function, including cell cycle regulation, show very little variation[75,76]. Similarly, the Myc binding

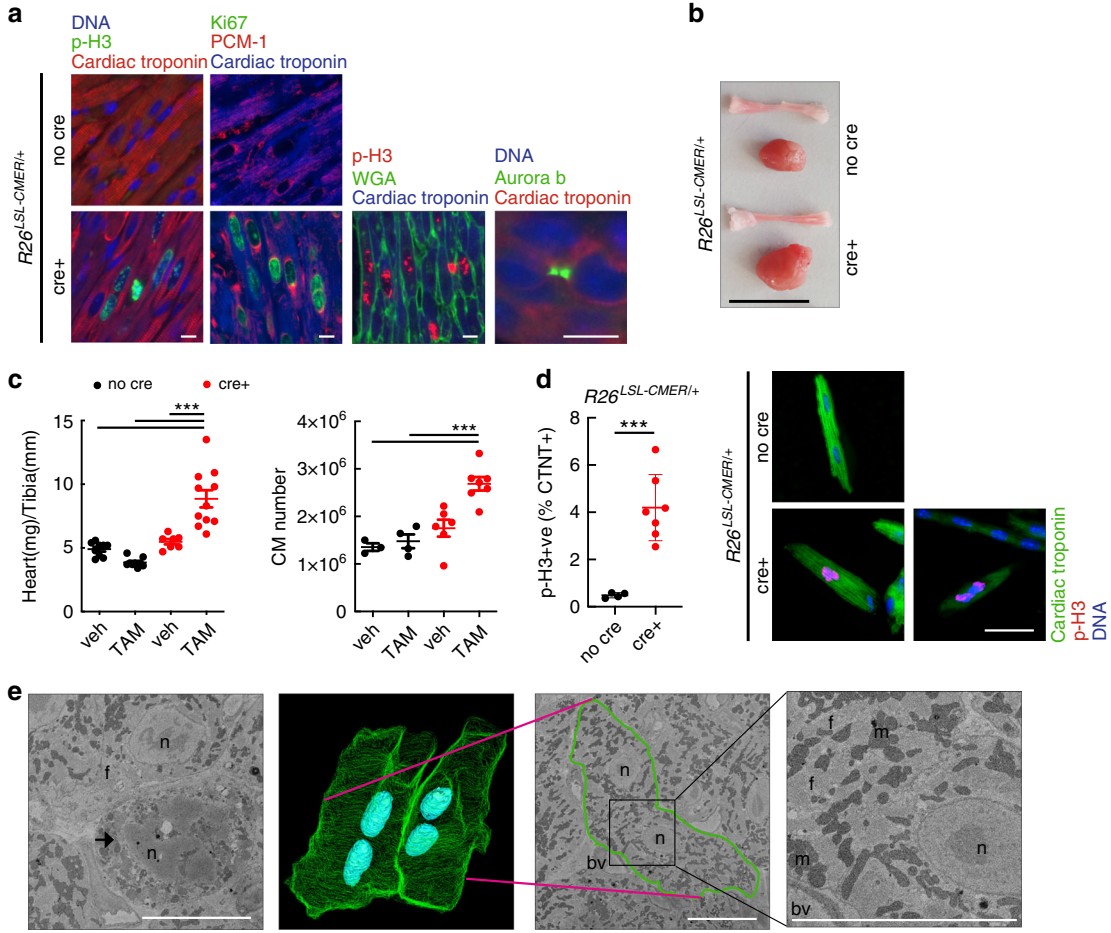

**Fig. 7 The juvenile heart is permissive to Myc-driven proliferation. a** Immunofluorescent staining of cardiac troponin, p-H3, Ki67, PCM1, wheat germ agglutinin and aurora B-positive mid-body in the heart of 15-day-old control (no cre) and *Myh6-Cre*; *R26^{LSL-CMER/+}* (cre+) mice. Representative images based on analysis of five independent mice. Scale bar represents 10 µm. **b** Image of the whole heart and a tibia isolated from a 15-day-old control (no cre) and *Myh6-Cre*; *R26^{LSL-CMER/+}* (cre+) mice. Scale bar represents 10 mm. Representative images based on analysis on 21 independent mice. **c** The weight (mg) of hearts (left, *R26^{LSL-CMER/+}*; no cre, vehicle = 9, no cre tamoxifen = 11; *R26^{LSL-CMER/+}*; cre+ vehicle n = 7, cre+ tamoxifen n = 10) and quantification of the number of cardiomyocytes (right) isolated from 15-day-old control (no cre, vehicle = 3, no cre tamoxifen = 4) and *Myh6-Cre;R26^{LSL-CMER/+}* (cre+ vehicle n = 6, cre+ tamoxifen n = 7) mice post administration of tamoxifen or oil (veh), expressed as fold change over the length (mm) of a tibia isolated from the same mouse. Mean and s.e.m. shown. One-way ANOVA with Tukey's multiple comparisons test; vehicle cre vs tamoxifen cre+, vehicle no cre vs tamoxifen cre+, tamoxifen no cre vs tamoxifen cre+ ***P = 0.001 (weight). Vehicle cre vs tamoxifen cre+ **P = 0.01, vehicle no cre vs tamoxifen cre+, tamoxifen no cre vs tamoxifen cre+ ***P = 0.001 (cardiomyocyte number). Replicate samples are derived from independent mice. **d** Flow cytometry quantification of p-H3 in cardiac troponin T-positive cardiomyocytes (left) and immunofluorescent staining (right) of cardiac troponin T (green), p-H3 (red) and DNA (blue) from 15-day-old control (no cre, n = 4) and *Myh6-Cre*; *R26^{LSL-CMER/+}* (cre+, n = 7) mice. Mean and s.d. shown. Unpaired *t*-test; no cre vs cre+ ***P = 0.0006. Replicate samples are derived from independent mice. Scale bar represents 50 µm. **e** Three-dimensional render (second from left) generated from 500 image z-stack (50 µm) of transmission electron micrographs from a *Myh6-Cre*; *R26^{LSL-CMER/+}* heart. Green shows outline of cells, blue shows nuclei, pink line shows site of cross section through cells, outlined in green (second from right) and expanded (right). Imaging performed on a single heart. Left image shows mitotic nuclei, arrow. Structures are indicated; f, myofibrils; n, nucleus; m, mitochondria; bv, blood vessel. Scale bars represent 5 µm. All analysis 48 h post administration of tamoxifen. Source data are provided as a Source Data file.

sites of the adult liver (proliferative) and the adult heart (non-proliferative) exhibit significant overlap, indicating that even after its maturation, adult, terminally differentiated heart retains an open chromatin architecture at sites required to drive transcriptional control of proliferation. Hence, if Myc could be induced by mitogenic stimulus in the heart, cardiac epigenome architecture does not preclude heart regenerative capacity: rather, it is the inability of Myc to drive transcriptional output from regenerative target genes (due to lack of the P-TEFb) that thwarts cardiomyocyte proliferation.

Overall, our data indicate that tissue regenerative capacity is tightly linked to the capacity of that tissue to respond to Myc and that tissue Myc responsiveness is governed principally by

availability of key components of the core transcriptional machinery, which Myc co-opts to drive its regenerative biological output. However, in addition to its core target genes, which are common across tissues, a significant number of its targets are expressed in a tissue-specific manner. By contrast, tissue-specific accessibility of target genes to Myc appears to be dictated principally by hard-wired, tissue-specific epigenome organisation. Our current study focuses in the main on liver and heart as, respectively, prototypical examples of regenerative and Myc-responsive vs non-regenerative and Myc-unresponsive tissues. However, it seems reasonable that our findings could be extrapolated more generally to offer insights into both regenerative medicine and cancer susceptibility.

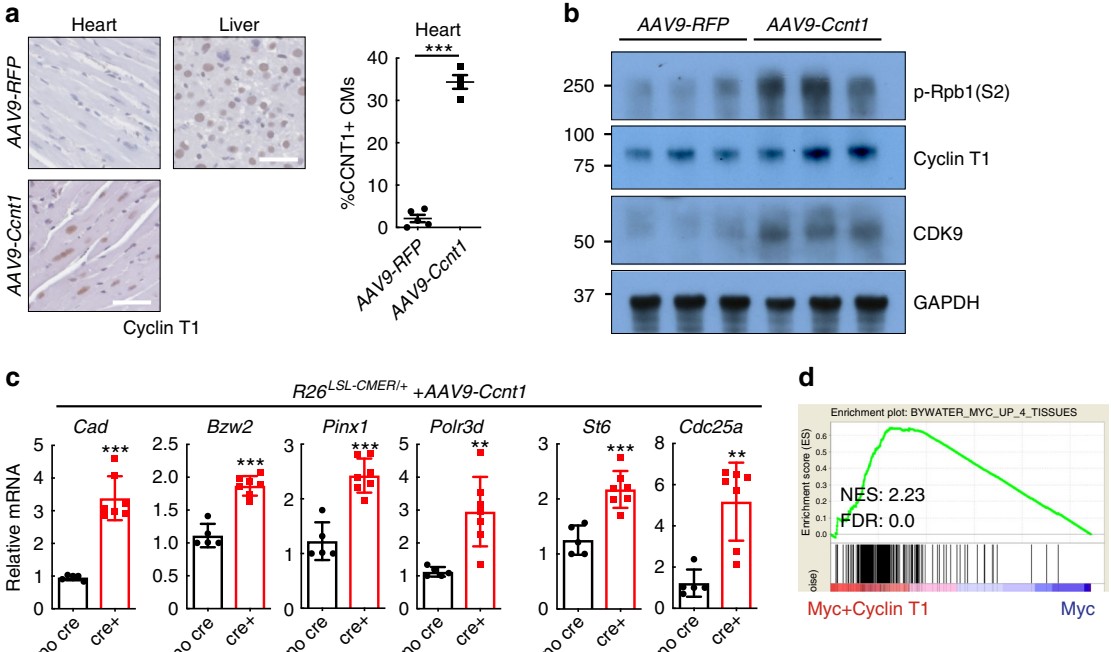

**Fig. 8 Overexpression of Cyclin T1 in the adult heart facilitates Myc-driven transcription in adult cardiomyocytes. a** Immunohistochemistry (left) and quantification (right) of Cyclin T1 in the liver isolated from adult $R26^{CMER/+}$ or heart isolated from $Myh6$-$Cre$; $R26^{LSL-CMER/+}$ mice systemically infected with adeno-associated virus (serotype 9, $1 \times 10^{11}$ vg/mouse) encoding either $RFP$ ($AAV9$-$RFP$, $n = 5$) or $Ccnt1$ ($AAV9$-$Ccnt1$, $n = 4$) 4 weeks post infection. Mean and s.e.m shown. Unpaired $t$-test; $AAV9$-$RFP$ vs $AAV9$-$Ccnt1$: ***$P < 0.0001$. Representative images based on analysis of five independent mice. **b** Immunoblot analysis of the C-terminal domain of phosphorylated RNA polymerase II (p-Rpb1(S2)), CDK9 and Cyclin T1 expression in the heart isolated from adult $Myh6$-$Cre$; $R26^{LSL-CMER/+}$ mice systemically infected with adeno-associated virus encoding either $RFP$ ($AAV9$-$RFP$) or $Ccnt1$ ($AAV9$-$Ccnt1$) 4 weeks post infection. Replicate samples are derived from independent mice. **c** Quantitative RT-PCR analysis of $Cad$, $Bzw2$, $Pinx1$, $Polr3d$, $St6$ and $Cdc25a$ expression in control ($R26^{LSL-CMER/+}$; no cre, $n = 5$) and $Myh6$-$Cre$; $R26^{LSL-CMER/+}$ ($R26^{LSL-CMER/+}$; cre+, $n = 7$) adult mouse heart isolated 4 weeks post systemic infection with an adeno-associated virus encoding $Ccnt1$ ($AAV9$-$Ccnt1$) and 4 h post administration of 4-OHT. Expression is relative to the respective wild type ($R26^{LSL-CMER/+}$). Mean and s.e.m shown. Unpaired $t$-test; no cre vs cre+: ***$P < 0.0005$ ($Cad$, Bzw2, $Pinx1$ and $St6$), **$P = 0.0013$ ($Cdc25a$), **$P = 0.0035$ ($Polr3d$). Replicate samples are derived from independent mice. **d** Enrichment of common Myc targets (liver, lung, kidney and heart) in cardiomyocytes isolated from the heart of AAV-CCNT1 infected $Myh6$-$cre$; $R26^{LSL-CMER/+}$ (Myc+Cyclin T1) mice in comparison to uninfected $Myh6$-$cre$; $R26^{LSL-CMER/+}$ mice (Myc) at 4 h post administration of 4-OHT, as determined by RNA sequencing. Including normalised enrichment score (NES) and FDR $q$-value (FDR).

## Methods

**Mice and mouse maintenance**. All experimental procedures received ethical approval and were conducted in accordance with the Home Office UK guidelines, under project licences 70/7586, 80/2396 (G.I.E.), which were evaluated and approved by the Animal Welfare and Ethical Review Body at the University of Cambridge. All animals were kept under SPF conditions. Mice were maintained on regular diet in a pathogen-free facility on a 12 h light/dark cycle with continuous access to food and water. Mouse strains, $Tg(Zp3$-$cre)93Knw$, $Gt(ROSA)26Sor^{tm4}$ $^{(ACTB-tdTomato,-EGFP)Luo}$ and $Tg(Myh6$-$tTA)6Smbf/J$ were obtained from the Jackson Laboratory. For generation of the $R26^{CAG-LSL - c-MycER}$ mice, $MycER^{T2}$-$IRES$-$eGFP$ cDNA was cloned into pBigT, and then into the $pROSA26CAG$-$PAS$ vector[77] to create the final $R26^{CAG-LSL-c-MycER}$ targeting vector. This vector was then linearised and introduced into mouse embryonic stem cells. Germ-line transmission was confirmed by PCR and verified by Southern analysis. Adult animals requiring 4-OHT were either i.p. injected with 1 mg (Z)-4-OHT (Sigma, H7904) in 10% ethanol and vegetable oil (5 mg/ml), and collected 4 h post injection or with 1 mg tamoxifen (Sigma, T5648) in 10% ethanol and vegetable oil (10 mg/ml) twice over a 24-h period and collected at 24–72 h post initial i.p. injection. Adolescent animals were given 0.5 mg of either 4-OHT or tamoxifen per injection. For 48 or 72-h collection points, mice were i.p. injected once with either 0.5 mg (adolescent) or 1 mg (adult) tamoxifen in 10% ethanol and vegetable oil. 5-BrdU (Sigma, B9285) was administered by i.p. injection (100 µl, 10 mg/ml) 2 h before culling. AZ5576 was administered to mice at 60 mg/kg in 20% Captisol® Solution by oral gavage.

**Mouse genotyping**. Ear biopsies were collected from 2–5-week-old mice and digested overnight at 55 °C in 10% Chelex 100 Resin (Bio-Rad Catalogue #142-1253), 0.1 % Tween-20 and 0.25 mg/ml Proteinase K (Sigma P8044). A typical PCR reaction was performed using GoTaq (Promega) with 1 µl of Chelex extracted gDNA following manufacturers' instructions. The following PCR conditions were applied: 5 min, 95 °C initial denaturation; 35 cycles of 30 s at 95 °C, 30 s at 60–65 °C and 1.5 min at 72 °C, followed by a final 5 min at 72 °C. PCR amplification

products were analysed by agarose gel electrophoresis. General primers for $Rosa26CAG$ were used to detect $R26^{CMER}$ and $R26^{mTmG}$ alleles, were universal forward: 5'-CTCTGCTGCCTCCTGGCTTCT-3' wild-type reverse: 5'-CGAGG CGGATCACAAGCAATA-3' and CAG reverse: 5'-TCAATGGGCGGGGGTCG TT-3'. General primers for Rosa26 used to detect the $R26^{MER}$ allele were R26F: 5'-AAAGTCGCTCTGAGTTGTTAT-3' R523: 5'- GGAGCGGGAGAAATGGATA TG-3' and R1295: 5'-GCGAAGAGTTTGTCCTCAACC-3'. Generic primers to recognise Cre were Cre1: 5'-GCTGTTTCACTGGTTATGCGG -3' and Cre2: 5'-TTGCCCCTGTTTCACTATCCAG -3'.

**Mouse embryonic fibroblasts**. MEFs were generated from embryos 13.5 d.p.f. and cultured in DMEM (Thermo Fisher 41966052) supplemented with penicillin–streptomycin (Thermo Fisher, 15140-122), L-glutamine (Thermo Fisher, 25030-024) and 10% BGS (Hyclone SH30541.03HI). All experiments were performed between passage 3 and 5. For experiments, MEFs were plated at $5 \times 10^5$ cells per 6 cm culture dish in 10% BGS (where appropriate, plus adenovirus). To render cells quiescent, the next day cells were washed twice with PBS and cultured in media containing 0.1% BGS for 48 h. After 48 h serum deprivation, cells were treated with 100 nM (Z)-4-OHT (Sigma, H7904) in ethanol to activate $MycER^{T2}$ and 1 µM AZ5576 (AstraZeneca) in DMSO to block CDK9 and RNA, or protein harvested 4 h later or cell cycle analysis performed after 16 h.

**Cardiomyocyte purification and culture**. Cardiomyocytes CMs were isolated from adult mouse hearts (8–10 weeks of age) using a simplified Langendorff-free method[78] and cultured in M199 (Sigma, M4530) supplemented with 0.1% BSA (supplier, number), 1X ITS Liquid Media Supplement (Sigma, I3146), 10 mmol/l BDM (Sigma, B0753), 1X Chemically Defined Lipid Concentrate (Fisher, 11548846) and penicillin–streptomycin (Thermo Fisher, 15140-122). When needed, adenovirus was administered to CMs in culture media immediately after removal of plating media. Medium was then replaced after 24 h with culture media

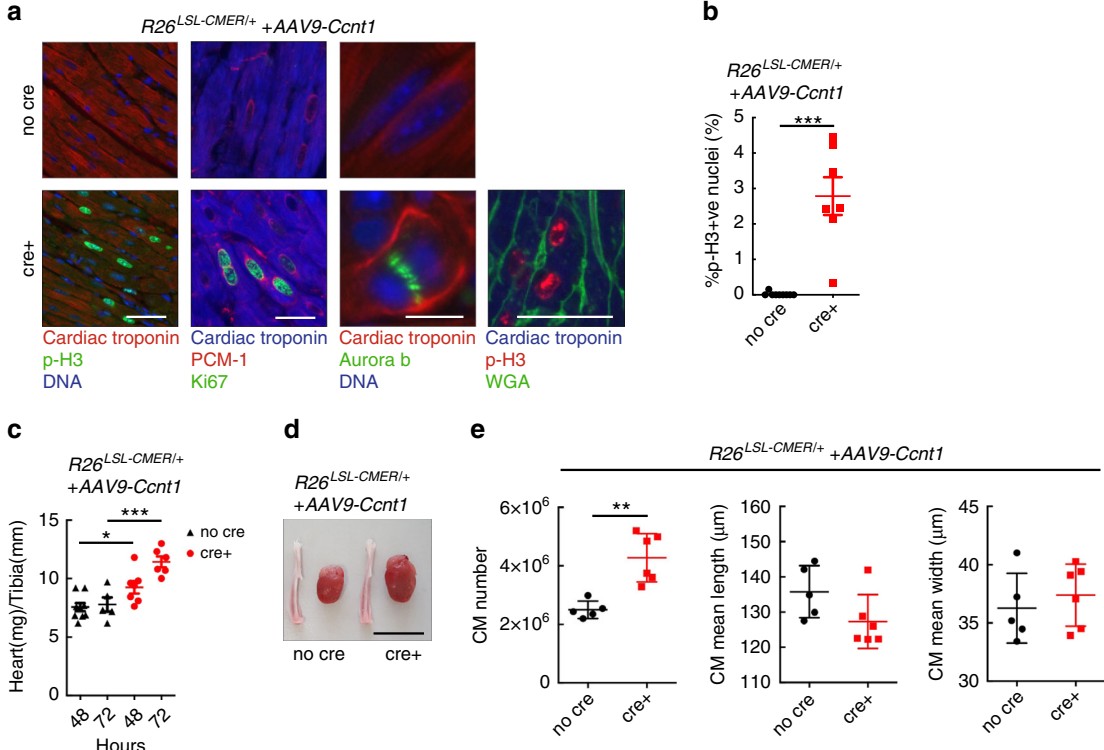

**Fig. 9 Overexpression of Cyclin T1 in the adult heart facilitates Myc-driven proliferation of adult cardiomyocytes. a** Immunofluorescent staining of cardiac troponin, p-H3,Ki67, PCM1, wheat germ agglutinin (WGA) and aurora B in control (*R26*^LSL-CMER/+; no cre) and *Myh6-Cre*; *R26*^LSL-CMER/+ (*R26*^LSL-CMER/+; cre+) adult mouse heart isolated 4 weeks post systemic infection with an adeno-associated virus encoding *Ccnt1* (*AAV9-Ccnt1*) and 48 h post administration of tamoxifen. Representative images based on analysis of five independent mice. Scale bars represents 10 μm (aurora B) and 50 μm (all others). **b** Quantification of p-H3-positive nuclei percentage in control (*R26*^LSL-CMER/+; no cre, *n* = 9) and *Myh6-Cre*; *R26*^LSL-CMER/+ (*R26*^LSL-CMER/+; cre+, *n* = 7) adult mouse heart isolated 4 weeks post systemic infection with an adeno-associated virus encoding *Ccnt1* (*AAV9-Ccnt1*) and 48 h post administration of tamoxifen. Means are taken from five images per mouse; Mean and s.e.m shown. Unpaired *t*-test; no cre vs cre+ *P* < 0.0001. **c** The weight (mg) of hearts isolated from control (*R26*^LSL-CMER/+; no cre, *n* = 9 48 h, *n* = 5 72 h) and *Myh6-Cre*; *R26*^LSL-CMER/+ (*R26*^LSL-CMER/+; cre+, *n* = 7 48 h, *n* = 6 72 h) adult mouse heart 4 weeks post systemic infection with an adeno-associated virus encoding *Ccnt1* (*AAV9-Ccnt1*) and 48 and 72 h post administration of tamoxifen, expressed as fold change over the length (mm) of a tibia isolated from the same mouse. Mean and s.e.m shown. One-way ANOVA with Tukey's multiple comparisons test; no cre vs cre+: **P* = 0.05 (48 h), ****P* = 0.001 (72 h). Replicate samples are derived from independent mice. **d** Image of the whole heart and a tibia from control (*R26*^LSL-CMER/+; no cre) and *Myh6-Cre*; *R26*^LSL-CMER/+ (*R26*^LSL-CMER/+; cre+) adult mouse heart isolated 4 weeks post systemic infection with an adeno-associated virus encoding *Ccnt1* (*AAV9-Ccnt1*) and 72 h post administration of tamoxifen. Scale bar represents 10 mm. Representative images based on analysis of 11 independent mice. **e** Quantification of the number and size of cardiomyocytes from control (*R26*^LSL-CMER/+; no cre, *n* = 5) and *Myh6-Cre*; *R26*^LSL-CMER/+ (*R26*^LSL-CMER/+; cre+, *n* = 6) adult mouse heart isolated 4 weeks post systemic infection with an adeno-associated virus encoding *Ccnt1* (*AAV9-Ccnt1*) and 72 h post administration of tamoxifen. Mean and s.d shown. One-way Mann–Whitney test; no cre vs cre+ ***P* = 0.0043. Replicate samples are derived from independent mice. Source data are provided as a Source Data file.

together with 100 nM (Z)-4-OHT (Sigma, H7904) in ethanol and collected after 4 h for RNA extraction.

**Adenovirus-mediated infections**. Ad5CMVeGFP (VVC-U of Iowa-1174) was obtained from the University of Iowa, Viral Vector Core and supplied at $3 \times 10^1$ pfu/ml. For MEFs, 0.5 μl of virus was used per $5 \times 10^5$ cells. For CMs, 0.25 μl of virus was used per $2 \times 10^4$ cells. Ad-RFP-mCCNT1 (ADV-223024) was obtained from Vector Biolabs and supplied at $5.6 \times 10^{10}$ pfu/ml. For MEFs, 20 μl of virus was used per $5 \times 10^5$ cells. For CMs, 10 μl of virus was used per $2 \times 10^4$ cells. AAV9-cTnT-3xFLAG-mCCNT1-WPRE and AAV-cTnT-RFP-WPRE (cat. no: VB5443) were obtained from Vector Biolabs and supplied at $2.6 \times 10^{13}$ GC/ml and $6.0 \times 10^{13}$ GC/ml, respectively. For systemic infection, AAV viruses were diluted in PBS, each mouse received $1 \times 10^{11}$ GC in 100 μl via tail vein injection at 4–7 weeks of age.

**Quantitative reverse transcription PCR**. Total RNA was isolated using TRIzol Reagent (Thermo Fisher, 15596-018) following manufacturers instructions. cDNA was synthesised from 1 μg of RNA using the High-Capacity cDNA Reverse Transcription Kit with RNase Inhibitor (Thermo Fisher, 4374966) following manufacturers instructions. Quantitative reverse transcription PCR (qRT-PCR) reactions were performed on a Applied Biosystems QuantStudio 5 Real-Time PCR System using Fast SYBR Green Master Mix (Thermo Fisher, 4385612), following

manufacturers instructions. Primer sequences in Supplementary Table 1. Expression is normalised to *Gapdh* and *Actin*.

**Flow cytometry analysis of DNA content**. For MEFs: Cells were trypsinised, washed in PBS and fixed in 70% ethanol overnight. Fixed cells were rehydrated, washed in PBS, treated with RNase A (10 μg) and PI (5 μg), and analysed by Flow cytometry using a BD Accuri C6 flow cytometer. For each sample, at least 10,000 events were recorded and analysed using FlowJo software (Tree Star).

**Fix-dissociation of cardiomyocytes**. Whole hearts were isolated from mice and washed in PBS followed by fixation in 1% PFA overnight at room temperature with agitation. The following day hearts were washed four times in PBS. Hearts were cut into 1–2 mm³ pieces and incubated with 0.5 U/mL collagenase B (Roche #11088807001) in 0.2% NaN3/PBS and left to oscillate at 1000 rpm at 37 °C. Every 12 h, the cardiomyocyte supernatant was collected and stored at 4 °C in 0.2% NaN3/FBS. Once dissociation was complete (~8 days) cells were centrifuged at $1000 \times g$ for 3 min, washed twice in PBS and stored in 0.2% NaN3/PBS at 4 °C. All of the following incubations were carried out at room temperature. A total of 500,000 cardiomyocytes were incubated with blocking buffer (4% BSA, +0.2% Triton X-100, +1 mM EDTA and +0.02% sodium azide) and agitated (1000 rpm) at room temperature for 3 min. Cells were then re-suspend in primary antibodies (Cardiac Troponin T (13-11; Thermo Fisher, MA5-12969; 1:100, p-H3 (Millipore,

1:400))) and incubated with agitation for 1 h. After two washes in blocking buffer, secondary antibodies (Alexa Fluor 488 Goat Anti-mouse IgG (H+L; Life Technologies, A11008)) were incubated together with Hoechst and agitated for 1 h at room temperature. Following two washes in blocking buffer, cells were re-suspend in blocking buffer, and either spun onto slides and mounted for imaging or analysed by flow cytometry. Cardiomyocyte length and width was measured using ImageJ software on images over-exposed for Cardiac Troponin T. A mean cell size for each heart was determined from measurements of >13 individual cells. Flow cytometry was performed on a BD LSRFortessa and analysed using FlowJo software (Tree Star).

**Immunoblotting.** Snap frozen animal tissues were ground on liquid nitrogen. Isolated cells were directly lysed on cell culture dishes and proteins extracted in buffer containing 1% SDS, 50 mM Tris pH 6.8 and 10% glycerol on ice for 10 min. Lysates were boiled for 10 min, followed by sonication (Bioruptor, Diagenode) for 15 min on ice. Total protein (50 µg) was electrophoresed on an SDS–PAGE gel and transferred onto immobilon-P (Millipore) membranes. These were then blocked in 5% non-fat milk and incubated with primary antibodies overnight at 4 °C. Secondary antibodies were applied for 1 h followed by chemiluminescent visualisation (Thermo Scientific, 32106 or Millipore, WBKLS0500). Immunoblots were developed on Fuji RX X-ray film, scanned on an Epson Perfection V500 Photo flatbed scanner. Images were cropped using Adobe Photoshop but otherwise unprocessed. Primary antibodies: GAPDH (D16H11) XP® (Cell Signaling Technology, 5174, used at 1:5000), β-actin (AC-15; Santa-Cruz Biotechnology, sc-69879, used at 1:5000), α-tubulin antibody (DM1A; Santa-Cruz Biotechnology, sc-32293, used at 1:5000), Myc (Y69; Abcam, ab32072, used at 1:10,000), Rpb1 CTD (4H8; Cell Signaling Technology, 2629, used at 1:2500), phospho-Rpb1 CTD (Ser2; E1Z3G; Cell Signaling Technology, 13499, used at 1:2500), phospho-Rpb1 CTD (Ser5; D9N5I; Cell Signaling Technology, 13523, used at 1:2500), CDK9 (C12F7; Cell Signaling Technology, 2316, used at 1:1000), Cyclin T1 (D1B6G; Cell Signaling Technology, 81464, used at 1:1000), LARP7 (Abcam, ab134746, used at 1:1000), anti-rabbit IgG HRP (Sigma, A0545, used at 1:10,000) and anti-mouse IgG HRP (Sigma, A0944, used at 1:10,000). Sample loading was normalised for equal protein content. Expression of GAPDH or α-tubulin is included as a confirmation of efficiency of protein isolation and comparable loading between individual tissue samples. Where necessary a composite figure was generated from the same samples loaded across multiple blots and a representative image of GAPDH is shown, Fig. 5a and Supplementary Fig. 5a.

**Immunohistochemistry.** Immunohistochemistry was performed on 4 µm sections. Sections were de-paraffinized and rehydrated. Antigens were retrieved by boiling in 10 mM citrate buffer (pH 6.0) for 10 min. Endogenous peroxidase activity was blocked with 0.3% hydrogen peroxide for 30 min. Sections were then treated with rabbit VECTASTAIN Elite ABC horseradish peroxidase kit (Vector Laboratories, PK-6101) following the manufacturers protocols with all incubations were carried out at room temperature. Briefly, sections were blocked in normal goat serum for 20 min, followed by 1 h incubation in primary antibody. Sections were then washed three times in PBST, and incubated for 30 min in secondary antibody, sections were then washed again in PBST and then incubated in ABC complex. Sections were developed in DAB (3,3′-diaminobenzidine) for 5 min, counterstained in haematoxylin, dehydrated and mounted in DPX. Staining was imaged on a Zeiss Axio Imager using the Zeiss AxioVision 4.8 software using the AutoLive setting and interactive white balance. Quantification was performed by counting number of cells per field of view for five images per organ/mouse, the mean of five raw counts was calculated and is represented by each data point per graph.

Primary antibodies; anti-Ki67 (SP6; Thermo Scientific, RM-9106; 1:200), anti-phospho-Histone H3 (Ser10; Merck Millipore, 06-570; 1:1250) and anti-Cyclin T1 (AbCam, ab238940; 1:500).

**Immunoflorescence.** Immunoflorescence was performed on 4 µm sections. Sections were de-paraffinized and rehydrated. Antigen retrieval was performed by boiling in 10 mM citrate buffer (pH 6.0) for 10 min. For BrdU staining, DNA was denatured with 2N HCL, 0.2% triton X for 10 mins at 37 °C. Sections were blocked in 2.5% goat serum (from number), 1% BSA in PBST for 20 mins at room temperature. Primary antibody, made up in blocking buffer, was added for 1 h at room temperature, followed by PBST washes and secondary antibodies added for 1 h at room temperature. Nuclei were visualised using Hoechst (Sigma, 861405) and sections mounted in prolong gold (Thermo Fisher, P36930). Antibodies: anti-phospho-Histone H3 (Ser10; Merck Millipore, 06-570: Antibody; 1:500), cardiac troponin T (13-11; Thermo Fisher, MA5-12969; 1:100), anti-BrdU (BU1/75 (ICR1)) antibody (Abcam, ab6326, 1:200), anti-aurora B antibody (Abcam, ab2254; 1:200), anti-PCM1 (SigmaAldrich, HPA023370, 1:100), anti-Ki67 (SolA15; Thermo Fisher, 1:100), Alexa Fluor 555 goat anti-rabbit IgG (H+L; Life Technologies, A21428), Alexa Fluor 555 goat anti-mouse IgG (H+L; Life Technologies, A21422), Alexa Fluor 555 goat anti-rat IgG (H+L; Life Technologies, A21434), Alexa Fluor 488 goat anti-rabbit IgG (H+L; Life Technologies, A11008), Alexa Fluor 488 goat anti-rat IgG (H+L; Life Technologies, A11006), Alexa Fluor 350 goat anti-mouse (H+L; Life Technologies A11045), wheat germ agglutinin, Alexa Fluor™ 488 conjugate (Thermo Fisher, W11261).

**Southern blotting.** DNA (≥10 µg) was digested overnight with appropriate restriction enzymes and run on a 0.8% agarose gel to separate fragments. Gels were subject to depurination and neutralisation before transferring DNA to Hybond-XL membrane (RPN203S, GE Healthcare Amersham) overnight by capillary action. The membrane was then washed and dried at 80 °C for 2 h. αP32–dCTP probes were labelled using Prime-It® II Random Primer Labelling Kit (300385, Agilent) and purified with Illustra ProbeQuant™ G-50 Micro columns (28-9034-08, GE Healthcare Amersham). Following pre-hybe for 1–2 h in Rapid-hyb™ Buffer (RPN1636, GE Healthcare Amersham), the membrane was hybridised with the probe for at least 5 h, washed, exposed to Fuji RX X-ray film and the film scanned on an Epson Perfection V500 Photo flatbed scanner.

**Transmission electron microscopy.** Day 15 $R26^{CMER/+}$ mouse heart was cut into 1 mm² pieces and fixed overnight in fresh 2.5% glutaraldehyde/PBS solution (Sigma, G7651), followed by three washes in PBS. Samples were then placed in fresh fix and irradiated for 6 min at 80 W under vacuum in a Pelco Biowave (Ted Pella In, Redding, CA). Fixed tissue was then stained with 3% potassium ferricyanide and 2% osmium tetroxide in 0.1 M cacodylate buffer for 30 min, then washed three times in distilled water and incubated in 1% thiocarbohydrazide for 30 min at room temperature. Samples were then washed three times in distilled water and immersed in 2% osmium tetroxide for 30 min at room temperature, then again washed three times in distilled water. Samples were contrasted with 1% aqueous uranyl acetate for 30 min at 4 °C and washed three times in distilled water. A solution of 0.06% lead nitrate was prepared by dissolving in aspartic acid (pH 5.5) at 60 °C, which was then filtered and added to samples for 20 min at 60 °C before washing three times with distilled water at room temperature. Pieces of tissue underwent serial dehydration by immersing in each increasing ethanol concentration twice (30%, 50%, 70%, 90% and 100%) and irradiating for 40 s at 250 W in a Pelco microwave. Samples were then infiltrated with increasing concentrations of epon LX112 resin (25%, 50% and 75%) in ethanol for 3 min at 250 W under vacuum in a Pelco microwave, then twice in 100% resin before polymerising at 60 °C overnight[79]. Samples were analysed on a Apreo FESEM equipped with a VolumeScope serial block face system and MAPS software (Thermo Fisher) with a slice thickness of 100 nm. Datasets were aligned, segmented and visualised in Imod (Bio3D) and Chimera (UCSF) software.

3View EM stacks were assembled and 3D modelling of structures was performed in IMOD 3D reconstruction and visualisation software (University of Colorado, Boulder).

**ChIP sequencing.** Mouse tissue was harvested from mice, washed in PBS and cut into small 2–3 mm pieces with a scalpel blade. Tissue was then fixed in 1% formaldehyde/PBS for 15 min, washed in PBS three times, snap frozen and stored at −80 °C until further processing. Upon thawing, tissue was incubated in 125 mM glycine (5 min, room temperature), washed in PBS and nuclei prepared on ice using a dounce homogeniser. All following steps were performed on ice. Nuclei were washed in PBS, resuspended in ChIP buffer (4 ml of buffer per 0.5 ml of tissue; 100 mM NaCl, 133 mM Tris pH 8.0, 5 mM EDTA, 0.02% NaN₃, 0.67% SDS and 1.67% Triton X-100) supplemented with protease inhibitors (Complete Protease Inhibitor Cocktail Tablets, Roche) and phosphatase inhibitors (50 mM NaF, 1 mM β-GP and 1 mM NaV). Chromatin was sonicated to an average length of 300 bp, pre-cleared twice with protein G-sepharose beads, pre-blocked with 0.5% *Escherichia. coli* tRNA (Sigma) and 0.5% BSA (ref. [80]). In this study, the following antibodies were used: Myc N262 (Santa Cruz, sc-764×), RNA PolII N20 (Santa Cruz, sc-899×). A total of 10 µg of antibodies were incubated overnight with 500 µl of chromatin (+500 µl of ChIP buffer to each a volume of 1 ml per reaction). Blocked protein G-sepharose beads were added, incubated for 2–3 h and washed as follows: three times with mixed micelle buffer (150 mM NaCl, 20 mM Tris pH 8.1, 5 mM EDTA, pH 8.0, 5.2% sucrose, 0.02% NaN₃, 1% Triton X-100 and 0.02% SDS), 2x with buffer 500 (50 mM HEPES pH 7.5, 0.1% sodium deoxycholate, 1 mM EDTA, 500 mM NaCl, 1% Triton X-100 and 0.02% NaN₃), two times with LiCl/detergent buffer (10 mM Tris pH 8.0, 0.5% sodium deoxycholate, 1 mM EDTA, 250 mM LiCl, 0.5% NP-40 and 0.02% NaN₃) and once with TE. DNA was decrosslinked overnight (2% SDS in TE, 65 °C), purified using QiaQuick columns (Qiagen) and quantified using Qbit (Thermo Fisher). A total of 2–5 ng of ChIP DNA were used for ChIP-seq library preparation, using the TruSeq ChIP Library Preparation Kit (Illumina), following manufacturers instructions.

**RNA sequencing.** Total RNA from tissues was purified using TRIzol (Invitrogen). Briefly, tissue was homogenised in Trizol (Invitrogen) and chloroform was added. After centrifugation, RNA was purified using RNeasy Mini columns (Qiagen) according to the manufacturer's protocol, including the on column DNaseI digestion step. Alternatively, total RNA was purified using PureLink RNA Mini Kit (Thermo Scientific) together with on column DNA digestion using PureLink DNase (Thermo Scientific). Then 1–5 µg of purified RNA was treated with Ribozero rRNA removal kit (Illumina) and ethanol precipitated. RNA quality and removal of rRNA were checked with the Agilent 2100 Bioanalyser (Agilent Technologies). Libraries for RNA-seq were then prepared with the TruSeq RNA Sample Prep Kits v2 (Illumina) or NEBNext® Ultra™ II DNA Library Prep Kit for

Illumina (NEB), following manufacturers instructions starting from the RNA fragmentation step.

**RNA-seq sample processing and analysis**. Adult tissues: Sequence reads were aligned to the mm9 mouse reference genome using TopHat aligner (version 2.0.8). In case of duplicated reads, only one read was kept. FeatureCounts software was used to associate read counts to genes (http://bioinf.wehi.edu.au/featureCounts/). Exon reads per kilobase per million mapped reads (eRPKM) defined gene expression using total library size as the number of reads aligned on exons. We applied a filter for a minimum expression level average eRPKM > 3.5 and Log2FC > 0.5, DEGs were identified using the Bioconductor package DESeq2 (ref. [81]).

AAV infected and day 15 heart + mouse samples: Sequence reads were adaptor trimmed using Cutadapt[82] (version 1.11) and aligned using BWA aln for short reads[83] (version 2.5.2a) to the GRCm38 (mm10) assembly with the gene, transcript and exon features of the Ensembl (release 70) gene model. Expression was estimated using RSEM[84] (version 1.2.30). Transcripts with zero read counts across all samples were removed prior analysis. Normalisation of RSEM expected read counts was performed by dividing by million reads mapped to generate counts per million, followed by the trimmed mean of M-values method from the edgeR package[85,86]. Differential expression analysis was performed using edgeR.

**ChIP-seq data analysis**. Data were generated and processed by aligning sequence reads to the mm9 genome with the BWA aligner using default settings. Peaks were called using the MACS software (v2.0.9)[87]. Normalised read counts within a genomic region were determined as the number of reads per million of library reads (total number of aligned reads in the sequencing library). Peak enrichment was determined as log2 (Peakw/Nc−inputw/Ni), where Peakw is the read count on the enriched region and inputw the read count on the same region in the corresponding input sample, Nc is the total number of aligned reads in the ChIP sample and Ni is the total number of aligned reads in the input sample. Promoter peaks were defined as all peaks with at least one base pair overlapping with the interval between −2 kb to +1 kb from the nearest TSS. Myc-binding peaks called in the heart but not in the liver were annotated as heart specific, and peaks called in the liver but not in the heart were annotated as liver specific. Myc-binding peaks identified in both tissues, heart and liver, were annotated as common Myc binding sites.

For the analysis of chromatin accessibility, regulatory elements and transcriptional activity, genomic regions of promoter Myc binding sites in liver and heart were extracted 6 kb around the peak centre. Read coverage of the 6 kb Myc-bound genomic regions, were extracted from the alignment bam files using the Genomation R package[88]. To overlap Myc peaks with chromatin accessibility and active promoter histone marks, we extracted publicly available data from 8 week adult C57BL/6 liver and heart tissue from ENCODE (DNaseI: http://genome.ucsc.edu/cgi-bin/hgFileUi?db=mm9&g=wgEncodeUwDnase, ChIP: http://genome.ucsc.edu/cgi-bin/hgFileUi?db=mm9&g=wgEncodeLicrHistone, Download: 01/08/2018). Specifically, chromatin accessibility was ascertained using a DNaseI hypersensitivity assay, and active promoters were identified based on H3K27ac and H3K4me3 ChIP sequencing. Read coverage of the 6 kb Myc-bound genomic regions were extracted using the genomation R package from bigwig files generated by ENCODE. For each ChIP (Myc, PolII, H3K27ac, H3K4Me1 and H3K4me3) and DNaseI experiment, read coverage of the 6 kb regions were binned into 100 equally sized bins with the maximum counts used for smoothing windows. Each experiment's smoothed read coverage was visualised in a heatmap with independent colour scales to account for differences in experimental protocols, processing and tissue. Genomic regions were clustered based on whether Myc binding was specific for heart, liver or common to both tissues.

**Statistical analysis**. Pipelines for primary analysis (filtering and alignment to the reference genome of the raw reads) and secondary analysis (expression quantification, differential gene expression and peak calling) have been integrated in the HTS-flow system[89]. Bioinformatic and statistical analysis were performed using R with Bioconductor packages and comEpiTools packages[90,91]. Motif analysis was performed using CentriMo software[92]. Intron analysis was performed with the INSPEcT tool[93,94]. Gene lists from RNA and ChIP sequencing were analysed in Enrichr[95]. Statistical analyses for IHC and qRT-PCR were performed using GraphPad Prism v6.0d (GraphPad Software, Inc., San Diego, CA, USA) as indicated with $P \leq 0.05$ considered to be statistically significant. Venn diagrams were draw in the interactive tool Venny [https://bioinfogp.cnb.csic.es/tools/venny/index.html].

**Reporting summary**. Further information on research design is available in the Nature Research Reporting Summary linked to this article.

## Data availability

All datasets generated in this study, including Myc ChIP sequencing, RNA PolII ChIP sequencing and RNA sequencing, have been deposited in ArrayExpress (www.ebi.ac.uk/arrayexpress) under accession codes; E-MTAB-7592, E-MTAB-7593, E-MTAB-7595, E-MTAB-8462, E-MTAB-8515 and E-MTAB-7636. The source data underlying Figs. 1a, c

d, 4a, 5a–e, 6a, b, d, 7c, d, 8a–c, 9b, c, e and, Supplementary Figs. 1b–d, f, h, 2a, 4a, b, 5a–f, 6a, b, d, 7c, 8a and 9a, d, e are provided as a Source Data file. Further information and requests for resources and reagents should be directed to, and will be fulfilled by, Gerard Evan (gie20@cam.ac.uk). Requests regarding AZ5576 should be directed to AstraZeneca PLC.

## Code availability

No previously unreported custom computer code or algorithm used to generate results.

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

## Acknowledgements

The authors thank the support staff in the Cambridge University Biomedical Services at the Gurdon Institute and CRUK Cambridge Institute. We would like to thank AstraZeneca and, specifically, Lisa Drew and Justin Cidado for AZ5576 supply and guidance. We would like to thank Steven Lane for laboratory support and critical advice on the manuscript. The authors acknowledge the use of the Microscopy Australia Research Facility at the Center for Microscopy and Microanalysis at The University of Queensland. We would like to thank Charles Ferguson, Robyn Webb and Marcel Sayre (CMM, UQ), for assistance with electron microscopy. We also thank Roger Foo and Peter Li from the National University Heart Centre, Singapore for their technical advice on the isolation and infection of adult cardiomyocytes. This work was primarily supported by CRUK (Programme Grant A19013 to GIE). M.J.B. was funded by an EMBO long-term fellowship and an Australian NHMRC Early Career Fellowship (APP1072477). This work was supported by funding from the European Research Council (grant agreement no. 268671-MYCNEXT), the Italian Health Ministry (RF-2011- 02346976) and the Italian Association for Cancer Research (AIRC, IG 2015-16768 and IG 2018-21594) to B.A., and from Worldwide Cancer Research (15-1260) to A.S. This work was also supported by the National Health and Medical Research Council of Australia (grants APP1140064 and APP1150083 and fellowship APP1156489 to R.G.P.). R.G.P. is supported by the Australian Research Council (ARC) Centre of Excellence in Convergent Bio-Nano Science and Technology.

## Author contributions

C.H.W., M.J.B., A.S., D.L.B., G.I.E. and B.A. conceptualised the work. Experiments were performed by C.H.W., M.J.B., A.S. and D.L.B. C.H.W. and M.J.B. wrote the manuscript. A.S., V.P., T.R.K., B.A. and J.S. performed and analysed the global sequencing analysis. J.E.H., E.R.P. and G.A.Q. performed the cardiomyocyte counting and analysed global sequencing analysis. J.R. and R.G.P. performed the TEM. G.I.E., B.A. and T.D.L. edited the manuscript.

## Competing interests

All authors declare no competing interests, except G.I.E. is a member of AstraZeneca's Oncology External Advisory Panel. M.J.B. and C.H.W. are named inventors on a provisional patent relating to data in this manuscript.
