## [Peer Review File · Nature Communications]

Reviewers' comments:

Reviewer #1 (Remarks to the Author):

The authors report the interesting observation that MYC binds to its target sites in open promoters even in adult heart, a tissue, in which it does not stimulate transcription, proliferation and organ growth. They go on to show that components of the basal machinery are expressed at very low levels in adult heart; ectopic of one of these components, cyclin T1, restores a moderate level of transcriptional regulation, proliferation and growth. These are interesting observations. At the current level of analysis, however, several major issues remain unresolved. The three major issues are listed first.

Comments

1. The authors look at relative expression of specific genes as a readout of MYC function. An alternate model of MYC function suggests that MYC acts as a global inducer of transcription. The clearest data supporting this "amplifier" model is the effect of MYC induction in primary B-cells, an experimental system, which is similar to the systems used here. In the amplifier model of MYC, relative changes in gene expression are not primarily relevant. While the authors may disagree with this model, it is unclear to this reviewer why they do not take an agnostic standpoint and record global changes in transcription activity +/- MYC across the critical tissues. In my view, determining the effects on overall RNA polymerase function in vivo (e.g. using some form of metabolic labeling) is essential, since the authors may simply miss that all genes are induced. This is very relevant, since the analysis shown in Figure 3, which suggests that there is some form of discrepancy between widespread MYC binding and selective regulation, may be completely misleading (all genes may go up). This is also relevant since the relative effects in Figure 4e are really marginal and it is somewhat hard to believe that these small changes cause a significant change in cell growth.

2. Figures 4, 5 and 6 compare mice with and without Cre (which excises the LSL cassette), but lack the obvious control without OHT. Without this control, the effects seen cannot be unequivocally attributed to the MYCER chimera. This control needs to be added to all panels in these figures. This is particularly relevant since the authors in the discussion spent a long time discussing that this is one of the first studies looking at immediate effects of MYC activation (which is a bit strange since there are dozens of papers on early responses, some of them from the authors own laboratory) and discussing the merits of the MYCER system, which this year is exactly 30 years old (Eilers, M. Nature, 1989. 340(6228): p. 66-8).

3. Figure 6a shows clearly that adult heart expresses very low levels of CDK9 and of RNAPII. Consistent with this, ectopic expression of cyclin T1 has no effect on the phosphorylation of Ser2 of the CTD, the substrate of CDK9; this is shown in Figure 4d, a panel that is shown but not discussed in the text. The finding that cyclin T1 expression restores gene expression is therefore surprising and shows that cyclin T1 must have another CDK partner. The authors need to identify the CDK partner for cyclin T1 in the restored hearts. Could it be that it switches to other CDK's, as seen in some CDK knockout animals? This is also relevant since the authors claim to "restore neonatal levels of P-TEFb activity", but do not show this.

4. Independent of this point, the statement "Myc transcriptional output is determined by P-TEFb" is not supported by any data in this paper.

5. The control staining for cyclin T1 in Figure 6a is not ok. The authors need to obtain more convincing

data that cyclin T1 is indeed expressed, which percentage of cells expresses cyclin T1 and what the relative levels of expression relative to proliferative tissue is. Quantifications need to be added for Figure 6c, this is again hard to see. In Figure 6e, the label "per field" needs to be replaced by "% of cells".

6. Similar to point 1: global transcription rates need to be shown for Figure 6g. It is very hard to imagine that this growth is not accompanied by an induction of general transcription.

7. The authors alter the genes that they use from one panel to the next without explanation. A consistent set of genes should be analyzed in all panels.

8. On a different note: have the authors considered the question of why endogenous MYC is expressed and chromatin bound in a non-proliferative tissue? I understand that it can be re-awakened, but what does it do without cyclin T1? Some insight into this would greatly strengthen the paper.

Reviewer #2 (Remarks to the Author):

Bywater et al. present a compelling account of the role of Myc-driven recruitment of P-TEFb and subsequent activation of RNAPII in the proliferative capacity of different tissues in mice. The authors created a series of mice with a range of expression levels of MycERT2, and using 4-OHT induced systemic Myc activity, promote a variety of tissue-dependent proliferative responses. Some tissues, such as the liver, showed a proliferative response, while others, such as the heart, appeared unresponsive to Myc activation. These differences are not due to differential binding of Myc to chromatin in these tissues, but appear to be a result of differences in the downstream capacity of Myc to drive transcriptional changes in different tissues. The authors attribute these changes to differences in the variations of P-TEFb (cyclin T1/Cdk9) levels, and subsequent RNAPII phosphorylation. Using a P-TEFb inhibitor, the authors demonstrated that this complex is required for Myc-driven proliferation of the liver, while overexpression of Cyclin T1 in the heart appears sufficient to allow this tissue to respond proliferatively to MYC.

This is an interesting manuscript that begins with an elegant and extremely informative system to understand how tissues respond to Myc activation. In general, the experiments are well conducted and controlled, and interpreted responsibly. The results themselves are extremely interesting, and will be of wide interest to those studying Myc and in developing ways to regenerate heart tissue.

That said, the story does start to unravel as the mechanism of differential response to Myc is investigated. The experiments with Cyclin T1, are provocative, but don't go far enough and are not adequately pursued to support the kind of mechanism the authors propose. The effects of Cyclin T1 expression in primary cardiomyocytes on p-Rpb1(S2) is modest, to say the least. In mice infected with adenovirus for overexpression of Cyclin T1 in cardiomyocytes, only immunohistochemistry of Cyclin T1 was used to confirm this overexpression; there is no validation that this is altering the phosphorylation status of Rpb1, which seems a major oversight. The authors have not looked to see whether CDK9 and Rpb1, which both apparently have low levels in the heart, are induced by overexpression of Myc and/or Cyclin T1 (which could explain the response). And most problematic for the proposed mechanism is the timing of the mechanistic experiments, which take place over relatively long time frames (days), making it difficult to know if the effects are direct or indirect.

Reviewer #3 (Remarks to the Author):

The manuscript by Bywater et al explores the physiological response of different tissues to acutely increased Myc activity. They focus on Myc's capacity to elicit mitosis as a function of Myc level. They use a series of tamoxifen-activatable MycERs knocked into the Rosa locus to enable controlled increases in Myc activity and they look at liver, lung, heart and kidney. First, they find that Myc binds at all open chromatin sites in each tissue, at both common genes that are commonly expressed as well as at genes that have tissue restricted expression. MycER activation elicited three response patterns in adult tissues: in already proliferating tissues such as thymus and spleen, proliferation persisted, but was not augmented. Resting tissues with mitogenic potential were driven to proliferate by increased Myc. Tissues lacking proliferative/regenerative potential were not induced to proliferate by activation of MycER. In general, the magnitude of the transcriptional response was attenuated in non-proliferative tissues. The authors next explore the hypothesis that Myc action is limited in non-proliferative tissues by a generalized insufficiency of transcription. Examination of extracts from the different tissues reveals that P-TEFb is reduced in non-proliferative tissues but increasing P-TEFb activity by increasing cyclin T levels confers Myc-induced proliferation on these tissues. So adult heart with increased P-TEFb becomes Myc responsive. Young heart that retains the ability to proliferate has higher P-TEFb levels and is Myc responsive.

This work is creative, well-executed, conceptually important and replete with therapeutic implications. It provides a rationale for many peculiar phenomena associated with Myc. Nevertheless, there are few points that should be addressed.

1. The authors state that CDK9 phosphorylation of the RNAPII CTD is what drives the transition from pausing to productive elongation. Actually, it is still debatable which CDK is the principle kinase of Ser2. A case can also be made for CDKs 12 and 13. But probably a more important issue is that the authors incorrectly seem to imply that Ser2 phosphorylation is the key event in pause release. Actually, probably more important substrates for CDK9 for pausing and pause release than Ser-2 are DSIF (spt4/5) and NELF. It would be nice to comment on (or better yet show) the amounts and the phosphorylation status of these complexes in the different tissues.
2. In figure 4, total Rpb1 levels also seem to follow the proliferative potential of the tissues. The authors should comment on whether or not other transcription machinery components besides P-TEFb might also contribute to the proliferation block.
3. The results in this manuscript indicate that the effects of Myc relate to quantitative changes in global transcription more than to qualitative changes in the transcription profile. This would seem to be in accord with the suggestion in refs. 25 and 26 that Myc is an amplifier. Considering that this has been a matter of some disagreement in the field, perhaps the authors could comment on this.

Reviewer #4 (Remarks to the Author):

In this highly interesting contribution Bywater et al describe that the responsiveness of cells to pro-proliferative signals provided by the oncogene Myc depends on the presence of P-TEFb, a transcriptional co-factor complex consisting of Cdk9 and cyclin T1. The authors found that cells which respond by proliferation to increased expression of Myc (i.e. liver cells) and those that don't (i.e.

cardiomyocytes), both bind Myc but show differential transcriptional responses. The authors demonstrate that proliferative responses to increased Myc activity in the liver depends on CDK9 activity. Activation of Myc in P15 cardiomyocytes, which unlike adult cardiomyocytes still contain high levels of P-TEFb, induced typical Myc-dependent transcriptional responses and robust proliferation. Importantly, adenovirus-mediated overexpression of cyclin T1 in adult hearts restored the ability of cardiomyocytes to respond by proliferation to increased Myc activity. Overexpression of both Myc and cyclin T1 in adult cardiomyocytes in vivo resulted in a remarkable increase in cardiomyocytes numbers and increased heart size without changes in cardiomyocyte volume.

A better understanding of the mechanisms that prevent proliferation of cardiomyocytes even under strong pro-proliferative conditions is direly needed, since e-enablement of cardiomyocyte proliferation in adult mammalian has been postulated to provide the basis for strategies to regenerate damaged hearts. The current study seems to provide at least some important clues, although direct translational approaches based on the overexpression of a highly potent oncogene are difficult, to say the least. The study contains several strong data sets. However, there are several important conceptual and technical problems that need to be addressed.

Specific comments

The authors describe that global activation of Myc (expression of a Myc-ER fusion in oocytod followed by a tamoxifen pulse) induced robust cell proliferation in liver, lung and pancreas but not in the heart or brain. I do not understand why the authors did not observe proliferative responses in heart TISSUES. The myocardium contains roughly four times more non-cardiomyocytes than cardiomyocytes, although cardiomyocytes provide the largest part to the volume/mass of the heart. It makes perfect sense that cardiomyocytes exposed to increased Myc activity remain cell cycle arrested but why is there no Ki67 staining and BrdU incorporation in heart and the brain tissues (Fig. 1b)? For example: endothelial cells are rather sensitive to increased Myc levels and should respond accordingly.

Related to the above topic: it might be helpful to determine expression (and nuclear localization after tamoxifen treatment) of MycER by immunostaining and not by western blot, which will miss absent Myc induction in specific cell types in different organs. Since the density of cells is different in different organs, ratios between total numbers of nuclei and p-H3, Ki67 positive nuclei should be given.

The authors found that CDK9 and Cyclin T1 levels are virtually absent (CDK9) or much lower in the heart compared to the liver (Cyclin T1). The P-TEFb complex consist of both CDK9 and Cyclin T1. Hence, it is difficult to understand why the overexpression of cyclin T1 compensates for the low/absent expression of both cyclin T1 and CDK9, in particular since inhibition of CDK9 in the liver prevented Myc induced proliferation. Does overexpression of cyclin T1 (with or without Myc activation, causes up-regulation of CDK9 in adult cardiomyocytes?

While cyclin T1 is still detectable in adult hearts, CDK9 expression is essentially absent, which makes it surprising that the authors used Cyclin T1 but not CDK9 overexpression to restore P-TEFb activity. Does overexpression of CDK9 compared to Cyclin T1 transduction restore proliferation of cardiomyocytes in Myc transgenic mice?

The authors use phosphorylation of the C-terminal end of Pol II (Rpb1) as a readout for P-TEFb and Pol II activity. However, Pol II is highly active (and phosphorylated) on genes required for cardiomyocyte function. The low phosphorylation of Pol II at ser 2 and ser5 in cardiomyocytes is therefore very hard to understand. Several other kinases (e.g. CDK8) although phosphorylate Rpb1. How can Pol II actively transcribe genes in cardiomyocytes without phosphorylation at ser 2 and ser5?

The authors propose that increased P-TEFb activity specifically enhances transcriptional elongation of Myc-primed promoters. There are techniques available (CoPro-Seq, PRO-Seq) to prove this hypothesis. Although such experiments would provide further mechanistic insights, they are not mandatory.

The study leaves several important questions unanswered. Of course, not of all them can be answered when first reporting a highly interesting observation. Nevertheless, one would like to learn more about the strong downregulation of CDK9 and cyclin T1 during cardiomyocyte maturation. What is responsible for the downregulation of CDK9 and cyclin T1 after P15? Furthermore: does the increased cardiomyocyte proliferation after Myc activation and cyclin T1 transduction restore heart regeneration in adults? Is cardiac function compromised due to increased Myc activity/cyclin T1 expression? Do the animals develop a tumor after extend Myc/cyclin T1 activation (or die in between because of heart failure)?

A comparison of the abundance of MycER DNA binding between heart and liver at promoter regions of mitotic cell cycle gene is missing. The authors should include a histogram of Myc DNA binding at such promoter regions.

There are several technical problems, which need to be addressed. A large part of the data relies on profiling of tissues rather of individual cell types, which is a severe shortcoming and might compromise the results. The authors need to use purified cardiomyocytes for all relevant experiments and not only for some. This is less of a problem when mice with cardiomyocyte-specific Myc expression are used but even after cardiomyocyte-specific expression of effectors, secondary changes in neighboring cells in the tissue might create confounding effects. Just as one of many examples: in Fig5c a heatmap of expression changes in different organs is shown. As in other experiments, the authors have to use isolated cardiomyocytes for the analysis.

The quantification of cycling and proliferating cardiomyocytes is not sufficient. Counting of p-H3 positive cardiomyocytes is notoriously difficult and yields unreliable data. The authors have to isolate cardiomyocytes first before counting BrdU or p-H3 positive cells to be sure that only cardiomyocytes are counted. An alternative approach is FACS-sorting of PCM1-positive nuclei after EdU injection. PCM1 specifically labels cardiomyocyte nuclei allowing specific and quantitative monitoring of cell cycle activity in cardiomyocytes.

AuroraB staining is a marker to identify mitosis of cardiomyocyte nuclei, which might result in multinucleation. The authors need to stain for anilin to visualize the contractile ring forming during cytokinesis. The morphometric data are highly suggestive of cytokinesis. Nevertheless, one would like to see an actual example of dividing cardiomyocytes. I fail to see the evidence based on the EM images, the movies or the staining in Fig. 5, 6. The authors should also determine the numbers of mono-, bi- and multinucleated cardiomyocytes.

If I got it correctly (was not described explicitly), the authors determined the absolute numbers of cardiomyocytes in different conditions by comparing yields of cardiomyocytes isolated from different hearts. The authors have to describe how exactly how the numbers of cardiomyocytes were determined. Comparison of yields from different isolation experiments is not a reliable technique. Proper morphometric measurements are necessary.

At page 11 line 8 the authors stated that there are modest, but non-significant, responses in the heart. The authors should avoid such statements. Either the FC changes are significant or not. "Tendencies" are hard to grasp in a scientific manner.

The manuscript contains numerous grammatical errors and odd phrases, which should be corrected.

Response to reviewers

We were delighted with the very positive comments from all reviewers, which have enabled us to further strengthen our manuscript. Please find below detailed responses to each of the reviewer's comments.

Reviewer #1

The authors report the interesting observation that MYC binds to its target sites in open promoters even in adult heart, a tissue, in which it does not stimulate transcription, proliferation and organ growth. They go on to show that components of the basal machinery are expressed at very low levels in adult heart; ectopic of one of these components, cyclin T1, restores a moderate level of transcriptional regulation, proliferation and growth. These are interesting observations. At the current level of analysis, however, several major issues remain unresolved. The three major issues are listed first.

Comments

1. The authors look at relative expression of specific genes as a readout of MYC function. An alternate model of MYC function suggests that MYC acts as a global inducer of transcription. The clearest data supporting this "amplifier" model is the effect of MYC induction in primary B-cells, an experimental system, which is similar to the systems used here. In the amplifier model of MYC, relative changes in gene expression are not primarily relevant. While the authors may disagree with this model, it is unclear to this reviewer why they do not take an agnostic standpoint and record global changes in transcription activity +/- MYC across the critical tissues. In my view, determining the effects on overall RNA polymerase function in vivo (e.g. using some form of metabolic labeling) is essential, since the authors may simply miss that all genes are induced. This is very relevant, since the analysis shown in Figure 3, which suggests that there is some form of discrepancy between widespread MYC binding and selective regulation, may be completely misleading (all genes may go up).

We thank the reviewer for this comment. We are of course very aware of the debate over whether Myc expression causes global transcriptional amplification versus specific changes in gene expression followed by RNA amplification^{1,2}. There is now a number of studies that advocate the view that Myc driven RNA amplification is better explained as a secondary consequence of the discrete gene expression programs set in place by MYC³⁻⁵. Nonetheless, we agree it is important to provide formal evidence in support of this latter hypothesis from our own datasets. We have now pursued 2 lines of investigation that support the notion that Myc has specific targets and that global transcriptional amplification is a secondary consequence.

First, if Myc acts as a general amplifier of transcription, this should already be evident at the earliest times. Accordingly, we would expect an increase in total cellular RNA content to be evident as early as 4 hours post Myc activation, the time point at which our transcriptional analysis was conducted. To determine RNA content changes, we have attempted to isolate an equal number of viable cells from tissues used for transcriptional analysis (liver, lung, kidney, heart). Unfortunately, the very different techniques needed to isolate component cells from different tissues often involve harsh and lengthy enzymatic digestion that results in substantial cell death and likely transcriptional perturbations. Our major concern is that RNA analysis on cells isolated under such conditions would not represent their *in vivo* state.

To overcome this limitation, we instead isolated Mouse embryo fibroblasts of the same genotype as that used in our pan-tissue analysis. After serum-free culture of MEFs for 3 days (to abrogate endogenous Myc expression), Myc^{ER}T² was acutely activated by the addition of 4OHT. Global RNA content only increases at 24hrs (see below, Supplementary Figure 3c), in line with similar published data sets³. Since our *in vivo* transcriptional analysis is performed at 4 hours after Myc^{ER}T² activation, well before observable RNA amplification, we conclude that the normalisation of our RNA sequencing data to be unaffected by (later) global RNA amplification and hence experimentally justified and appropriate.

Nonetheless, we agree that it would still be preferable to confirm this conclusion with data generated in the tissues being analysed. Given that metabolic labelling of nascent transcripts cannot be performed efficiently *in vivo*, due to either poor cellular uptake or low stability of the modified ribonucleotides *in vivo*, we have instead compared the relative abundance of intronic RNA as a surrogate read-out of the rate of transcription. A growing number of publications have provided compelling evidence that a comparison of the relative abundance of premature (unspliced) to mature (spliced) mRNA can be reliably used to determine the rate of nascent transcription of a given target⁵⁻⁷.

Accordingly, RNA sequencing data from liver, lung, heart and kidney after 4 hours of Myc activation was analysed with the INSPECT tool^{7,8} to determine the relative abundance of premature and mature mRNA for a given transcript and plotted these values for both the R26^{+/+} and R26^{CMER/+} genotypes. First, we compared genes that were determined to be commonly upregulated in response to Myc activation. In all tissues analysed, the R26^{CMER/+} slope was significantly higher than R26^{+/+} slope, indicating that, at 4 hours, Myc upregulated genes have a larger proportion of premature mRNA (Supplementary Figure 3d). Second, we compared expression of genes reported to be highly expressed in individual tissues (mouse gene atlas) - in this case, the R26^{CMER/+} and R26^{+/+} slopes were similar for each tissue analysed (Supplementary Figure 3e). These data indicate that, at 4 hours, Myc activation does not elicit global amplification of the pre-existing transcriptional repertoire in an

individual tissue. These new data have been added to the manuscript, along with the following explanation.

“Ectopic expression of Myc in other experimental systems has been proposed to result in the global amplification of the pre-existing transcriptional repertoire of a cell^{1,2}. However, an alternative view is that such amplification is a secondary consequence of the earlier engagement of a discrete Myc-specific transcriptional programme^{3,5,9}. In support of the latter, activation of ectopic Myc in primary, non-immortalised MEFs isolated from $R26^{CMER/+}$ mice resulted in no general increase in cellular RNA content at 4 hours (Supplementary Fig.3c). We also used our RNA sequencing data in combination with the INPEcT tool^{7,8} to determine the relative abundance of premature and mature mRNA for a given transcript as a surrogate read-out of the rate of transcription. When these values were plotted for genes that are commonly upregulated across multiple tissues in response to Myc, the $R26^{CMER/+}$ slope was significantly higher than that of the $R26^{c+/+}$ control, demonstrating that common Myc targets have a larger proportion of premature mRNA, indicative of an increased rate of transcription (Supplementary Fig.3d). In contrast, values for other genes that are highly expressed in individual tissues, were comparable, indicating that Myc does not elicit a global amplification of the pre-existing transcriptional repertoire of an individual tissue, at least at 4 hours after activation (Supplementary Fig.3e).”

This is also relevant since the relative effects in Figure 4e are really marginal and it is somewhat hard to believe that these small changes cause a significant change in cell growth.

We have now included RNA sequencing analysis of Myc activation in the presence of ectopic Cyclin T1 expression in the adult heart (Supplemental Figure 6b, Data S4 and raw data deposited into ArrayExpress accession E-MTAB-8462) and in purified adult cardiomyocytes (Figure 6d, Data S5, Supplementary Figure 6c and raw data deposited into ArrayExpress accession E-MTAB-8515). An increase in the rate of transcription of common Myc targets in cardiomyocytes purified from Cyclin T1-infected $Myh6Cre; R26^{LSL-CMER/+}$ hearts in comparison to Cyclin T1 infected $R26^{LSL-CMER/+}$ control hearts was indicated by a significant increase in the slope of premature mRNA plotted against mature mRNA (Supplementary Figure 6c). In contrast, no slope change was observed when values were plotted for highly expressed genes in the heart (Supplementary Figure 6c), indicating that Myc activation in the presence of ectopic Cyclin T1 does not stimulate global amplification of the pre-existing transcriptional repertoire in the adult heart, at least at 4 hours after Myc activation. Moreover, these transcriptional changes *in vivo* (Figure 6c) were comparable to those observed in adult cardiomyocytes *in vitro* (Figure 4e). Thus, the increase in Myc transcriptional activity in the presence of ectopic Cyclin T1 is sufficient to drive normally quiescent mature adult cardiomyocytes into the cell cycle, as evident by an increase in Ki67, phospho-H3 and cardiomyocyte number.

2. Figures 4, 5 and 6 compare mice with and without Cre (which excises the LSL cassette), but lack the obvious control without OHT. Without this control, the effects seen cannot be unequivocally attributed to the MYCER chimera. This control needs to be added to all panels in these figures. This is particularly relevant since the authors in the discussion spent a long time discussing that this is one of the first studies looking at immediate effects of MYC activation (which is a bit strange since there are dozens of papers on early responses, some

of them from the authors own laboratory) and discussing the merits of the MYCER system, which this year is exactly 30 years old (Eilers, M. Nature, 1989. 340(6228): p. 66-8).

We apologise to the reviewer if our explanation of the genetic strains used in the manuscript were unclear. Please allow us to clarify:

For Figures 1-4, 5a-d and Supplementary Figures 1-4, 5a-e and 6a, d, e-g, we have utilised the $R26^{CMER}$ allele in which the lox-stop-lox element present in the parent strain has been excised from the germline. Thus, these mice constitutively express MycER^{T2} in all tissues analysed, as demonstrated in Figure 1d. For controls we either treated $R26^{+/+}$ mice with 4OHT or we treated $R26^{CMER/+}$ mice with oil as a vehicle control and compared these to $R26^{CMER/+}$ mice treated with 4OHT. We have now included all these extra controls in Figure 1c. As the vehicle treated $R26^{CMER/+}$ condition proved to be phenotypically comparable to wild-type 4OHT treated mice, we decided to proceed with the latter control only for the CHIP sequencing and RNA sequencing analysis.

For Figures 5e-i, 6 and Supplementary Figures 5f-i and 6b-c, h-i, we utilised the $R26-IsI-CMER$ allele in which expression of MycER^{T2} is conditional on prior Cre recombinase action. We crossed this allele with a transgenic *Myh6-Cre* strain such that Cre expression is restricted to cardiomyocytes. This allowed targeted expression of MycER^{T2} exclusively in the cardiomyocytes. Cardiomyocyte-restricted MycER^{T2} expression also overcame the deleterious effects of ubiquitous Myc activation across all tissues. To address the reviewer's concerns, we have now included the untreated Cre control for Figure 5h (now Figure 5g). Supplementary Figure 6 includes AAV-RFP, 4OHT treated mice as controls for both adenoviral and Myc expression.

We apologise if we gave the impression that the MycER^{T2} switchable system was novel. We were rather trying to highlight the novelty of a genetic model that allows activation of the same elevated level of MycER^{T2} in all tissues of a mouse, so allowing us to investigate whether the Myc transcriptional response is tissue-dependent. To clarify this, we have made the following changes to the discussion:

“In addition, previous studies have been limited by an inevitable focus on the longer-term outcomes of Myc activation, which makes it very difficult to parse the direct impact of Myc from the indirect pleiotropic consequences of the primary programmes that Myc engages across different tissues.”

“Second, replacing Myc with the well-validated, reversibly switchable 4-OHT-dependent MycER^{T2} variant allows the analysis of the immediate and direct impact of Myc on its target genome through rapid and synchronous MycER^{T2} activation by 4-OHT^{52,53}.”

3. Figure 6a shows clearly that adult heart expresses very low levels of CDK9 and of RNAPII. Consistent with this, ectopic expression of cyclin T1 has no effect on the phosphorylation of Ser2 of the CTD, the substrate of CDK9; this is shown in Figure 4d, a panel that is shown but not discussed in the text. The finding that cyclin T1 expression restores gene expression is therefore surprising and shows that cyclin T1 must have another CDK partner. The authors need to identify the CDK partner for cyclin T1 in the restored hearts. Could it be that it

switches to other CDK's, as seen in some CDK knockout animals? This is also relevant since the authors claim to "restore neonatal levels of P-TEFb activity", but do not show this. All reviewers expressed concern that we had not shown that overexpression of Cyclin T1 compensates for the low/absent expression of both Cyclin T1 and CDK9. We completely agree with the reviewer that the increase in phospho-Rpb1 in Figure 4d is subtle. This experiment was very technically challenging due to the small number of adult cardiomyocytes that can be successfully maintained *ex vivo*. It is our oversight that we did not show increased CDK9 after Ccnt1 expression in our AAV adult system.

We have now included a Western blot in Figure 6b showing that elevated ectopic expression of Cyclin T1 drives an increase in CDK9 and phosphorylated RNA PolII in the adult heart. This result is supported by previously published data implicating Ccnt1 protein level as the key factor in controlling the amount of the P-TEFb complex in a cell^{10,11}. Others have demonstrated that whereas transgenic expression of CDK9 had no effect on mouse heart transgenic mice with elevated expression of Ccnt1 exhibited increased levels of both CDK9 and phosphorylated RNA PolII^{12,13}. These data indicated that that Cyclin T1 levels dictate CDK9 stability and P-TEFb activity. To clarify this, we have made the following changes to the results section of the manuscript:

"As described previously^{47,49} and confirmed here, elevated expression of Ccnt1 in cardiomyocytes leads to an increase in CDK9 and phosphorylated RNA PolII (S2) (Fig. 6b)."

We do however appreciate that the claim to "restore neonatal levels of P-TEFb activity" is an overstatement. To address this, we have made the following changes in the Discussion:

"Remarkably, we show that an increase in the level of P-TEFb activity in the adult heart is sufficient to support ectopic Myc-dependent transcription, proliferation and cytokinesis of cardiomyocytes *in vivo*, resulting in increased heart size and cardiomyocyte number within a very short timeframe."

4. Independent of this point, the statement "Myc transcriptional output is determined by P-TEFb" is not supported by any data in this paper.

We have made the following changes to the manuscript:

"Myc transcriptional output is limited by P-TEFb activity."

5. The control staining for cyclin T1 in Figure 6a is not ok. The authors need to obtain more convincing data that cyclin T1 is indeed expressed, which percentage of cells expresses cyclin T1 and what the relative levels of expression relative to proliferative tissue is. Quantifications need to be added for Figure 6c, this is again hard to see. In Figure 6e, the label "per field" needs to be replaced by "% of cells". As recommended by the reviewer, we have quantified the percentage of cardiomyocytes that express Ccnt1 following AAV infection and added these data to the manuscript. We have added an image showing the endogenous levels of Ccnt1 expressed in liver as a reference.

In addition, we now express p-H3 quantification as a percentage (Figure 1c, Figure 5d, Figure 6f and Supplemental Figure 6d) in the revised manuscript. We would like to note that we had intended the Ki67 data simply to be supportive of the p-H3 quantification in Figure 6f. We have replaced Figure 6e with IF images clearly showing Ki67 positivity in cardiomyocytes stained with PCM1. For completeness, IF data is also included in Figure 5.

6. Similar to point 1: global transcription rates need to be shown for Figure 6g. It is very hard to imagine that this growth is not accompanied by an induction of general transcription. We have now performed RNA sequencing to compare the transcriptional consequence of activating ectopic Myc at 4 hours post 4OHT treatment in adult heart in the presence versus absence of ectopic levels of Cyclin T1. In contrast to the wild-type adult heart in which the activation of ectopic Myc results in the differential expression of only 341 genes, the activation of ectopic Myc in the adult heart in which Cyclin T1 is overexpressed results in the differential expression of 1483 genes. These genes significantly overlap with the UP DEGs in response to ectopic Myc in the other, Myc-responsive tissues. Consistent with the analysis in Figure 3d, these common UP DEGs appear to be involved in processes relating rRNA and gene expression.

We have also performed similar RNAseq analysis on RNA isolated from purified cardiomyocytes. As expected, the common, cross-tissue Myc target gene set was enriched when Myc was activated in the context of ectopic Cyclin T1 ($R26^{LSL-CMER/+}; Myh6-cre + AAV-CCNT1$) in comparison to its absence ($R26^{LSL-CMER/+}; Myh6-cre + no AAV$; Figure 6d). We determined the abundance of premature mRNA compared to mature mRNA for a given transcript for $R26^{LSL-CMER/+}; Myh6-cre + AAV-CCNT1$ vs. $R26^{LSL-CMER/+} (no cre) + AAV-CCNT1$ hearts, at 4 hours post Myc activation. As expected, common Myc targets show an increase in nascent transcription with ectopic Myc in the presence of AAV-CCNT1 in adult hearts (Supplementary Figure 6c). In contrast, when the analysis was performed on genes known to be highly expressed in the heart, no difference was observed (Supplementary Figure 6c). These results suggest that the activation of Myc in the presence of ectopic Cyclin T1 results in the activation of a discrete Myc-specific transcriptional programme rather than global amplification of the existing adult cardiomyocyte transcriptional repertoire.

We have modified the manuscript to include the following:

“An increase in the rate of transcription of common Myc targets in cardiomyocytes purified from Cyclin T1 infected Myh6Cre; $R26^{LSL-CMER/+}$ hearts in comparison to Cyclin T1 infected control $R26^{LSL-CMER/+}$ hearts was inferred by a significant increase in the ratio of premature/mature mRNA (Supplementary Fig. 6c) 4 hours after Myc activation. In contrast, this ratio was unchanged for genes known to be highly expressed in the heart (Supplementary Fig. 6c), indicating that activation of Myc in the presence of ectopic Cyclin T1 does not cause a global amplification of the pre-existing transcriptional repertoire of the adult heart.”

7. The authors alter the genes that they use from one panel to the next without explanation. A consistent set of genes should be analyzed in all panels.

We appreciate that the reviewer found our justification for the change in genes analysed between Figure 4c and Supplementary Figure 4d; and Figures 4e, 5b and 6b unclear.

In Figure 4c and Supplementary Figure 4d we determined whether Myc transcriptional activity was dependent on maintained P-TEFb activity. To address this, we used Myc-responsive liver tissue as our exemplar, selecting a panel of genes that we knew be commonly regulated by Myc across all tissues (*Cad*, *Gnl3*, *Smpdl3b*, *Polr3g*).

In contrast, in Figures 4e, 5b and 6b we demonstrate the effect of increasing P-TEFb activity on Myc transcriptional activity in the heart, a tissue that is refractory to Myc activation in the absence of ectopic Cyclin T1. Thus, we selected a panel of genes that respond to Myc in the liver, lung and kidney, but failed to respond in the heart (*Bzw2*, *Polr3d*, *Cdc25a*, *Pinx1*, *St6*). The one exception is *Cad*, a gene that responds to Myc in all tissues tested and included as a control.

We have included the following justification in the manuscript:

“AZ5576 effectively inhibited CDK9 activity, as attested by a decrease in phosphorylated Rpb1 levels (Fig. 4b) and, notwithstanding the presence of active MycER^{T2}, such inhibition of CDK9 significantly attenuated transcription of *Smpdl3b*, *Cad*, *Gnl3* and *Polr3g*, genes regulated by Myc in multiple tissues (Fig. 4c).”

“Conversely, ectopic over-expression of Cyclin T1 in adult *R26^{CMER/+}* cardiomyocytes *in vitro* efficiently abrogated their normal refractoriness to Myc and permitted efficient expression of Myc target genes which were previously shown to be unresponsive in the adult heart (*Bzw2*, *Pinx1*, *Polr3d*, *St6* and *Cdc25a*) (Fig. 4d and e, Supplementary Fig. 4f and g).”

8. On a different note: have the authors considered the question of why endogenous MYC is expressed and chromatin bound in a non-proliferative tissue? I understand that it can be re-awakened, but what does it do without cyclin T1? Some insight into this would greatly strengthen the paper.

We apologise that we have not clearly conveyed the relevance of endogenous Myc expression in our experimental system.

In generating *R26-CMER* mice we aimed to express similar, superphysiologically high levels of MycER^{T2} expression across all tissues in an adult mouse to determine tissue-specific responses to Myc activation. Figure 1 demonstrates that the proliferative response to MycER^{T2} activation varies in different tissues (Fig. 1b and c) and falls into three general classes: 1. Adult tissues such as liver, lung and pancreas, with normally very low levels of endogenous Myc (Fig. 1d), but which are capable of significant regeneration after injury. Such tissues showed a marked induction of proliferation upon Myc activation. 2. Adult tissues, such as kidney, heart and brain, with limited capacity to regenerate, that demonstrate only a negligible rise in proliferation. 3. Tissues with constitutively high proliferative rates and substantial constitutive levels of endogenous Myc (Fig. 1d), such as thymus and spleen, in which activation of ectopic Myc elicits no significant additional proliferation above their already high basal levels.

By way of clarification, endogenous Myc is not expressed at appreciable levels in adult tissues with an intrinsically low proliferative index, such as the liver and heart (Figure 1d),

and we detect very few Myc binding sites in these tissues when isolated from wild-type animals (Figure 2a).

To aid clarification of this point, we have made the following changes to the manuscript:

“2. Adult tissues, such as kidney, heart and brain, with normally low levels of endogenous Myc and a limited capacity to regenerate, demonstrated only a negligible rise in proliferation.”

“We observed a dramatic increase in gene occupancy upon MycER^{T2} activation in both tissues, from a few hundred sites in wild-type control, reflecting the low levels of endogenous Myc expression in these tissues, to ~30,000 after Myc activation (Fig. 2a). Hence, the failure of the heart to proliferate in response to Myc activation cannot be attributed to the inability of ectopic Myc to access its target genes.”

“Hence, tissue non-responsiveness to Myc is not due to failure of ectopic Myc to access its chromatin targets.”

Reviewer #2 (Remarks to the Author):

Bywater et al. present a compelling account of the role of Myc-driven recruitment of P-TEFb and subsequent activation of RNAPII in the proliferative capacity of different tissues in mice. The authors created a series of mice with a range of expression levels of MycERT2, and using 4-OHT induced systemic Myc activity, promote a variety of tissue-dependent proliferative responses. Some tissues, such as the liver, showed a proliferative response, while others, such as the heart, appeared unresponsive to Myc activation. These differences are not due to differential binding of Myc to chromatin in these tissues, but appear to be a result of differences in the downstream capacity of Myc to drive transcriptional changes in different tissues. The authors attribute these changes to differences in the variations of P-TEFb (cyclin T1/Cdk9) levels, and subsequent RNAPII phosphorylation. Using a P-TEFb inhibitor, the authors demonstrated that this complex is required for Myc-driven proliferation of the liver, while overexpression of Cyclin T1 in the heart appears sufficient to allow this tissue to respond proliferatively to MYC.

This is an interesting manuscript that begins with an elegant and extremely informative system to understand how tissues respond to Myc activation. In general, the experiments are well conducted and controlled, and interpreted responsibly. The results themselves are extremely interesting, and will be of wide interest to those studying Myc and in developing ways to regenerate heart tissue.

That said, the story does start to unravel as the mechanism of differential response to Myc is investigated. The experiments with Cyclin T1, are provocative, but don't go far enough and are not adequately pursued to support the kind of mechanism the authors propose. The effects of Cyclin T1 expression in primary cardiomyocytes on p-Rpb1(S2) is modest, to say the least. In mice infected with adenovirus for overexpression of Cyclin T1 in cardiomyocytes, only immunohistochemistry of Cyclin T1 was used to confirm this overexpression; there is no validation that this is altering the phosphorylation status of Rpb1, which seems a major oversight. The authors have not looked to see whether CDK9

and Rbp1, which both apparently have low levels in the heart, are induced by overexpression of Myc and/or Cyclin T1 (which could explain the response).

We would like to thank the reviewer for her/his comments. We completely agree it was an oversight not to include our data showing increased levels of CDK9 and phosphorylated RNA PolII following Ccnt1 overexpression in the heart.

As detailed above for reviewer 1, we have now included evidence (Figure 6b) demonstrating that the overexpression of Cyclin T1 drives an increase in CDK9 and phosphorylated RNA PolII in the adult heart. This is consistent with previously reports demonstrating Ccnt1 protein level as the key factor controlling the amount of the P-TEFb complex in a cell^{10,11}. Moreover, while transgenic expression of CDK9 had no discernible effect on the heart, the Ccnt1 transgene led to increased levels of both CDK9 and phosphorylated RNA PolII^{12,13} demonstrating that Cyclin T1 levels dictate CDK9 stability and activity. To clarify this, we have made the following changes to the manuscript:

“As described previously^{12,13} and confirmed here, elevated expression of Ccnt1 in cardiomyocytes leads to an increase in CDK9 and phosphorylated RNA PolII (S2) (Fig. 6b).”

And most problematic for the proposed mechanism is the timing of the mechanistic experiments, which take place over relatively long time frames (days), making it difficult to know if the effects are direct or indirect.

It is indeed difficult to disentangle direct and indirect targets of Myc. Myc has been shown to have a large transcriptional repertoire and consequently its activation leads to a cascade of events which result in RNA amplification and cellular proliferation. To overcome this complication, we have employed an acutely switchable genetic system that allows us to determine the proximal consequences of Myc activation.

All of the RNA and ChIP sequencing reported in this manuscript was performed at a very early time point after Myc activation (4 hours) with the precise aim of restricting our analysis to the early, direct, Myc targets. This time point is well before global RNA amplification is manifest (Supplementary Figure 3c). We see no evidence of global RNA amplification of nascent transcription at this early stage (Supplementary Figure 3d-e). Hence, we conclude that combined Myc ChIP sequencing and RNA sequencing analysis under these conditions allows us to determine transcriptional effects that are most likely directly due to Myc.

However, any assessment of the impact of Myc activation on cellular proliferation requires analysis over a longer timeframe. Previously we determined that S phase entry of mouse embryo fibroblasts *in vitro* is around 16 hours post activation of ectopic MycER^{T2} with 4OHT. So clearly, we cannot assess any concrete impact of Myc activation on cell cycle progression until this delayed time-point. This problem is exacerbated in cardiomyocytes, which have shown to have an inter-mitotic time of around 30 to 35 hours¹⁴. The fact that Myc exerts its immediate transcriptional impact by 4 hours in both fibroblasts and cardiomyocytes, yet each cell type then takes its own time to enter S phase as a consequence, illustrates the fact that S-phase entry is dependent upon processes collateral to the activation of direct Myc target genes. Nonetheless, it is still valid to argue that the activation of Myc is the ultimate, obligate driver of this fate, as we demonstrated previously¹⁵. This is supported by the

observed enrichment for gene sets involved in cell cycle progression 4 hours after Myc activation in both 15 day old hearts (existing data-set) and AAV-CCNT1 infected adult hearts (new data-set). We have included these data in the revised manuscript (Supplementary Figure 5e and Supplementary Figure 6b).

Reviewer #3 (Remarks to the Author):

The manuscript by Bywater et al explores the physiological response of different tissues to acutely increased Myc activity. They focus on Myc's capacity to elicit mitosis as a function of Myc level. They use a series of tamoxifen-activatable MycERs knocked into the Rosa locus to enable controlled increases in Myc activity and the look at liver, lung, heart and kidney. First, they find that Myc binds at all open chromatin sites in each tissue, at both common genes that are commonly expressed as well as at genes that have tissue restricted expression. MycER activation elicited three response patterns in adult tissues: in already proliferating tissues such as thymus and spleen, proliferation persisted, but was not augmented. Resting tissues with mitogenic potential were driven to proliferate by increased Myc. Tissues lacking proliferative/regenerative potential were not induced to proliferate by activation of MycER. In general, the magnitude of the transcriptional response was attenuated in non-proliferative tissues. The authors next explore the hypothesis that Myc action is limited in non-proliferative tissues by a generalized insufficiency of transcription. Examination of extracts from the different tissues reveals that P-TEFb is reduced in non-proliferative tissues but increasing P-TEFb activity by increasing cyclin T levels confers Myc-induced proliferation on these tissues. So adult heart with increased P-TEFb becomes Myc responsive. Young heart that retains the ability to proliferate has higher P-TEFb levels and is Myc responsive.

This work is creative, well-executed, conceptually important and replete with therapeutic implications. It provides a rationale for many peculiar phenomena associated with Myc. Nevertheless, there are few points that should be addressed.

1. The authors state that CDK9 phosphorylation of the RNAPII CTD is what drives the transition from pausing to productive elongation. Actually, it is still debatable which CDK is the principle kinase of Ser2. A case can also be made for CDKs 12 and 13. But probably a more important issue is that the authors incorrectly seem to imply that Ser2 phosphorylation is the key event in pause release. Actually, probably more important substrates for CDK9 for pausing and pause release than Ser-2 are DSIF (spt4/5) and NELF. It would be nice to comment on (or better yet show) the amounts and the phosphorylation status of these complexes in the different tissues.

Prompted by your comment, we assessed the expression of NELF-B (COBRA1), NELF-C/D (TH1L), SPT4 and SPT5 across the heart, liver, lung and kidney and included these data in Supplementary Figure 4a. The heart expresses very low levels of these proteins.

2. In figure 4, total Rpb1 levels also seem to follow the proliferative potential of the tissues. The authors should comment on whether or not other transcription machinery components besides P-TEFb might also contribute to the proliferation block.

We thank the reviewer for her/his comment and the opportunity to expand on this interesting point. Indeed, we do observe that the ability of a tissue to respond to Myc at the transcriptional level correlates with the level of Rpb1. It is, therefore, possible the RNA

Polymerase II machinery is also limiting for Myc transcriptional activity in the refractory tissues such as the heart. Interestingly, there is evidence suggesting that Myc-mediated transcriptional repression is more usually indirect and likely due to increased competition for limiting components of the core transcriptional machinery^{16,17}. Consistent with this, we find that Myc-repressed genes are by and large those constitutively highly expressed in each tissue and generally associate with its normal function. We speculate in the discussion that, in addition to driving a core transcriptional programme required for mitotic cell cycle progression, Myc competes for limiting core transcriptional components necessarily resulting in the transcriptional repression of genes involved in specifying differentiated functions, some of which may be incompatible with the process of proliferation. We have made appropriate changes to the revised manuscript to highlight this point.

We refer the reviewer to the changes in the relevant paragraph in the discussion:
“A simple hypothetical explanation for the pervasive suppression of differentiation by Myc is that certain components of transcriptional machinery, such as P-TEFb or even total levels of RBP1, available for loading onto promoters are limiting, in which case repression of differentiation-specific genes is simply a consequence of their redeployment to Myc target genes, a mechanism recently proposed in 3T3 fibroblasts¹⁶. Analogous scarcities in transcriptional machinery could also explain why some tissues fail to respond transcriptionally to Myc. Myc has been shown to modify both RNA PolII loading and its phosphorylation by P-TEFb^{16,18–23}. Prompted by the recent observation that MYC overexpression is the key molecular determinant dictating sensitivity to CDK9 inhibition in hepatocellular carcinoma²⁴, we showed that responsiveness of individual tissues to Myc correlates with the levels of expression of the basal transcription factors, P-TEFb and RNA PolII, the pause factors, DSIF and NELF, and, inversely, with the expression of the P-TEFb repression complex.”

3. The results in this manuscript indicate that the effects of Myc relate to quantitative changes in global transcription more than to qualitative changes in the transcription profile. This would seem to be in accord with the suggestion in refs. 25 and 26 that Myc is an amplifier. Considering that this has been a matter of some disagreement in the field, perhaps the authors could comment on this.

Thank you for your comment. We agree and have addressed the issue of the general amplifier model above (Reviewer 1, comment 1) and added relevant data to Supplementary Figure 3e-d and 6c.

Reviewer #4 (Remarks to the Author):

In this highly interesting contribution Bywater et al describe that the responsiveness of cells to pro-proliferative signals provided by the oncogene Myc depends on the presence of P-TEFb, a transcriptional co-factor complex consisting of Cdk9 and cyclin T1. The authors found that cells which respond by proliferation to increased expression of Myc (i.e. liver cells) and those that don't (i.e. cardiomyocytes), both bind Myc but show differential transcriptional responses. The authors demonstrate that proliferative responses to increased Myc activity in the liver depends on CDK9 activity. Activation of Myc in P15 cardiomyocytes, which unlike adult cardiomyocytes still contain high levels of P-TEFb, induced typical Myc-dependent transcriptional responses and robust proliferation. Importantly, adenovirus-mediated overexpression of cyclin T1 in adult hearts restored the

ability of cardiomyocytes to respond by proliferation to increased Myc activity. Overexpression of both Myc and cyclin T1 in adult cardiomyocytes in vivo resulted in a remarkable increase in cardiomyocytes numbers and increased heart size without changes in cardiomyocyte volume.

A better understanding of the mechanisms that prevent proliferation of cardiomyocytes even under strong pro-proliferative conditions is direly needed, since enablement of cardiomyocyte proliferation in adult mammalian has been postulated to provide the basis for strategies to regenerate damaged hearts. The current study seems to provide at least some important clues, although direct translational approaches based on the overexpression of a highly potent oncogene are difficult, to say the least. The study contains several strong data sets. However, there are several important conceptual and technical problems that need to be addressed.

Specific comments

The authors describe that global activation of Myc (expression of a Myc-ER fusion in oocytated followed by a tamoxifen pulse) induced robust cell proliferation in liver, lung and pancreas but not in the heart or brain. I do not understand why the authors did not observe proliferative responses in heart TISSUES. The myocardium contains roughly four times more non-cardiomyocytes than cardiomyocytes, although cardiomyocytes provide the largest part to the volume/mass of the heart. It makes perfect sense that cardiomyocytes exposed to increased Myc activity remain cell cycle arrested but why is there no Ki67 staining and BrdU incorporation in heart and the brain tissues (Fig. 1b)? For example: endothelial cells are rather sensitive to increased Myc levels and should respond accordingly.

We thank the reviewer for her/his comment. As suggested, we do see a very small number of p-H3⁺ endothelial, immune and epicardial cells (images are included below) in the heart in mice with ubiquitous MycER^{T2} activation.

Whereas p-H3 marks mitosis, the Ki67 antigen is expressed throughout the cell cycle and Ki67 immunohistochemistry reveals a small but significant increase in proliferation in the heart that is restricted to non-cardiomyocytes (shown below).

We have not included these data as we don't believe it aids in the interpretation of Figure 1. However, we have made the following change in the manuscript:

“Of note, quantification of the broad proliferative marker Ki67 revealed a small but significant increase in some cells within the heart in response to global Myc activation. However, these changes were confined to non-cardiomyocytes (data not shown).”

Related to the above topic: it might be helpful to determine expression (and nuclear localization after tamoxifen treatment) of MycER by immunostaining and not by western blot, which will miss absent Myc induction in specific cell types in different organs. Since the density of cells is different in different organs, ratios between total numbers of nuclei and p-H3, Ki67 positive nuclei should be given.

We appreciate the reviewer’s suggestion to assess Myc function. However, the Myc protein contains nuclear localisation signals that effectively translocates MycER^{T2} into the nucleus irrespective of the presence of the 4-OHT. Hence, subcellular location cannot be used as an indicator of MycER^{T2} activity. As requested, we have now displayed the quantification of p-H3 staining as a percentage of nuclei in all relevant figures.

The authors found that CDK9 and Cyclin T1 levels are virtually absent (CDK9) or much lower in the heart compared to the liver (Cyclin T1). The P-TEFb complex consist of both CDK9 and Cyclin T1. Hence, it is difficult to understand why the overexpression of cyclin T1 compensates for the low/absent expression of both cyclin T1 and CDK9, in particular since inhibition of CDK9 in the liver prevented Myc induced proliferation. Does overexpression of cyclin T1 (with or without Myc activation, causes up-regulation of CDK9 in adult cardiomyocytes?

We now provide evidence that Cyclin T1 stabilises CDK9, consistent with existing evidence in the literature. Please see the explanation and extra data described above (Reviewer 1, comment 3; Reviewer 3, comment 1).

While cyclin T1 is still detectable in adult hearts, CDK9 expression is essentially absent, which makes it surprising that the authors used Cyclin T1 but not CDK9 overexpression to restore P-TEFb activity. Does overexpression of CDK9 compared to Cyclin T1 transduction restore proliferation of cardiomyocytes in Myc transgenic mice?

Our decision to overexpress Cyclin T1 was based on data from Sano and colleagues demonstrating that transgenic overexpression of CDK9 alone has no effect in the heart, whereas overexpression of Cyclin T1 leads to increased levels of phosphorylated RNA PolII and cardiac hypertrophy^{12,13}. Analysis in other cells types has consistently shown that P-TEFb levels are dependent on the level of Cyclin T1 and not Cdk9^{10,11}. Consequently, we

chose to ectopically express Cyclin T1 alone. We have now provided formal evidence that elevated expression of Cyclin T1 does, indeed, increase Cdk9 and RNA PolII (S2) in the heart (Figure 6b). We have not attempted to over-express CDK9 alone as the evidence strongly indicates it will not increase P-TEFb activity. It is worthy of note that CDK9 is expressed in the adult heart, albeit at a much reduced level when compared with juvenile heart or liver (Figure 5a).

Accordingly, we have made the following changes to the manuscript:

“As described previously^{12,13} and confirmed here, elevated expression of Ccnt1 in cardiomyocytes leads to an increase in CDK9 and phosphorylated RNA PolII (S2) (Fig. 6b).”

The authors use phosphorylation of the C-terminal end of Pol II (Rpb1) as a readout for P-TEFb and Pol II activity. However, Pol II is highly active (and phosphorylated) on genes required for cardiomyocyte function. The low phosphorylation of Pol II at ser 2 and ser5 in cardiomyocytes is therefore very hard to understand. Several other kinases (e.g. CDK8) although phosphorylate Rpb1. How can Pol II actively transcribe genes in cardiomyocytes without phosphorylation at ser 2 and ser5?

We agree with the reviewer, this is a really interesting question. Although both total and phosphorylated levels of Rpb1 are significantly lower in the adult heart in comparison to the liver, they are not absent (longer exposures of Figure 5a that show this are now included in Supplementary Figure 5b). We reason that these low levels of Rpb1 must at least be sufficient to allow transcription of heart specific genes required to maintain heart function. Indeed, when we performed a PolII ChIP in the adult heart we detected heart specific genes loaded with significant amounts of PolII.

There are several potential mechanisms that may concentrate the available PolII at specific locations:

First, it is possible that the generation of “transcriptional hotspots” might concentrate essential factors at the relevant genes²⁵. Second, transcription in the presence of low Rpb1 may be aided by tissue-specific super enhancers that recruits a large portion of the enhancer-associated PolII and its associated cofactors and chromatin regulators²⁶. It is interesting to note that Myc activation represses genes encoding tissue-specific differentiated functions (Figure 3e, g and h and, Supplemental Figure 3a and b) in all tested tissues. This may indicate that Myc commandeers PolII and general co-factors from genes that maintain tissue differentiation. Unfortunately, testing of these hypotheses lies outside the scope of this manuscript.

The authors propose that increased P-TEFb activity specifically enhances transcriptional elongation of Myc-primed promoters. There are techniques available (CoPro-Seq, PRO-Seq) to prove this hypothesis. Although such experiments would provide further mechanistic insights, they are not mandatory.

We agree that the kinetics of transcriptional elongation of Myc target genes in this model might be revealing. We had previously considered the possibility of GRO-seq, the major caveat being the requirement to isolate intact nuclei for transcription *in vitro*. Our experience was that such isolation requires lengthy protocols involving harsh enzymatic digestion and considerable collateral cell death, and that are differently tailored for each

tissue/organ type. This makes accurate and reliable comparisons across differing tissues very difficult to achieve and verify. We also considered indirect assessment of elongation by performing Spt5 or p-Rpb1 ChIP sequencing. Although a worthy line of investigation, we suggest this lies beyond the scope of the current manuscript.

The study leaves several important questions unanswered. Of course, not of all them can be answered when first reporting a highly interesting observation. Nevertheless, one would like to learn more about the strong downregulation of CDK9 and cyclin T1 during cardiomyocyte maturation. What is responsible for the downregulation of CDK9 and cyclin T1 after P15? A host of signalling and metabolic changes occur as the neonate heart develops into the adult organ. These include, but are not limited to, polyploidisation, changes in cardiomyocyte structure, the development of postnatal endothermy, an increased metabolic state, high oxygen concentration and increased blood pressure^{27,28}; any of which could be responsible for the downregulation of P-TEFb. Interestingly, both the neonatal mouse heart and zebrafish larval heart express high levels of P-TEFb. However, where the zebrafish heart retains 70% of the larval levels of P-TEFb and the ability to regenerate throughout development, P-TEFb levels in the mouse heart drop to 15% and regenerative ability is lost^{12,29}. It is possible that comparison of the regulation of P-TEFb between maturing Zebrafish and mammals might provide insights to the mechanisms regulating P-TEFb levels.

The *Ccnt1* promoter exhibits remarkable redundancy with many features associated with typical housekeeping genes^{30,31}. It has a number of predicted transcription factor (TF) binding sites, none of these is individually essential for its transcription. The most conserved of these transcription factor recognition elements are four ETS binding sites, but again, *Ccnt1* transcription is still not entirely dependent on any or all of them. It is possible that the *Ccnt1* promoter has evolved to be robustly active in a wide variety of cell types, both proliferating and non-proliferating, in which different TFs are present³¹.

Ccnt1 is also regulated post-transcriptionally and protein stability responds to mitogens and cytokines³², indicating that the level of *Ccnt1* is actively regulated to promote transcription in cells stimulated to proliferate. Notably, the mitogen activated Ras signalling pathway promotes *Ccnt1* stabilisation³³⁻³⁵ and oncogenic Ras signalling increases *Ccnt1* expression in the heart leading to cardiac hypertrophy^{36,37}.

Elevated *Ccnt1* expression in the heart leads to a reduction in PGC-1 (PPARGC1A), a master regulator of mitochondrial biogenesis and function¹³. The reduction in PGC-1 leads to mitochondrial dysfunction which can be rescued by ectopic addition of PGC-1. These data imply that low levels of P-TEFb in the adult heart are essential to maintain mitochondrial function. Consistent with this notion, reduction in P-TEFb levels in the developing mouse heart coincides with weaning by which time glucose oxidation predominates³⁸.

We have included a brief summary of these points in the manuscript:

“Interestingly, both the neonatal mouse heart and the zebrafish larval heart express high levels of P-TEFb. Whereas the zebrafish adult heart retains 70% of the larval levels of P-TEFb and the ability to regenerate throughout development, P-TEFb levels in the adult mouse

heart drop to 15%^{47,73}. Little is known regarding the regulation of P-TEFb during mammalian cardiac development. Levels of Cyclin T1 are regulated at the transcriptional and post-transcriptional level¹¹⁻¹⁴, with mitogens and cytokine signalling known to increase Cyclin T1 protein stability^{74,75}. We and others have shown that the level of cyclin T1 is the key factor regulating the level and activity of P-TEFb within cardiomyocytes^{47,49}. Oncogenic Ras signalling increases Cyclin T1 in the heart, leading to cardiac hypertrophy⁶⁰. It is tempting to speculate that the noted oncogenic cooperation between Ras and Myc may, in part, rely on a Ras-dependent increase in P-TEFb, promoting Myc-driven transcription.”

Furthermore: does the increased cardiomyocyte proliferation after Myc activation and cyclin T1 transduction restore heart regeneration in adults?

We agree. This is an exciting avenue of investigation that we are actively pursuing, but lies outside the scope of this current manuscript. Long-term expression of ectopic *Ccnt1* in the heart has been shown to cause hypertrophy^{12,13}. Deregulated Myc activity is a feature of most, if not all, cancers and in conjunction with persistent high levels of Cyclin T1 may drive unregulated proliferation. Furthermore, experimental induction of even modest constitutive proliferation of cardiomyocytes results in heart failure^{39,40}. The key to developing cardiac regenerative therapy will likely rely on transiently switchable systems.

Is cardiac function compromised due to increased Myc activity/cyclin T1 expression? Do the animals develop a tumor after extend Myc/cyclin T1 activation (or die in between because of heart failure)?

Ubiquitously high levels of Myc activity is not tolerated in adult mice for longer than 4 to 6 days. Proliferative organs in such animals rapidly increase in size and the mice die of some form of systemic organ failure. For example, persistent elevated expression in the adult liver causes significant liver enlargement and mortality within 6 days. Thus, precocious Myc-driven proliferation in a single organ is not tolerated. We have not detected any tumours in this short time period.

Therefore, to perform more protracted analysis of Myc activation in the heart, we utilised the cardiomyocyte-specific cre strain, *Myh6-cre*, to direct expression of MycER^{T2} only in cardiomyocytes. In 15 day-old *Myh6-Cre; R26^{LSL-CMER/+}* mice we collected tissues 48 hours after activation of MycER^{T2} with a single injection of tamoxifen. These mice exhibited a near doubling in heart size and cardiomyocyte number (Figure 5d and g). In contrast, two injections of tamoxifen 1 day apart that maintained MycER^{T2} activity for around 48 hours resulted in the death of the mice 4 days later. The hearts of these mice had quadrupled in size, consistent with extensive BrdU incorporation and Ki67 and p-H3 immunoreactivity, and contained large numbers of cardiomyocytes displaying disassembled sarcomeres. We have included these new data in Supplementary Fig 5i and added the following sentence to the manuscript:

“Over the longer term, transient elevated MycER^{T2} activity in 15 day old *Myh6Cre; R26^{LSL-CMER/+}* mice caused a large increase in heart size, increased p-H3 immunoreactivity, a large proportion of cardiomyocytes displaying disassembled sarcomeres and mice did not survive beyond 4 days post MycER^{T2} activation (Supplemental Fig. 5i).”

In contrast to the 15 day-old study above, transient activation of MycER^{T2} in AAV9-Ccnt1 treated adult *Myh6-Cre;R26^{LSL-CMER/+}* mice results in a concomitant burst of cardiomyocyte proliferation limited to the ~30% cells that also co-expresses elevated Ccnt1, resulting in 3 to 5% p-H3 positivity. Despite the induction of proliferation and subsequent increase in cardiomyocyte number observed at 48 hours after Myc activation, such mice survive to at least 28 days post MycER^{T2} activation with no histological abnormalities visible by H&E or Gomori trichrome staining. We have included these data in Supplementary Figure 6i and added the following sentences to the manuscript:

“A single transient burst of MycER^{T2} activity in Cyclin T1 expressing *Myh6Cre; R26^{LSL-CMER/+}* adult cardiomyocytes was compatible with long-term survival, with no histological abnormalities detected at 28 days after MycER^{T2} activation (Supplementary Fig. 6i).”

“Importantly this transient wave of proliferation induced in a smaller fraction of total cardiomyocytes than in juvenile mice is compatible with long-term survival. It will be of great interest to determine if this genetic combination will prove beneficial in models of cardiac injury.”

A comparison of the abundance of MycER DNA binding between heart and liver at promoter regions of mitotic cell cycle gene is missing. The authors should include a histogram of Myc DNA binding at such promoter regions.

A comparison of Myc binding across promoters regulating mitotic cell cycle genes was presented in the heatmap in original Supplementary Figure 2d. We have now included data on Myc ChIP sequencing signal enrichment for mitotic cell cycle gene promoters compared to heart-specific and liver-specific Myc-bound promoters (Supplementary Figure 2e).

There are several technical problems, which need to be addressed. A large part of the data relies on profiling of tissues rather of individual cell types, which is a severe shortcoming and might compromise the results. The authors need to use purified cardiomyocytes for all relevant experiments and not only for some. This is less of a problem when mice with cardiomyocyte-specific Myc expression are used but even after cardiomyocyte-specific expression of effectors, secondary changes in neighboring cells in the tissue might create confounding effects. Just as one of many examples: in Fig5c a heatmap of expression changes in different organs is shown. As in other experiments, the authors have to use isolated cardiomyocytes for the analysis.

We have now performed RNA sequencing in isolated cardiomyocytes in order to confirm that the ectopic expression of Cyclin T1 can facilitate Myc-driven transcriptional activation specifically in this cell type (Figure 6d and Supplementary Figure 6c). We have already demonstrated that Myc binding in the whole heart correlates with accessible regions of chromatin in purified cardiomyocytes (Supplementary Figure 2c).

The quantification of cycling and proliferating cardiomyocytes is not sufficient. Counting of p-H3 positive cardiomyocytes is notoriously difficult and yields unreliable data. The authors have to isolate cardiomyocytes first before counting BrdU or p-H3 positive cells to be sure that only cardiomyocytes are counted. An alternative approach is FACS-sorting of PCM1-positive nuclei after EdU injection. PCM1 specifically labels cardiomyocyte nuclei allowing specific and quantitative monitoring of cell cycle activity in cardiomyocytes.

Quantification of p-H3 immunohistochemistry on sections of the whole heart was used as an indication of cardiomyocyte proliferation (specifically, late G2 and mitosis). To confirm cardiomyocyte-specific cell cycle entry, we have now co-stained for Ki67, PCM-1 and cardiac troponin (Figure 5e and 6e). These data are in addition to p-H3, WGA and cardiac troponin co-staining that also confirm cardiomyocyte-specific mitosis (Figure 5e and 6e). Furthermore, we have included images of images of p-H3⁺ cardiac troponin⁺ isolated cardiomyocytes (Figure 5h). Finally, we quantified the total number of cardiomyocytes in the resulting enlarged hearts to demonstrate the increase in cardiomyocyte number (Figure 5g and 6i).

AuroraB staining is a marker to identify mitosis of cardiomyocyte nuclei, which might result in multinucleation. The authors need to stain for anilin to visualize the contractile ring forming during cytokinesis. The morphometric data are highly suggestive of cytokinesis. Nevertheless, one would like to see an actual example of dividing cardiomyocytes. I fail to see the evidence based on the EM images, the movies or the staining in Fig. 5, 6. Anillin and Aurora B change their localisation throughout the cell cycle. Both are expressed in the nucleus during G2. During metaphase, Anillin is detected in the cytoplasm and Aurora B associates with the metaphase chromosomes. During anaphase, Anillin is found at the cortex of the cell, and Aurora B is present at the mid-zone. During cytokinesis, both proteins are located at the mid-body⁴¹. Either protein can therefore be used to identify cytokinesis. However, the placement of the mid-body during cytokinesis can be used to distinguish if a cardiomyocyte will likely complete cytokinesis or fail and therefore increase ploidy^{42,43}. We have included images of all the stages of Aurora B localisation (Supplementary Figure 5f), including images specifically visualising central mid-body localisation of Aurora B (Figure 5e). We see both centrally located mid-bodies and laterally displaced mid-bodies (Supplementary Figure 5f). When combined with the large increase in cardiomyocyte number (Figure 5g) our data support the conclusion that there are cardiomyocytes completing cytokinesis.

We have modified the manuscript in the following way:

“Within 48 hrs, *Myh6Cre; R26^{LSL-CMER/+}* hearts displayed phospho histone H3 and Ki67 positivity specifically in cardiac troponin T and PCM1 positive cardiomyocytes, displayed Aurora B Kinase positivity at centrally located mid-bodies (Fig. 5e, Supplementary Fig. 5f) and had doubled in size compared with *R26^{LSL-CMER/+}* controls (Fig. 5f and g).”

“The large increase in cardiomyocyte number and Aurora B localization to centrally located mid-bodies suggests that cardiomyocytes isolated from juvenile mice can progress through the cell cycle and many can complete cytokinesis”

The authors should also determine the numbers of mono-, bi- and multinucleated cardiomyocytes.

We have now included analysis of cardiomyocyte nucleation in Supplementary Figure 6h.

We have modified the manuscript in the following way:

“Furthermore, 72 hours post MycER^{T2} activation in combination with Cyclin T1 overexpression led to a further increase in heart size accompanied by an increase in cardiomyocyte number without any change in cardiomyocyte size (Fig. 6g to i) or nucleation (Supplementary Fig. 6h).”

If I got it correctly (was not described explicitly), the authors determined the absolute numbers of cardiomyocytes in different conditions by comparing yields of cardiomyocytes isolated from different hearts. The authors have to describe how exactly how the numbers of cardiomyocytes were determined. Comparison of yields from different isolation experiments is not a reliable technique. Proper morphometric measurements are necessary. The method we used for determining the absolute number of cardiomyocytes is as follows: Whole hearts isolated from mice and washed in PBS were fixed in 1% PFA overnight at room temperature with agitation. The following day hearts were washed 4 times in PBS. Hearts were cut into 1-2 mm³ pieces and incubated with 0.5 U/mL collagenase B (Roche #11088807001) in 0.2% NaN₃/PBS and left to oscillate at 1000 rpm at 37 °C. Every 12 hours the cardiomyocyte supernatant was collected and stored at 4 °C in 0.2% NaN₃/FBS. Once dissociation was complete (~8 days) cells were centrifuged at 1000xg for 3 minutes, washed twice in PBS and stored in 0.2% NaN₃/PBS at 4°C. Cardiomyocytes number per heart was estimated by determining the average of eight individual haemocytometer counts.

Subsequent to isolation, cardiomyocytes were stained for p-H3 and cardiac troponin as follows:

All of the following incubations were carried out at room temperature. 500,000 cardiomyocytes were incubated with blocking buffer (4% BSA, +0.2% Triton X-100, +1mM EDTA, +0.02% sodium azide) and agitated (1000 rpm) at room temperature for 3 minutes. Cells were then re-suspend in primary antibodies (Cardiac Troponin T (13-11)(ThermoFisher, MA5-12969; 1:100) and P-H3 (Millipore, 1:400)) and incubated with agitation for 1 hour. After 2 washes in blocking buffer, secondary antibodies were incubated together with Hoechst and agitated for 1 hour at room temperature. Following 2 washes in blocking buffer cells were re-suspend in blocking buffer for flow cytometry or centrifuged onto slides and mounted for imaging. The average length and width of purified cardiomyocytes was determined with Image J software.

Although we find this method reproducible, to further reduce experimental variability fixed hearts were collected and stored and cardiomyocyte dissociation performed on all samples at the same time. The exception were untreated *Myh6-Cre;R26^{CMER/+}* hearts which have been included as an extra control in the revision process. We have included images of single cardiomyocytes obtained by this method (Supplementary Figure 5g) and clarified the procedure in the methods section.

At page 11 line 8 the authors stated that there are modest, but non-significant, responses in the heart. The authors should avoid such statements. Either the FC changes are significant or not. “Tendencies” are hard to grasp in a scientific manner.

We absolutely agree, references to non-statistically significant trends should be avoided. Although transcriptional changes in the heart are not significant, they are present, albeit highly attenuated, which we attribute to the limited availability of RNA PolIII. To convey this with a statistical comparison we have now confirmed the Myc-dependent enrichment of

genes that are commonly activated by Myc in other tissues (liver, lung and kidney) in the adult heart as well. This enrichment is significant despite none of these genes demonstrating a significant difference in expression individually (Figure 3j). We have included the following sentence into the revised manuscript.

“Indeed, gene-set enrichment analysis in the heart confirmed a significant trend in the increased expression of common Myc targets, even when defined as targets commonly upregulated in response to Myc in all tissues excluding the heart (Fig. 3j).”

The manuscript contains numerous grammatical errors and odd phrases, which should be corrected.

We are sorry for any grammatical errors that have slipped through our editing procedures. We hope we have corrected these in our revised manuscript.

References

1. Lin, C. Y. *et al.* Transcriptional amplification in tumor cells with elevated c-Myc. *Cell* **151**, 56–67 (2012).
2. Nie, Z. *et al.* c-Myc Is a Universal Amplifier of Expressed Genes in Lymphocytes and Embryonic Stem Cells. *Cell* (2012). doi:10.1016/j.cell.2012.08.033
3. Sabò, A. *et al.* Selective transcriptional regulation by Myc in cellular growth control and lymphomagenesis. *Nature* (2014). doi:10.1038/nature13537
4. Kress, T. R., Sabò, A. & Amati, B. MYC: Connecting selective transcriptional control to global RNA production. *Nature Reviews Cancer* 593–607 (2015). doi:10.1038/nrc3984
5. Tesi, A. *et al.* An early Myc-dependent transcriptional program orchestrates cell growth during B-cell activation. *EMBO Rep.* **20**, 1–14 (2019).
6. Manno, G. La *et al.* RNA velocity of single cells. *Nature* **560**, 494–498 (2018).
7. Furlan, M. *et al.* Genome-wide dynamics of RNA synthesis, processing and degradation without RNA metabolic labeling. *bioRxiv Prepr.* 1–26 (2019). doi:https://doi.org/10.1101/520155
8. De Pretis, S. *et al.* Gene expression INSPEcT : a computational tool to infer mRNA synthesis, processing and degradation dynamics from RNA- and 4sU-seq time course experiments. *Bioinformatics* **31**, 2829–2835 (2015).
9. Kress, T. R. *et al.* Identification of MYC-dependent transcriptional programs in oncogene-addicted liver tumors. *Cancer Res.* **76**, 3463–3472 (2016).
10. Keeffe, B. O., Fong, Y., Chen, D., Zhou, S. & Zhou, Q. Requirement for a Kinase-specific Chaperone Pathway in the Production of a Cdk9 / Cyclin T1 Heterodimer Responsible for P-TEFb-mediated Tat Stimulation of HIV-1 Transcription *. *J. Biol. Chem.* **275**, 279–287 (2000).
11. Chiu, Y., Cao, H., Jacque, J., Stevenson, M. & Rana, T. M. Inhibition of Human Immunodeficiency Virus Type 1 Replication by RNA Interference Directed against Human Transcription Elongation Factor P-TEFb (CDK9 / CyclinT1). *J. Virol.* **78**, 2517–2529 (2004).
12. Sano, M. *et al.* Activation and function of cyclin T-Cdk9 (positive transcription elongation factor-b) in cardiac muscle-cell hypertrophy. *Nat. Med.* **8**, 1310–1317 (2002).
13. Sano, M. *et al.* Activation of cardiac Cdk9 represses PGC-1 and confers a

- predisposition to heart failure. *EMBO J.* **23**, 3559–3569 (2004).
14. Hashimoto, H., Yuasa, S., Tabata, H., Tohyama, S. & Hayashiji, N. Time-lapse imaging of cell cycle dynamics during development in living cardiomyocyte. *J. Mol. Cell. Cardiol.* **72**, 241–249 (2014).
 15. Waters, C., Littlewood, T. D., Hancock, D., Moore, J. & Evan, G. I. c-myc protein expression in untransformed fibroblasts. *Oncogene* **6**, 797–805 (1991).
 16. De Pretis, S. *et al.* Integrative analysis of RNA polymerase II and transcriptional dynamics upon MYC activation. *Genome Res.* **10**, 1658–1664 (2017).
 17. Baluapuri, A., Hofstetter, J., Stankovic, N. D., Cramer, P. & Eilers, M. Article MYC Recruits SPT5 to RNA Polymerase II to Promote Processive Transcription Elongation Article MYC Recruits SPT5 to RNA Polymerase II to Promote Processive Transcription Elongation. *Mol. Cell* **74**, 674–687 (2019).
 18. Rahl, P. B. *et al.* C-Myc regulates transcriptional pause release. *Cell* **141**, 432–445 (2010).
 19. Kanazawa, S., Soucek, L., Evan, G., Okamoto, T. & Peterlin, B. M. c-Myc recruits P-TEFb for transcription, cellular proliferation and apoptosis. *Oncogene* **22**, 5707–5711 (2003).
 20. Eberhardy, S. R. & Farnham, P. J. Myc recruits P-TEFb to mediate the final step in the transcriptional activation of the cad promoter. *J. Biol. Chem.* **277**, 40156–40162 (2002).
 21. Shao, W. & Zeitlinger, J. Paused RNA polymerase II inhibits new transcriptional initiation. *Nat. Genet.* (2017). doi:10.1038/ng.3867
 22. Jonkers, I. & Lis, J. T. Getting up to speed with transcription elongation by RNA polymerase II. *Nat. Rev. Mol. Cell Biol.* **16**, 167–176 (2015).
 23. Herold, S. *et al.* accumulation of stalled RNA polymerase. *Nature* **567**, 545–549 (2019).
 24. Huang, C. H. *et al.* CDK9-mediated transcription elongation is required for MYC addiction in hepatocellular carcinoma. *Genes Dev.* **28**, 1800–1814 (2014).
 25. Hadzhiev, Y. *et al.* A cell cycle-coordinated Polymerase II transcription compartment encompasses gene expression before global genome activation. *Nat. Commun.* **10**, 1–14 (2019).
 26. Hnisz, D. *et al.* Super-enhancers in the control of cell identity and disease. *Cell* **155**, 934–947 (2013).
 27. Vivien, C. J., Hudson, J. E. & Porrello, E. R. Evolution , comparative biology and ontogeny of vertebrate heart regeneration. *Nat. Publ. Gr. Regen. Med.* **1**, 1–14 (2016).
 28. Hirose, K. *et al.* Evidence for hormonal control of heart regenerative capacity during endothermy acquisition. *Science (80-.)*. **364**, 184–188 (2019).
 29. Matrone, G. *et al.* CDK9 and its repressor LARP7 modulate cardiomyocyte proliferation and response to injury in the zebrafish heart. *J. Cell Sci.* (2015). doi:10.1242/jcs.175018
 30. Liu, H. & Rice, A. P. Isolation and characterization of the human cyclin T1 promoter. *Gene* **252**, 39–49 (2000).
 31. Martin-serrano, J., Li, K. & Bieniasz, P. D. Cyclin T1 Expression Is Mediated by a Complex and Constitutively Active Promoter and Does Not Limit Human Immunodeficiency Virus Type 1 Tat Function in Unstimulated Primary Lymphocytes. *J. Virol.* **76**, 208–219 (2002).

32. Marshall, R. M. *et al.* Cyclin T1 Expression Is Regulated by Multiple Signaling Pathways and Mechanisms during Activation of Human Peripheral Blood Lymphocytes. *J. Immunol.* **175**, 6402–6411 (2005).
33. Contreras, X., Barboric, M., Lenasi, T. & Peterlin, B. M. HMBA Releases P-TEFb from HEXIM1 and 7SK snRNA via PI3K / Akt and Activates HIV Transcription. *plos Pathog.* **3**, e146 (2007).
34. Fujita, T., Ryser, S., Piuz, I. & Schlegel, W. Up-Regulation of P-TEFb by the MEK1 – Extracellular Signal-Regulated Kinase Signaling Pathway Contributes to Stimulated Transcription Elongation of Immediate Early Genes in Neuroendocrine Cells \dagger . *Mol. Cell. Biol.* **28**, 1630–1643 (2008).
35. Kim, Y. K., Mbonye, U., Hokello, J. & Karn, J. T-Cell Receptor Signaling Enhances Transcriptional Elongation from Latent HIV Proviruses by Activating P-TEFb through an ERK-Dependent Pathway. *J. Mol. Biol.* **410**, 896–916 (2011).
36. Wei, B.-R. *et al.* Capacity for Resolution of Ras–MAPK-Initiated Early Pathogenic Myocardial Hypertrophy Modeled in Mice. *Comp. Med.* **61**, 109–118 (2011).
37. Abdellatif, M. *et al.* A Ras-Dependent Pathway Regulates RNA Polymerase II Phosphorylation in Cardiac Myocytes: Implications for Cardiac Hypertrophy. *Mol. Cell. Biol.* **18**, 6729–6736 (1998).
38. Lopaschuk, G. D. & Jaswal, J. S. Energy Metabolic Phenotype of the Cardiomyocyte During Development , Differentiation , and Postnatal Maturation. *J. Cardiovasc. Pharmacol.* **56**, 130–140 (2010).
39. Monroe, T. O. *et al.* Article YAP Partially Reprograms Chromatin Accessibility to Directly Induce Adult Cardiogenesis In Vivo YAP Partially Reprograms Chromatin Accessibility to Directly Induce Adult Cardiogenesis In Vivo. *Dev. Cell* **48**, 765-779.e7 (2019).
40. Gabisonia, K. *et al.* repair after myocardial infarction in pigs. *Nature* **569**, 418–422 (2019).
41. Engel, F. B., Schebesta, M. & Keating, M. T. Anillin localization defect in cardiomyocyte binucleation. *J. Mol. Cell. Cardiol.* **41**, 601–612 (2006).
42. Hesse, M. *et al.* New Methods in Cardiovascular Biology Midbody Positioning and Distance Between Daughter Nuclei Enable Unequivocal Identification of Cardiomyocyte Cell Division in Mice. *Circ. Res.* **123**, 1039–1052 (2018).
43. Kadow, Z. A. & Martin, J. F. A Rigorous Examination of the Midbody Provides Clarity. *Circ. Res.* **123**, 1012–1014 (2018).

Reviewers' comments:

Reviewer #1 (Remarks to the Author):

The manuscript continues to have two very different aspects. One is the very interesting observation that cyclin T1 is limiting in heart for MYCs proliferative capacity, whereas there is sufficient cyclin T1 in other tissues. Restoring cyclin T1 helps MYC regain proliferative capacity. This tells us something about regenerative capacity is regulated during organismal growth. As such, the manuscript could well be published in Nature Communications.

Unfortunately, this interesting observation is mixed with a series of mechanistic statements about MYC function and its relationship to CDK9 as well as data to describe whether MYC acts to specifically or globally regulate expression, which are either confusing or re-iterate data and conclusions, which the authors and others have by now repeatedly published in multiple previous publications and reviews.

Most importantly, the authors interpretation of gene expression data is simply confusing and misleading. Ectopically expressed MYC binds to all open promoters. After MYC activation, the hearts have increased in size when cyclin T1 is restored (and other organs also increase in size or cell number), so presumably mRNA levels have probably roughly doubled for all genes. So why not simply conclude that MYC binding to all active genes contributed to their increased activity? Why all this debate about common and non-common genes based on minimal differences with arbitrary cut-offs in relative expression when all genes go up? I am simply lost by the logic that only the binding to a subset is productive and the rest is "non-productive", only since some genes go up a bit more and a bit earlier than others? Cyclin B goes up later than cyclin D in these cells, for example, (since cells arrested in G0) so does that make cyclin B a "non-productive" MYC target and cyclin D a "productive" one? Are S-phase genes not targets since they are not induced at 4 hrs? In my view, Figure 3 and the subsequent discussion are as misleading as in the original version.

Frustratingly, the key data presented as argument against a global function are actually misleading. In Supplementary Figure 3c, the authors show total RNA quantity of cells, which measures the content in ribosomal RNA, which has nothing to do with the question of how MYC acts on RNAPII, since ribosomal RNA is made by polymerases I and III. The data are plotted on a per cell basis: an increase in this parameter requires that cells grow but do not divide, which is (i) unlikely and (ii) not a part of the amplifier model. In my view, Figure 5f, 6g,h unequivocally show that levels of all RNA molecules have gone up after MYC induction. As long as MYC binds to all active promoters and total mRNA levels in the cell population increase as cells grow and divide after MYC stimulation, it is very hard to see how one can reach the conclusion that only some of the binding is productive.

Furthermore, the authors make a very strong statement that MYC recruits CDK9 and cyclin T1 to regulate genes and never show this. The only data is to show that a CDK9 inhibitor stops gene activation by MYC, but CDK9 inhibition is well known to globally stop transcription elongation (Ref 8-10 quoted by the authors), so how should MYC activate genes in the presence of a CDK9 inhibitor? The same data are likely to hold for every component of the basal transcription machinery, including RNAPII itself.

Other comment

The experiments shown in Figures 4e and 6c look very similar

Reviewer #2 (Remarks to the Author):

I was previously quite enthusiastic about this work, but had concerns that the links between CDK9 and MYC were not very strong, and the detailed mechanistic inferences that were being made from experiments with long time frames. I am delighted to say that both of these concerns have been ameliorated in the revision, both by the inclusion of new data and by the detailed rebuttal and changes to the text.

Reviewer #3 (Remarks to the Author):

In this reviewer's opinion, the manuscript by Bywater et al has not been improved by the revisions. In attempting to satisfy all three reviewers some of the emphasis has been removed from the striking, clean observation that cyclin T/cdk9 levels determine the magnitude of the response to increasing levels of MYC. This is an interesting and important observation with biological and pathological ramifications and biochemical implications for transcription especially mechanisms of pausing, pause release, and early elongation. This neat result is muddled by inconclusive and poorly reasoned experiments to assess whether or not MYC is acting as a global amplifier of transcription—whether MYC is a global amplifier or a gene specific activator is not all that important here. I would recommend removing experiments and discussion of this matter and simply side-step the issue.

The experiment that the investigators conducted to test whether supraphysiological MYC is acting as a general amplifier was to take MEFs from R26+/+ vs. R26CMER/+ mice, serum-starve for 3 days, add tamoxifen to activate MYCER and monitor the rate of RNA increase attributable to the extra MYC activity. As noted in the rebuttal the additional MYC did not increase total RNA until 24-48 hours when the increase could reasonably be called indirect. Total RNA is mostly rRNA, and so this timeline should reflect mostly rRNA production. Let's reason our way through what to expect if MYC amplifies pre-existing cellular transcriptional programs without launching new ones. After 3 days without serum, both the R26+/+ and the R26CMER/+ MEFs are in a deep G0. These quiescent cells should not be proliferating and should be executing only maintenance programs that do not require new ribosome synthesis (ribosomes are VERY stable and essentially do not turnover under these conditions, so they will require little if any replacement). Now, let's push up MYC activity. Without other signals to launch the proliferative and ribosome synthetic programs, there is no substrate program(s) upon which MYC can act. In the amplifier metaphor, the volume is turned up, but there is no music playing. So, this experiment argues neither for nor against global amplification.

It should also be noted that the authors used DESEQ2 to analyze differential gene expression between their samples. DESEQ2 includes two strict types of normalization that are designed to suppress global differences in RNA levels in order to visualize relatively differentially expressed genes. DESEQ2 should not be used to assess amplification (see the Bioconductor website).

With respect to p-TEFb, the authors state in the introduction that "P-TEFb phosphorylates Serine 2 of the C-terminal Domain (CTD) of paused RNA PolII leading to productive elongation", This isn't right. While cdk9 phosphorylates Ser2, there is not much evidence that this is the key step leading to elongation. Rather, as I stated in my earlier review, there is much more evidence that phosphorylation of DSIF and NELF by CDK9 is actually the most important step in pause release. This needs to be corrected in the text, if not experimentally by looking at phospho-DSIF and NELF.

Also, increased transcription by raising cyclin T levels does not establish that MYC activates P-TEFb, rather it shows that whatever MYC does is upstream of, limited by cyclin T-CDK9 levels. The fact that

these experiments do not define the biochemical action of MYC should be explicitly acknowledged.

In summary, the authors should focus more narrowly on a cool and important observation and be agnostic about the pro- or con- amplifier implications that do not fundamentally affect their story.

Reviewer #4 (Remarks to the Author):

Bywater et al have submitted a revised version of their manuscript, describing that the responsiveness of cells to the oncogene Myc depends on the presence of P-TEFb, a transcriptional co-factor complex consisting of Cdk9 and cyclin T1.

The authors have responded to all criticisms raised and improved the manuscript considerably. They now show that within the heart very small numbers of endothelial immune and epicardial cells respond to global activation of Myc by expression of cell cycle markers. I still find it surprising that the number is so low, since endothelial cells are rather sensitive to Myc activity. Yet, it is probably indeed not helpful to discuss this issue more extensively. In addition, a quantification of the pH3 staining is now provided.

Importantly, the authors now demonstrate that overexpression of cyclin T1 in cardiomyocytes increase Cdk9 levels and also lead to increased phosphorylation of RNA PolII (S2). It is still puzzling to me that despite the presence of cyclin T1 in the heart of WT mice (at low levels), CDK9 is essentially absent. If cyclin T1 stabilizes CDK9 as claimed by the authors, why does that not happen in WT hearts? Is there a specific threshold of cyclin T1 that is required to stabilize CDK9? Despite these open questions, I was satisfied by the new data, which are now shown in Fig. 6b.

It still remains an open question why the authors detect only a very low phosphorylation of Pol II at ser 2 and ser5 in cardiomyocytes, although cardiomyocytes show a high degree of transcriptional activity. The authors speculate about potential reasons for the low phosphorylation of Pol II in cardiomyocytes in the response letter but did not discuss this apparent paradox in the manuscript, which is a shortcoming in my view. The reader should be informed about this.

On the other hand, the authors have now included data showing that transient elevation of MycER activity in 15-day old hearts increases heart size, most likely due to hyperplasia, leads to dedifferentiation of cardiomyocytes, and causes death already after 4 days post Myc induction. These findings are very surprisingly, since we and others made different observations after overexpression of Myc in cardiomyocytes. I assume that such differences might be due to different activity/expression levels of Myc.

Several technical issues that bothered me were addressed as well. The authors now present co-staining with PCM-1. The red channel for PCM-1 in Fig. 5e, particularly after overlay with the Ki67 signal, is rather weak (much better in Fig. 6e). It would be good to increase the intensity to clearly demonstrate that the Ki67-positive nuclei indeed express PCM-1. → I appreciate that the authors now also show images of isolated cardiomyocytes stained for p-H3. However, I also asked for a quantification of the number of isolated cardiomyocytes that are pH3-positive. This quantification is still missing as far as I gather.

In regard to the anillin and Aurora B staining: During bi-nucleation of cardiomyocytes (i.e. lack of cytokinesis), Aurora B still localizes to the midbody but anillin does not. This is precisely the reason why I asked for an anillin staining, since Aurora B does not distinguish between cytokinesis (i.e. true proliferation) and binucleation. The paper cited by authors (Engel et al, 2006) exactly makes this point. Hence, I was surprised that the authors used that paper to reject my criticism. Nevertheless, the authors provide now additional evidence for cardiomyocyte proliferation, which is why I do not insist on this staining.

Reviewers #1 and Reviewer #3 continue to highlight concerns regarding the specific versus global role of MYC, and we suggest that at this time it would be best to refocus your work and tone down these conclusions. In addition, Reviewer #4 has re-highlighted the need to directly quantify pH3 expression in isolated cardiomyocytes, among other concerns. Importantly, all concerns require a response, whether this requires additional analysis, discussion or experimentation. Please highlight all changes in the manuscript text file.

Reviewers' comments:

Reviewer #1 (Remarks to the Author):

The manuscript continues to have two very different aspects. One is the very interesting observation that cyclin T1 is limiting in heart for MYCs proliferative capacity, whereas there is sufficient cyclin T1 in other tissues. Restoring cyclin T1 helps MYC regain proliferative capacity. This tells us something about regenerative capacity is regulated during organismal growth. As such, the manuscript could well be published in Nature Communications.

Unfortunately, this interesting observation is mixed with a series of mechanistic statements about MYC function and its relationship to CDK9 as well as data to describe whether MYC acts to specifically or globally regulate expression, which are either confusing or re-iterate data and conclusions, which the authors and others have by now repeatedly published in multiple previous publications and reviews.

Most importantly, the authors interpretation of gene expression data is simply confusing and misleading. Ectopically expressed MYC binds to all open promoters. After MYC activation, the hearts have increased in size when cyclin T1 is restored (and other organs also increase in size or cell number), so presumably mRNA levels have probably roughly doubled for all genes. So why not simply conclude that MYC binding to all active genes contributed to their increased activity? Why all this debate about common and non-common genes based on minimal differences with arbitrary cut-offs in relative expression when all genes go up? I am simply lost by the logic that only the binding to a subset is productive and the rest is “non-productive”, only since some genes go up a bit more and a bit earlier than others? Cyclin B goes up later than cyclin D in these cells, for example, (since cells arrested in G0) so does that make cyclin B a “non-productive” MYC target and cyclin D a “productive” one? Are S-phase genes not targets since they are not induced at 4 hrs? In my view, Figure 3 and the subsequent discussion are as misleading as in the original version.

Frustratingly, the key data presented as argument against a global function are actually misleading. In Supplementary Figure 3c, the authors show total RNA quantity of cells, which measures the content in ribosomal RNA, which has nothing to do with the question of how MYC acts on RNAPII, since ribosomal RNA is made by polymerases I and III. The data are plotted on a per cell basis: an increase in this parameter requires that cells grow but do not

divide, which is (i) unlikely and (ii) not a part of the amplifier model. In my view, Figure 5f, 6g,h unequivocally show that levels of all RNA molecules have gone up after MYC induction.

As long as MYC binds to all active promoters and total mRNA levels in the cell population increase as cells grow and divide after MYC stimulation, it is very hard to see how one can reach the conclusion that only some of the binding is productive.

Furthermore, the authors make a very strong statement that MYC recruits CDK9 and cyclin T1 to regulate genes and never show this. The only data is to show that a CDK9 inhibitor stops gene activation by MYC, but CDK9 inhibition is well known to globally stop transcription elongation (Ref 8-10 quoted by the authors), so how should MYC activate genes in the presence of a CDK9 inhibitor? The same data are likely to hold for every component of the basal transcription machinery, including RNAPII itself.

We thank the reviewer for their comments. It is not our intention to rehash well-trodden ground arguing for or against the global amplifier model. However, due to the comments raised by yourself and two others during the initial review process in addition to the current relevance of this topic in the field of Myc biology, we considered it important to address this issue. Our experiments were not designed to address whether or not Myc acts as a global amplifier of transcription and we agree entirely with yourself and reviewer three that this is best approached in an agnostic way so not to detract from the overall message of the manuscript.

In order to achieve this, we have deleted data that addresses the global amplifier model from the results section of the manuscript (supplemental figures 3c to e and 6c). We agree with the caveats raised regarding the quantitation of RNA content in the MEFs. Although we find the relative abundance of premature mRNA versus mature mRNA informative, we consider that its inclusion disrupts the flow of the manuscript and detracts from the main conclusions. We have instead introduced the global amplifier model in the Discussion and raised the difficulty in drawing conclusions from our model given the normalization strategies employed in the analysis of bulk RNA. We hope that this provides the reader with sufficient information on potential alternate interpretations of the data presented.

“It has been proposed that stringent normalization strategies employed in the analysis of bulk RNA sequencing data can mask the more global impact of Myc on transcription that might be predicted from its promiscuous binding to open chromatin^{26,52}. Such general amplification is in no way excluded by our analyses. Nonetheless, it is clear that the magnitude of transcriptional change that the activation of ectopic Myc can instruct is significantly limited in tissues that lack regenerative potential, like the heart.”

Despite a reported direct association between Myc and the components of PTEFb (Eberhardy and Farnham 2002 JBC, Kanazowa 2003 Oncogene), we do not intend to claim that MYC recruits CDK9 and Cyclin T1 and have not attempted to demonstrate this in our own experiments. For this reason, we have been careful to state that Myc transcriptional

activity is limited by PTEFb rather than infer a direct association. This statement is supported by both Myc-dependent transcriptional attenuation in the presence of the CDK9 inhibitor and transcriptional activation promoted by ectopic expression of Cyclin T1.

Other comment

The experiments shown in Figures 4e and 6c look very similar

We have included a 'source data file' containing all the primary data presented within the manuscript to demonstrate the differences.

Reviewer #2 (Remarks to the Author):

I was previously quite enthusiastic about this work, but had concerns that the links between CDK9 and MYC were not very strong, and the detailed mechanistic inferences that were being made from experiments with long time frames. I am delighted to say that both of these concerns have been ameliorated in the revision, both by the inclusion of new data and by the detailed rebuttal and changes to the text.

We thank the reviewer for their time and critical revision of the manuscript.

Reviewer #3 (Remarks to the Author):

In this reviewer's opinion, the manuscript by Bywater et al has not been improved by the revisions. In attempting to satisfy all three reviewers some of the emphasis has been removed from the striking, clean observation that cyclin T/cdk9 levels determine the magnitude of the response to increasing levels of MYC. This is an interesting and important observation with biological and pathological ramifications and biochemical implications for transcription especially mechanisms of pausing, pause release, and early elongation. This neat result is muddled by inconclusive and poorly reasoned experiments to assess whether or not MYC is acting as a global amplifier of transcription—whether MYC is a global amplifier or a gene specific activator is not all that important here. I would recommend removing experiments and discussion of this matter and simply side-step the issue.

The experiment that the investigators conducted to test whether supraphysiological MYC is acting as a general amplifier was to take MEFs from R26+/+ vs. R26CMER/+ mice, serum-starve for 3 days, add tamoxifen to activate MYCER and monitor the rate of RNA increase attributable to the extra MYC activity. As noted in the rebuttal the additional MYC did not increase total RNA until 24-48 hours when the increase could reasonably be called indirect. Total RNA is mostly rRNA, and so this timeline should reflect mostly rRNA production. Let's reason our way through what to expect if MYC amplifies pre-existing cellular transcriptional programs without launching new ones. After 3 days without serum, both the R26+/+ and the R26CMER/+ MEFs are in a deep G0. These quiescent cells should not be proliferating and should be executing only maintenance programs that do not require new

ribosome synthesis (ribosomes are VERY stable and essentially do not turnover under these conditions, so they will require little if any replacement). Now, let's push up MYC activity. Without other signals to launch the proliferative and ribosome synthetic programs, there is no substrate program(s) upon which MYC can act. In the amplifier metaphor, the volume is turned up, but there is no music playing. So, this experiment argues neither for nor against global amplification.

We thank the reviewer for their comments. As stated in our response to Reviewer 1, It is not our intention to rehash well-trodden ground arguing for or against the global amplifier model. However, due to the comments raised by yourself and others during the initial review process we considered it prudent to address this issue. We agree that it is best approached in an agnostic way so not to detract from the overall message of the manuscript.

We refer you to our response to Reviewer 1. Briefly, all data that addresses the global amplifier model has been removed from the results section of the manuscript. Instead, we have restricted consideration of the global amplifier model to the Discussion.

It should also be noted that the authors used DESEQ2 to analyze differential gene expression between their samples. DESEQ2 includes two strict types of normalization that are designed to suppress global differences in RNA levels in order to visualize relatively differentially expressed genes. DESEQ2 should not be used to assess amplification (see the Bioconductor website).

We agree and have included the following in the discussion:

"It has been proposed that stringent normalization strategies employed in the analysis of bulk RNA sequencing data can mask the more global impact of Myc on transcription that might be predicted from its promiscuous binding to open chromatin."

With respect to p-TEFb, the authors state in the introduction that "P-TEFb phosphorylates Serine 2 of the C-terminal Domain (CTD) of paused RNA PolII leading to productive elongation", This isn't right. While cdk9 phosphorylates Ser2, there is not much evidence that this is the key step leading to elongation. Rather, as I stated in my earlier review, there is much more evidence that phosphorylation of DSIF and NELF by CDK9 is actually the most important step in pause release. This needs to be corrected in the text, if not experimentally by looking at phospho-DSIF and NELF.

We have altered the text in the following way;

"P-TEFb phosphorylates DSIF, NELF and Serine 2 of the C-terminal Domain (CTD) of paused RNA PolII leading to productive elongation"

Also, increased transcription by raising cyclin T levels does not establish that MYC activates P-TEFb, rather it shows that whatever MYC does is upstream of, limited by cyclin T-CDK9

levels. The fact that these experiments do not define the biochemical action of MYC should be explicitly acknowledged.

We agree, we have provided no data to suggest that MYC either directly activates or recruits P-TEFb. Our data purely supports the conclusion that Myc-driven transcriptional activation is dependent on P-TEFb levels. We have removed two references to P-TEFb activity that may have been misleading. Following description of the effects of ectopic expression of Cyclin T1 in purified cardiomyocytes we have included the following statement:

“Further experiments are required to determine the mechanistic basis of this dependency”

In summary, the authors should focus more narrowly on a cool and important observation and be agnostic about the pro- or con- amplifier implications that do not fundamentally affect their story.

We agree and hope that we have achieved this with the changes outlined above.

Reviewer #4 (Remarks to the Author):

Bywater et al have submitted a revised version of their manuscript, describing that the responsiveness of cells to the oncogene Myc depends on the presence of P-TEFb, a transcriptional co-factor complex consisting of Cdk9 and cyclin T1.

The authors have responded to all criticisms raised and improved the manuscript considerably. They now show that within the heart very small numbers of endothelial immune and epicardial cells respond to global activation of Myc by expression of cell cycle markers. I still find it surprising that the number is so low, since endothelial cells are rather sensitive to Myc activity. Yet, it is probably indeed not helpful to discuss this issue more extensively. In addition, a quantification of the pH3 staining is now provided.

Importantly, the authors now demonstrate that overexpression of cyclin T1 in cardiomyocytes increase Cdk9 levels and also lead to increased phosphorylation of RNA PolII (S2). It is still puzzling to me that despite the presence of cyclin T1 in the heart of WT mice (at low levels), CDK9 is essentially absent. If cyclin T1 stabilizes CDK9 as claimed by the authors, why does that not happen in WT hearts? Is there a specific threshold of cyclin T1 that is required to stabilize CDK9? Despite these open questions, I was satisfied by the new data, which are now shown in Fig. 6b.

It still remains an open question why the authors detect only a very low phosphorylation of Pol II at ser 2 and ser5 in cardiomyocytes, although cardiomyocytes show a high degree of transcriptional activity. The authors speculate about potential reasons for the low phosphorylation of Pol II in cardiomyocytes in the response letter but did not discuss this apparent paradox in the manuscript, which is a shortcoming in my view. The reader should be informed about this.

We have now included the following in the discussion of the manuscript;

“We observed that both total and phosphorylated levels of Rpb1 are significantly lower in the adult heart in comparison to the liver (Supplemental Fig 5b). However, we presume that the levels of Rpb1 are sufficient to allow the transcription of genes required to maintain heart function. Indeed, when we performed RNA PolII ChIP sequencing in the adult heart we detected heart specific genes loaded with significant amounts of RNA PolII. There are several potential mechanisms that may concentrate the available RNA PolII at specific locations. First, it is possible that the generation of “transcriptional hotspots” might concentrate essential factors at the relevant genes²⁵. Second, transcription in the presence of low Rpb1 may be aided by tissue-specific super enhancers that recruits a large portion of the enhancer-associated RNA PolII and its associated cofactors and chromatin regulators²⁶. Unfortunately, testing of these hypotheses lies outside the scope of this manuscript.”

On the other hand, the authors have now included data showing that transient elevation of MycER activity in 15-day old hearts increases heart size, most likely due to hyperplasia, leads to dedifferentiation of cardiomyocytes, and causes death already after 4 days post Myc induction. These findings are very surprisingly, since we and others made different observations after overexpression of Myc in cardiomyocytes. I assume that such differences might be due to different activity/expression levels of Myc.

We have developed 2 mouse models with different levels of MycER expression, Rosa26-MycER (low Myc) and R26-Cag-MycER (high Myc), and we indeed see phenotypic differences between these models. For instance, long-term activation high MycER (R26-Cag-MycER mice) in adult liver stimulates cell cycle entry in over 80 % of the hepatocytes within 24 hours. In contrast, in low MycER mice (Rosa26-MycER) less than 10% of the hepatocytes enter the cell cycle within the first 24 hours.

Acute activation of low MycER in hearts of 15 day old mice causes a small but significant increase in heart size by 72 hours post-activation. These mice survive beyond 7 days post Myc activation.

Several technical issues that bothered me were addressed as well. The authors now present co-staining with PCM-1. The red channel for PCM-1 in Fig. 5e, particularly after overlay with the Ki67 signal, is rather weak (much better in Fig. 6e). It would be good to increase the intensity to clearly demonstrate that the Ki67-positive nuclei indeed express PCM-1.→ I appreciate that the authors now also show images of isolated cardiomyocytes stained for p-H3. However, I also asked for a quantification of the number of isolated cardiomyocytes that are pH3-positive. This quantification is still missing as far as I gather.

We have altered Figure 5e to increase the intensity of PCM-1 and included quantitation of p-H3+ purified cardiomyocytes by flow cytometry in Figure 5h.

In regard to the anillin and Aurora B staining: During bi-nucleation of cardiomyocytes (i.e. lack of cytokinesis), Aurora B still localizes to the midbody but anillin does not. This is precisely the reason why I asked for an anillin staining, since Aurora B does not distinguish between cytokinesis (i.e. true proliferation) and binucleation. The paper cited by authors

(Engel et al, 2006) exactly makes this point. Hence, I was surprised that the authors used that paper to reject my criticism. Nevertheless, the authors provide now additional evidence for cardiomyocyte proliferation, which is why I do not insist on this staining.

We thank the reviewer for not insisting on Anillin staining. We did try Anillin staining with a commercial antibody (ab5910, Abcam) and 2 antibodies obtained from Christine Field, who has provided her antibodies to many researchers in the field. Unfortunately, Dr Field was unable to provide the original antibody that is cited in most publications but kindly provided 2 different antibodies raised to either the actin binding domain or the C-terminus. Technically we could not get any of these antibodies to work in our tissues.

REVIEWERS' COMMENTS:

Reviewer #1 (Remarks to the Author):

The authors have responded well to the suggestions to tone down the mechanistic implications and focus on their exciting biology. With the revisions, this is now an important manuscript that should be published as is.

Martin Eilers

Reviewer #3 (Remarks to the Author):

The authors have satisfactorily addressed my comments, and in my view the comments of the other referees as well. The work now constitutes a valuable and not over-wrought contribution.